# Anoctamin-1 is a core component of a mechanosensory anion channel complex in *C. elegans*

Wenjuan Zou[1,2,3,7] ✉, Yuedan Fan[2,3,4,7], Jia Liu[3,4,7], Hankui Cheng[2,3,4,7], Huitao Hong[2,3,4], Umar Al-Sheikh [3,4], Shitian Li[3,4], Linhui Zhu[2,3,4], Rong Li [2,3,4], Longyuan He[1], Yi-Quan Tang [5], Guohua Zhao [2], Yongming Zhang[6], Feng Wang[1], Renya Zhan [1], Xiujue Zheng[1] & Lijun Kang [1,2,3,4] ✉

Mechanotransduction channels are widely expressed in both vertebrates and invertebrates, mediating various physiological processes such as touch, hearing and blood-pressure sensing. While previously known mechanotransduction channels in metazoans are primarily cation-selective, we identified Anoctamin-1 (ANOH-1), the *C. elegans* homolog of mammalian calcium-activated chloride channel ANO1/TMEM16A, as an essential component of a mechanosensory channel complex that contributes to the nose touch mechanosensation in *C. elegans*. Ectopic expression of either *C. elegans* or human Anoctamin-1 confers mechanosensitivity to touch-insensitive neurons, suggesting a cell-autonomous role of ANOH-1/ANO1 in mechanotransduction. Additionally, we demonstrated that the mechanosensory function of ANOH-1/ANO1 relies on CIB (calcium- and integrin- binding) proteins. Thus, our results reveal an evolutionarily conserved chloride channel involved in mechanosensory transduction in metazoans, highlighting the importance of anion channels in mechanosensory processes.

Mechanotransduction is a fundamental process involved in numerous physiological functions, including touch, hearing, pain, proprioception and blood pressure regulation[1–3]. At the molecular level, certain ion channels transduce mechanical forces into electrical signals[1,3,4]. Ion channels such as PIEZO proteins, transmembrane channel-like protein TMC1, OSCA/TMEM63 channels, degenerin, epithelial, and acid-sensing Na[+] channels (DEG/ENAC/ASIC), N-type transient receptor potential (TRP) channels, and two-pore K[+] (K2P) channels have been identified as mechanogated ion channels in various tissues and

organisms in metazoans[1,3–6]. However, these channels are generally presumed to be either cation-selective or non-selective cation channels with some level of anion permeability[1–5].

Among the anion channels, the chloride channel superfamily has been reported in diverse functions and cellular processes such as cell volume regulation, muscle contraction, apoptosis and endocytosis[7]. In addition, chloride channelopathies encompass various disorders, such as cystic fibrosis, myotonia, and Bartter syndrome[7]. Based on activation mechanisms, chloride channels are mainly categorized into

[1]Department of Neurosurgery of the First Affiliated Hospital and School of Brain Science and Brain Medicine, Zhejiang University School of Medicine, Zhejiang, China. [2]Department of Neurology of the Fourth Affiliated Hospital, Zhejiang University School of Medicine, Hangzhou, China. [3]Liangzhu Laboratory, MOE Frontier Science Center for Brain Science and Brain-machine Integration, State Key Laboratory of Brain-machine Intelligence, Zhejiang University, Hangzhou, China. [4]NHC and CAMS Key Laboratory of Medical Neurobiology, School of Medicine, Zhejiang University, Hangzhou, Zhejiang, China. [5]State Key Laboratory of Medical Neurobiology, MOE Frontiers Center for Brain Science, Institutes of Brain Science, ENT Institute and Otorhinolaryngology, Department of Affiliated Eye and ENT Hospital, Key Laboratory of Hearing Medicine of NHFPC, Fudan University, Shanghai, China. [6]Department of Ophthalmology of the Fourth Affiliated Hospital, Zhejiang University School of Medicine, Hangzhou, China. [7]These authors contributed equally: Wenjuan Zou, Yuedan Fan, Jia Liu, Hankui Cheng. ✉e-mail: zouwenjuan2008@163.com; kanglijun@zju.edu.cn

voltage-gated, ligand-gated and calcium-gated channels[7]. Notably, SWELL1 (LRRC8A) is identified as an anion channel that regulates cell volume by activation through ionic strength[8]. Despite its crucial role, SWELL1 is not considered a primary mechanotransduction channel due to its activation kinetics being much slower than other mechanically activated channels[8]. Additionally, CFTR has been implicated to be necessary for the activation of airway epithelial cells in response to membrane stretching[9]. Further investigation suggests that CFTR may be activated by increased cell volume and hydrostatic pressure, but not by shear stress[10]. Moreover, evidence showing that ectopic or heterologous expression of CFTR confers mechanosensibility to mechanoinsensitive cells is still lacking[10]. The direct activation of chloride channels by mechanical forces and their potential functions in mechanosensation remain largely unknown.

Identifying novel mechanotransduction channels in mammalian systems consistently presents a formidable challenge[4,11,12]. Despite the advent of genetic manipulation technologies such as CRISPR-Cas9, which have made mammalian genetics much easier than ever, establishing large-scale genetic screens remains difficult. Furthermore, mechanoreceptor cells in mammals are sparse, and the expression levels of mechano-gated channel proteins are typically very low[4]. Additionally, the machinery involved in mechanotransduction often forms protein complexes, making functional reconstruction in heterologous systems extremely challenging[3,4,12]. For instance, while there is ample evidence indicating that the TMC1 complex serves as the mechanotransduction channel in the mammalian inner ear, reconstructing this complex in heterologous systems has yet to be achieved[3,13].

*Caenorhabditis elegans* exemplifies an ideal model organism for the discovery and elucidation of novel mechanotransduction channel candidates or complexes in a living system. In their intricate three-dimensional soil environment, they encounter external forces stemming from surface tension, collisions with adjacent soil particles and other organisms, as well as forces generated through their own locomotion[14]. Among their 302 neurons, hermaphroditic worms possess over 40 sensory neurons potentially tasked with detecting these forces, while males feature an additional 52 putative mechanoreceptor neurons[12,14,15]. A range of putative mechano-gated ion channels, encompassing PIEZO, TMC1, TRP-4, as well as ENaC channels such as MEC-4 and DEG-1, have been implicated in mechanotransduction within *C. elegans*[16–21]. Consequently, these nematodes exhibit a diverse array of behavioral responses to mechanical stimuli, including nose touch response, gentle touch response, harsh touch response, basal slowing response, head withdrawal response, proprioception, modulation of food intake and mating behaviors, and even the ability to "hear airborne sound"[14,15,17,19,21–24]. The tractable nature of behaviors observed in *C. elegans*, coupled with expedited genetic screening, opens avenues for the exploration of novel candidate channels and complexes[12,15,21,25]. Moreover, sophisticated functional assays, such as electrophysiological recordings and calcium imaging in live worms, facilitate the determination of whether a candidate protein serves as a primary mechano-gated channel in vivo[18–21].

In this study, we demonstrate that the ANOH-1, the *C. elegans* homolog of mammlian calcium-activated channel anoctamin-1 (TEME16A), is required for mechanical stimulation responses, thereby playing a critical role in avoidance behavior elicited by nose touch. Employing molecular genetics and electrophysiological recordings, we reveal that Anoctamin-1 is necessary for chloride-selective mechanoreceptor currents. Notably, mechanosensory impairments observed in *anoh-1* mutants can be effectively rescued through the expression of either nematode ANOH-1 or human ANO1, implying an evolutionarily conserved significance of ANOH-1 in mechanotransduction. Moreover, our findings reveal that the mechanosensory function of ANOH-1/ANO1 is dependent on the presence of CIB and ankyrin proteins, which are recognized as auxiliary subunits of the mechanotransduction TMC1 complex in the mammalian inner ear and the nematode mechanosensitive OLQ neurons[17,21,26–29]. In summary, our findings provide compelling evidence that Anoctamin-1, in collaboration with CIB and ankyrin, contributes to mechanosensory transduction.

## Results

### ANOH-1 is required for the nose touch-evoked calcium responses

*Caenorhabditis elegans* consists of hermaphrodites and males, and hermaphrodites can reproduce through self-fertilization[30]. Male mating requires copulation with hermaphrodites, relying heavily on sensory abilities such as chemosensation to detect pheromones and mechanosensation to locate the hermaphrodites and insert copulatory spicules into the vulva for sperm transfer[31–33]. Male worm mating relies on polycystin-2 (PKD-2), homologous to the human polycystic kidney disease gene PKD2, which is expressed in male-specific sensory neurons, including the cephalic male neurons (CEMs) in the head (Fig. 1a), type B ray neurons (RnBs) in the rays, and hook type B neurons (HOBs) in the hook[34,35]. Males with loss-of-function mutations in *pkd-2* exhibit significantly reduced responses to contact with hermaphrodites and difficulty in identifying the vulva[34,35]. Previous studies have shown that GFP-tagged PKD-2 extracellular vesicles (ECVs) are released from the tips of the nose, where the CEM cilia are located, and from the tips of the male tail rays, where the RnB cilia are present[36,37]. Additionally, mechanical stimulation induces the release of PKD-2-containing ECVs from cilia[36,37]. Our earlier research demonstrated that mechanical stimulation leads to substantial calcium increases in the RnB neurons[31]. We questioned whether CEM neurons could also be mechanically activated. We first performed calcium imaging of CEM neurons in response to nose touch stimulation using a genetically encoded calcium indicator GCaMP5[31]. We observed that mechanical force with a displacement of 15 μm evoked a robust calcium transient in the CEM neurons (Fig. 1b). We next sought to determine whether the nose touch-triggered calcium responses in CEM neurons primarily depend on cations or anions. Given that the cilia of CEM neurons are embedded in the cuticle, we dissected a section of the cuticle from the worm's head to improve the effectiveness of ion exchange. After dissection, the worm was immersed in either a Na$^+$-free or Cl$^-$-free bath solution for 20 minutes before calcium imaging. Interestingly, we found that the nose touch induced calcium increases of CEM neurons were not reduced in the Na$^+$-free bath solution (Fig. 1c). However, the calcium transients were eliminated in the Cl$^-$-free bath solution (Fig. 1c). Niflumic acid (NFA) is known to inhibit several anion channels, including Ca$^{2+}$-activated and swelling-activated Cl$^-$ channels, as well as various membrane ion transport systems like anion exchangers and TRP channels[7]. We observed that touch-evoked calcium transients in CEM neurons were eliminated, when the worm was dissected and immersed in normal bath solution with 50 μM NFA (Fig. 1c). We further screened chloride transporter mutant worms and identified the loss of potassium/chloride cotransporter KCC-3 or anion bicarbonate transporter ABTS-3 both reduced the calcium levels of CEM neurons in response to nose touch (Fig. 1d). Taken together, these results suggest that CEM neurons are mechanosensitive, and their mechanosensitivity is regulated by chloride ions.

We screened known chloride channel mutations including CLC-type chloride channels, anoctamin, bestrophin channels, and ligand-gated ionotropic receptors[38–41]. We found that ANOH-1 (UniProt identifier G5EBW3), a homolog of the mammalian calcium-activated chloride channel ANO1/TMEM16A, and BEST-2, a bestrophin homolog, contributed to the nose touch-evoked calcium increases in CEM neurons (Fig. 1e, f). Notably, the absence of *anoh-1* reduced the calcium transients of CEM neurons in response to nose touch, which was restored by the expression of either nematode ANOH-1 or human ANO1 in CEM neurons (Figs. 1e–g).

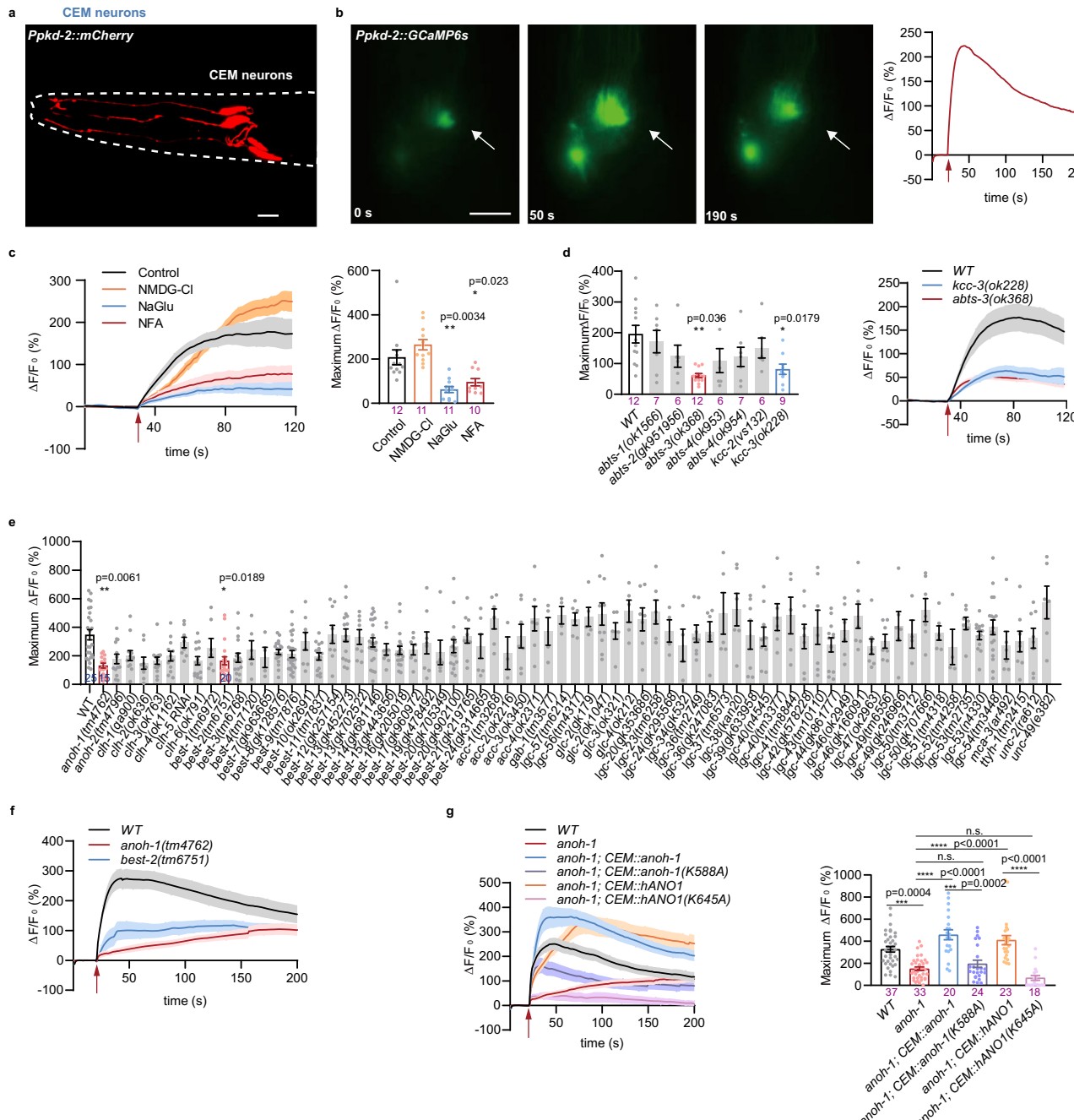

**Fig. 1 | Mechanical stimulation evokes ANO1-dependent calcium increases in CEM neurons. a** Micrograph showing four CEM neurons in the head of an adult male worm. Scale bars, 10 μm. **b** Nose touch evoked a calcium increase in CEM. Left: representative time-lapse images of GCaMP5.0- based calcium responses in CEM induced by nose touch; Right: calcium response indicated by fluorescence changes of GCaMP5.0. The arrows indicate the application of mechanical force (15 μm displacement). **c** The touch-evoked calcium responses in CEM were dependent on chloride ions and were abolished by NFA. Left: calcium responses; Right: maximum ΔF/F₀ changes. Solid lines show the average fluorescence changes and the shading indicates SEM. *P* values were calculated using the Brown-Forsythe and Welch ANOVA tests. **d** Chloride transporter mutants affected the nose touch-evoked calcium increases in CEM. Left: maximum ΔF/F₀ changes; Right: calcium responses. P values were calculated using the Brown-Forsythe and Welch ANOVA tests. **e, f** The

absence of ANOH-1 or BEST-2 significantly reduced the mechanically activated chloride channels in CEM. Calcium responses (**f**); Maximum ΔF/F₀ changes (**e**). P values were calculated using the Kruskal-Wallis test. **g** Transgenic expression of either nematode or human *anoctamin-1* genes rescued the nose touch-evoked calcium increases in CEM of *anoh-1(tm4762)* mutant male worms. Left: calcium responses; Right: maximum ΔF/F₀ changes. The mutations of *anoh-1(K588A)* and *ANO1(K645A)* will be further elucidated in the subsequent sections. P values were calculated using the Kruskal-Wallis test. Day 2 adult male animals were used in these experiments. The numbers of independent assays are indicated in each column of the panel. Each dot represents 1 animal. Data are presented as mean ± SEM. *ns* not significant, *P < 0.05, **P < 0.01, ***P < 0.001, and ****P < 0.0001. Source data are provided as a Source Data file.

Utilizing transgenic reporter lines, we found that ANOH-1 is expressed in ASH and ASJ sensory neurons other than CEM neurons (Fig. 2a). We observed that the CEM neurons were not labeled in this transgenic line. To address the possibility of an

incomplete expression pattern due to the chosen *anoh-1* promoter element, we employed the CRISPR/Cas9 knock-in method to insert an in-frame mNeonGreen cassette into the endogenous *anoh-1* locus[42]. Although the resulting mNeonGreen signal was

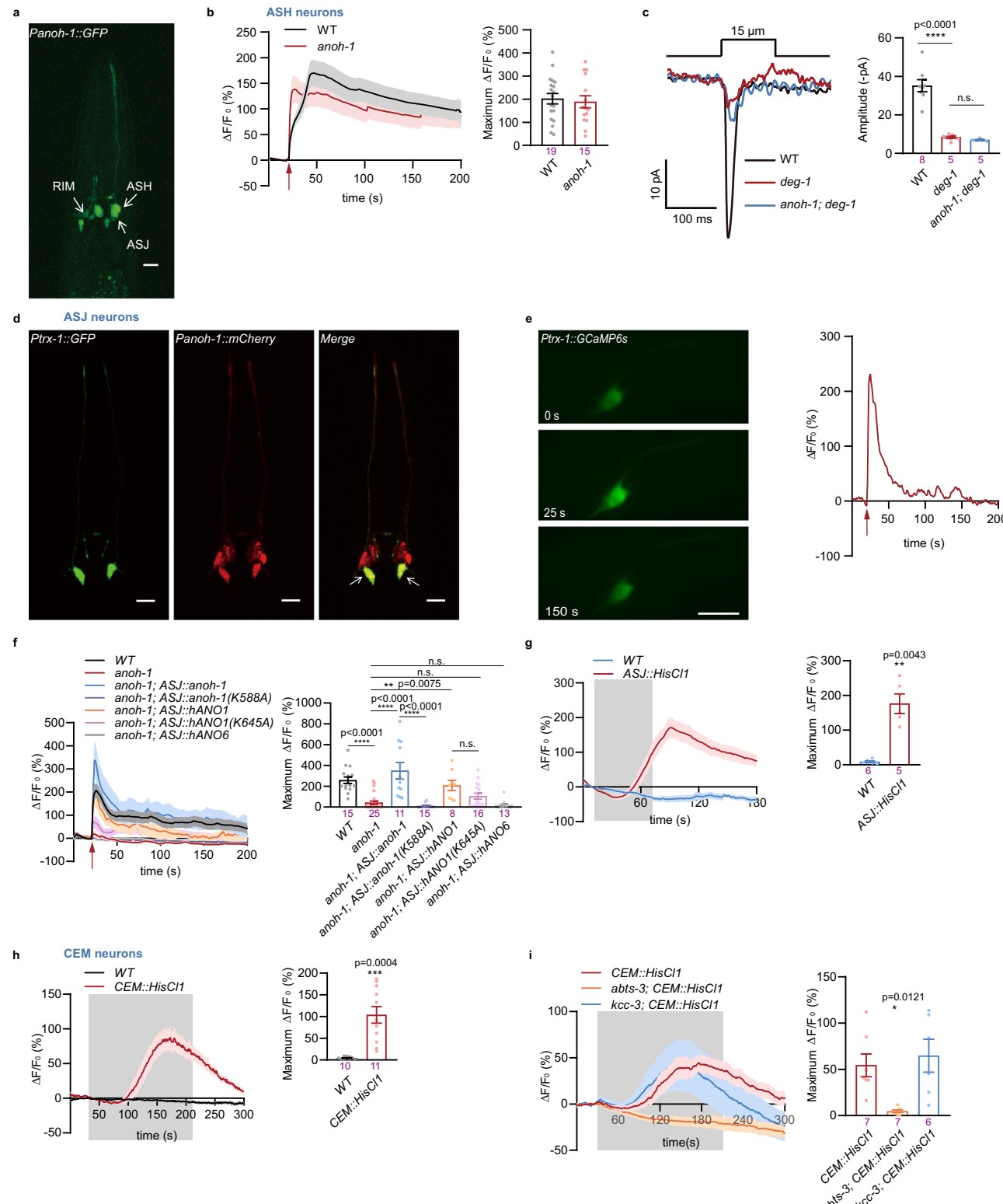

relatively faint, anti-mNeonGreen staining allowed us to detect the distribution of ANOH-1::mNeonGreen in the neurons (Supplementary Fig. 1a).

The ASH neurons are known to be the primary nociceptors that initiate avoidance behaviors upon chemical or mechanical stimulation[18,43–45]. Nevertheless, while we confirmed the previous report that DEG-1, an ENaC channel, is required for mechanotransduction in ASH neurons[18], we did not observe a role for ANOH-1 in either calcium increases or mechanoreceptor currents in ASH

neurons during nose touch (Fig. 2b, c). In contrast, tactile stimulation produced significant calcium increases in ASJ neurons, which were absent in *anoh-1* mutants (Fig. 2d–f). Expression of nematode ANOH-1 or human ANO1, but not human ANO6, specifically in ASJ neurons, successfully restored the calcium defects induced by the *anoh-1* mutation (Fig. 2e, f). Additionally, no observable defects in morphology were detected in either CEM or ASJ neurons of *anoh-1* mutants (Supplementary Fig. 1b, c). These findings suggest that the defects observed in nose touch-evoked calcium increases in these neurons in

**Fig. 2 | ANOH-1 is required for touch-evoked calcium responses in ASJ neurons.**
**a** Expression pattern of *anoh-1* in hermaphroditic worm. Scale bars, 10 μm. Repeated independently with 10 worms. **b** The nose touch-induced calcium increases in ASH. Left: calcium responses; Right: maximum ΔF/F₀ changes. Mechanical stimulation: 20 μm displacement. **c** MRCs in ASH require DEG-1 but not ANOH-1. Left: sample traces; Right: peak MRC amplitudes. Mechanical stimulation: 15 μm displacement. P values were calculated using the Brown-Forsythe and Welch ANOVA tests. **d** Micrograph showing two ASJ neurons of an adult hermaphroditic worm. Scale bars, 10 μm. Repeated independently with 10 worms. **e** Nose touch evoked a calcium increase in ASJ. Left: representative time-lapse images of GCaMP6s-based calcium responses in ASJ induced by nose touch; Right: calcium response. Mechanical force: 20 μm displacement. **f** Mechanical stimulation-induced calcium increases in ASJ. Left: calcium responses; Right: maximum ΔF/F₀ changes. The red arrow indicates the application of mechanical force (20 μm displacement). The

mutations of *anoh-1(K588A)* and *ANO1(K645A)* will be further elucidated in the subsequent sections. P values were calculated using the Kruskal-Wallis test. **g** Histamine-induced calcium responses in ASJ with the transgenetic expression of HisCl1 channel. Left: calcium responses; Right: maximum ΔF/F₀ changes. *P* values were calculated using the Mann-Whitney test. **h** Histamine-induced calcium responses in CEM with the transgenetic expression of HisCl1 channel. Left: calcium responses; Right: maximum ΔF/F₀ changes. P values were calculated using the Welch's *t* test. **i** Histamine-induced calcium responses in CEM in chloride transporter mutants. Left: calcium responses; Right: maximum ΔF/F₀ changes. P values were calculated using the Brown-Forsythe and Welch ANOVA tests. The numbers of independent assays are indicated in each column of the panel. Each dot represents 1 animal. Data are presented as mean ± SEM. *ns* not significant, **P* < 0.05, ***P* < 0.01, ****P* < 0.001, and *****P* < 0.0001. Source data are provided as a Source Data file.

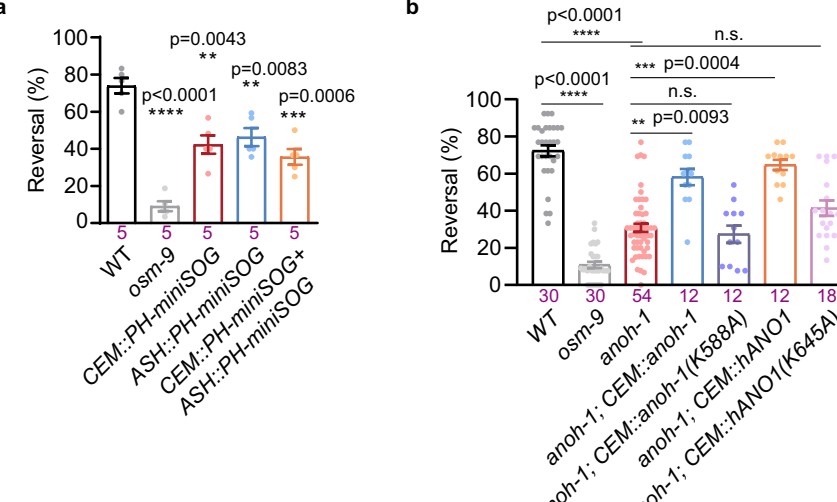

**Fig. 3 | ANOH-1 is required for nose touch behavior. a** CEM neurons were essential for the nose-touch behaviors of male worms. Each dot represents 1 independent assays, and 10 animals were used for each assay. P values were calculated using Brown-Forsythe and Welch ANOVA test. **b** ANOH-1 was essential for avoidance behaviors in response to nose touch. Each dot represents 1 independent assays, and 10–15 animals were used for each assay. P values were calculated using Kruskal-Wallis test. The mutations of *anoh-1(K588A)* and *ANO1(K645A)* will be further elucidated in the subsequent sections. Data are presented as mean ± SEM. *ns*: not significant, **P* < 0.05, ***P* < 0.01, ****P* < 0.001, *****P* < 0.0001. Source data are provided as a Source Data file.

*anoh-1* mutants are unlikely to be due to abnormalities in neuronal development.

The opening of a chloride channel can either excite or inhibit cells, depending on the equilibrium potential for chloride ions, which is determined by the chloride gradient across the plasma membrane, and the cell's resting membrane potential[46,47]. Mechanical stimulation can trigger significant calcium increases in CEM and ASJ neurons, promoting an excitatory effect due to chloride efflux. To test this hypothesis, we co-expressed the histamine-gated chloride channel HisCl1 along with the calcium indicator GCaMP in CEM and ASJ neurons[48]. Upon activation of HisCl1 by histamine, we observed robust calcium increases in both CEM and ASJ neurons (Fig. 2g, h). This indicates that the depolarization of these neurons resulted from chloride ion efflux, activating voltage-gated calcium channels and consequently raising intracellular calcium levels. Interestingly, while the *abts-3* mutation disrupted histamine-HisCl1 triggered calcium responses in CEM neurons, the absence of KCC-3 did not alter these responses (Fig. 2h, i). Prior research has indicated that KCC-3 is endogenously expressed in glial cells and influences the structural characteristics of neuronal receptive endings[49,50]. These observations, along with findings from this study, suggest that the reduced calcium responses to nose-touch stimuli in CEM neurons of *abts-3* mutants, but not *kcc-3* mutants, are likely due to alterations in intracellular chloride concentration within these neurons.

## ANOH-1 is critical for mechanosensory behavior

We then investigated the physiological role of ANOH-1. We used the genetically encoded photosensitizer miniSOG (mini Singlet Oxygen Generator) to kill specific neurons in male worms[51]. Consistent with previously reported[43,52], the wild-type worms, but not *osm-9* mutants, exhibit avoidance behavior in response to nose-touch stimulation (Fig. 3a). However, the nose touch-avoidance response was defective when either CEM neurons, the primary nociceptor ASH neurons, or both were ablated, emphasizing the role of CEM neurons in mechanosensation (Fig. 3a). We further observed that *anoh-1* mutant worms had significantly reduced nose-touch avoidance responses, which were rescued by the expression of nematode ANOH-1 in CEM neurons (Fig. 3b). Next, we expressed human ANO1 (cDNA clone IMAGE:4837404) in CEM neurons. Although part of the N-terminal (1–265 aa) and a few amino acids (476–501 aa) between TM3 and TM4 are truncated in this hANO1, it successfully rescued the nose-touch avoidance responses in *anoh-1* mutant worms (Fig. 3b). Altogether, our findings demonstrate that ANOH-1 is involved in mechanosensation, and this function may be evolutionarily conserved.

A recent study has reported that PEZO-1, a nematode homolog of the mammalian mechano-gated Piezo channels, regulates pharyngeal pumping and food intake in *C. elegans*[16]. Additionally, the mechanosensitive channel TMC-1 has been found to regulate egg-laying in worms[21,53]. We observed that *anoh-1* mutant worms exhibited

increased pumping frequencies compared to wild-type worms (Supplementary Fig. 2a), indicating that ANOH-1 is involved in the regulation of pharyngeal pumping and food intake. However, we did not observe any defects in male mating efficiency or egg-laying in *anoh-1* mutants (Supplementary Fig. 2b, c). This suggests that, while CEM neurons are essential for pheromone sensing[32,54], they may not play a mechanosensory role in mating behaviors. Additionally, the absence of *anoh-1* does not impact locomotion abilities (Supplementary Fig. 2d, e), indicating that the observed defects in nose touch behavior and accelerated pharyngeal pumping in *anoh-1* mutant worms are unlikely to be attributed to alterations of basal neuronal activity.

## ANOH-1 mediates mechanoreceptor currents (MRCs)

To further verify the mechanosensory properties of ANOH-1, we aimed to record MRCs in living worms. However, due to the significantly more slender body of adult males compared to hermaphrodites,

exposing the soma of CEM neurons for recording in living worms proved to be exceedingly difficult. We then conducted whole-cell patch clamp recordings on ASJ neurons in hermaphrodites (Fig. 4a)[55]. Nose touch evoked robust transient currents in ASJ neurons with a very short latency ($3.35 \pm 0.26$ ms with a 15 μm displacement of the touch probe towards the worm's nose tip) (Fig. 4b, c). One of the distinctive features of mechano-gated channels is that their activation latency is usually shorter than that of known fastest second-messenger pathways, typically less than 5–20 ms, which excludes the involvement of second messengers in channel gating[11,18,19]. Thus, the rapid activation kinetics and short latency of MRCs in ASJ neurons imply that the transduction channel responsible for these responses is directly mechanogated.

Notably, MRCs observed in ASJ neurons were absent in *anoh-1* mutants, but this defect was restored by the expression of nematode ANOH-1 or human ANO1, but not human ANO6, specifically in ASJ

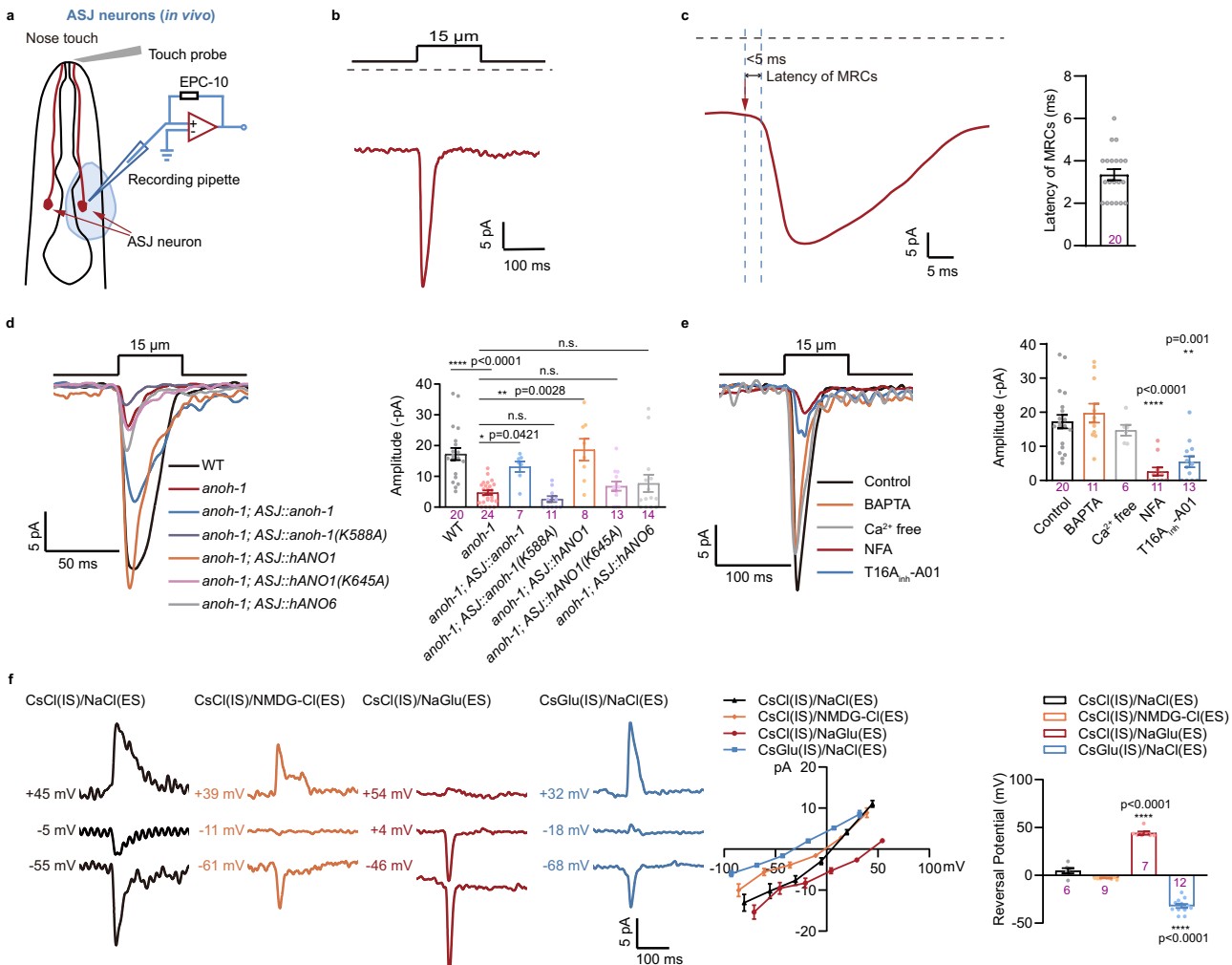

**Fig. 4 | ANOH-1 mediates Ca²⁺-independent mechanoreceptor currents in ASJ neurons. a** Schematic illustration of mechanoreceptor currents (MRCs) recording in ASJ neurons from a dissected worm. A glass probe with a diameter of about 10 μm was placed near the cilium of the ASJ neuron and mechanical force was applied by a piezo actuator. **b** Representative MRC in ASJ evoked by mechanical stimulation with 15 μm displacement. Holding potential: −70 mV. **c** Latency of MRCs in ASJ. Left: sample trace. Right: latency values. Mechanical stimulation: 15 μm displacement. Holding potential: −70 mV. **d** MRCs in ASJ. Left: sample trace. Right: the amplitude of MRCs. Mechanical stimulation: 15 μm displacement. Holding potential: −70 mV. P values were calculated using Kruskal-Wallis test. **e** MRCs in ASJ were independent of extra- and intracellular Ca²⁺, and blocked by NFA and T16Ainh-A01. Left: sample traces. Right: peak MRC amplitudes. Holding potential: −70 mV. P values were

calculated using Kruskal-Wallis test. **f** MRCs in ASJ neurons were dependent on both intracellular and extracellular chloride concentrations. Left: representative traces. The cell membrane was initially voltage-clamped at −50 mV, 0 mV, and 50 mV, respectively, and the displayed voltages have been corrected posthoc for liquid junction potentials (LJPs). Middle: *I–V* relationship of MRCs in ASJ. Peak current values were used here and throughout the manuscript. Right: the reversal potentials of MRCs. Mechanical stimulation: 15 μm displacement. P values were calculated using Brown-Forsythe and Welch ANOVA tests. Day 2 adult hermaphroditic animals were used in these experiments. Each dot represents 1 animal. Data are presented as mean ± SEM. *ns* not significant, *P < 0.05, **P < 0.01, ***P < 0.001, ****P < 0.0001. Source data are provided as a Source Data file.

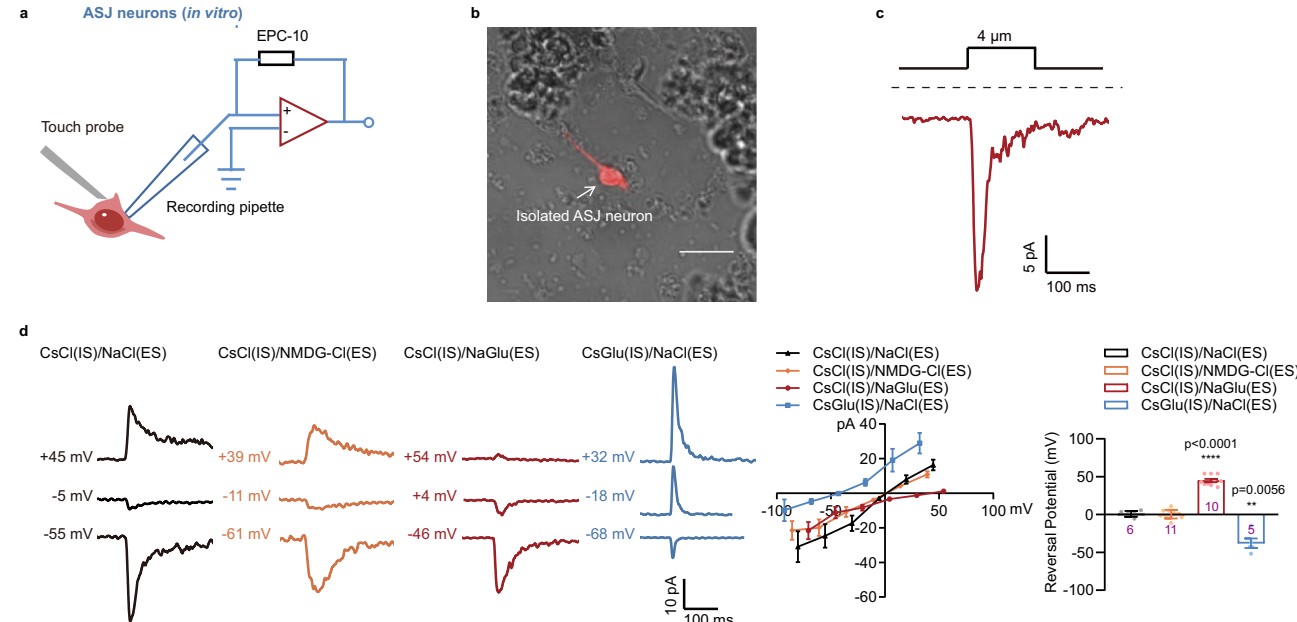

**Fig. 5 | MRCs recorded in the isolated ASJ neurons. a** Schematic illustration of MRCs recording in the isolated ASJ neurons. **b** An isolated ASJ neuron was labeled by *Ptrx-1::mCherry*. Scale bar, 10 μm. Repeated independently with 10 worms. **c** A representative MRC recorded from an isolated ASJ neuron. Mechanical stimulation: 4 μm displacement. Holding potential: −70 mV. **d** MRCs recorded in isolated ASJ neurons were dependent on both intracellular and extracellular chloride concentrations. Left: representative traces of MRCs. The cell membrane was initially voltage-clamped at −50 mV, 0 mV, and 50 mV, with the displayed voltages corrected posthoc for liquid junction potentials (LJPs). Middle: I–V relationship of MRCs. Right: the reversal potentials of MRCs. Mechanical stimulation: 4 μm displacement. Each dot represents 1 single cell. P values were calculated using Brown-Forsythe and Welch ANOVA tests. Data are presented as mean ± SEM. *ns* not significant, \**P* < 0.05, \*\**P* < 0.01, \*\*\**P* < 0.001, \*\*\*\**P* < 0.0001. Source data are provided as a Source Data file.

neurons (Fig. 4d). The human ANO1 (TMEM16A) gene encodes a Ca²⁺-activated chloride channel (CaCC) that plays important roles in various physiological functions, including epithelial secretion, muscle contraction, gastrointestinal motility, and sensory signal transmission[41,56–58]. We did not observe any calcium increases in ASJ neurons in response to nose touch in Ca²⁺-free bath solution (Supplementary Fig. 3). However, MRCs in ASJ neurons were not reduced in either Ca²⁺-free bath solution or Ca²⁺-free pipette solution (Fig. 4e). Further, the human ANO1 can be inhibited by NFA and a TMEM16A-specific inhibitor T16Ainh-A01[56]. We observed that both NFA and T16Ainh-A01 diminished MRCs in ASJ neurons in response to nose touch (Fig. 4e). To confirm that MRCs in ASJ neurons are mediated by chloride ions, we employed a traditional method of studying anion selectivity by replacing chloride with gluconate in the bath solution or pipette solution. Although substituting extracellular sodium with NMDG did not affect the reversal potential of MRCs in ASJ neurons, replacing extracellular chloride with gluconate caused a rightward shift in the reversal potential. In contrast, substituting intracellular chloride with gluconate resulted in a leftward shift of the reversal potential (Fig. 4f).

We further carried out embryonic cultures and recorded MRCs in isolated ASJ neurons[44] (Fig. 5a, b). Consistent with our in vivo recordings, mechanical stimulation induced MRCs in isolated ASJ neurons, and the reversal potential was influenced by the chloride concentration in both the bath and pipette solutions (Fig. 5c, d).

Taken together, these findings suggest that MRCs in ASJ neurons are facilitated by the ANOH-1 channel, and its activation through tactile stimulation is not dependent on calcium ions. Notably, we observed residual MRCs in ASJ neurons of *anoh-1* mutants, indicating that some other mechanogated channel(s) might be expressed in these neurons.

### Ectopic expression of ANOH-1/ANO1 confers mechanosensitivity on mechano-insensitive neurons

The head neurons ASK and ASI are chemosensors that do not detect mechanical stimulation[17,21]. We then tested whether the expression of

nematode ANOH-1 or human ANO1 conferred ectopic mechanosensitivity to these mechano-insensitive neurons. We recorded robust MRCs upon nose touch in both ASK and ASI neurons with ectopic expression of either nematode ANOH-1 or human ANO1 (Fig. 6a; Supplementary Fig. 4a). Like ANOH-1-mediated MRCs in ASJ neurons, MRCs recorded in ASK neurons ectopically expressing ANOH-1 were independent of Ca²⁺, and blocked by NFA and T16Ainh-A01 (Fig. 6b). In addition, the reversal potentials of MRCs in ASK and ASI neurons with ectopic expression of ANOH-1 were chloride-dependent, varying according to chloride concentrations in both the intracellular solution and pipette solution (Fig. 6c; Supplementary Fig. 4b, 5a). MRCs in ASK neurons expressing human ANO1 show weak outward rectification, similar to the nematode ANOH-1 mediated touch currents in ASJ, ASI, and ASK neurons (Supplementary Fig. 5b). These findings further confirm that ANOH-1 mediates mechanoreceptor currents.

Interestingly, we did not observe nose touch-evoked calcium increases in ASK neurons with the ectopic expression of ANOH-1 (Supplementary Fig. 6). This is not unexpected, as the intracellular chloride concentration may differ among ASJ, CEM, and ASK neurons. Due to this variation, the opening of ANOH-1 by mechanical stimulation may not depolarize ASK neurons under physiological conditions.

### The K588 amino acid residue is crucial for the mechanotransductive function of ANOH-1

We next used phylogenetics to analyze ANOH-1 protein homologs across *Caenorhabditis* species, *Mus musculus* and *Homo sapiens*. We noticed that the positively charged residues K and R between the TM5-TM6 regions are highly conserved (Fig. 6d), which are implicated are implicated in chloride ion permeability[59]. We then generated mutant worms by altering these positively charged amino acids to either negative or uncharged (Fig. 6d, e). When ANOH-1 with either K588E or K588A mutations was ectopically expressed in ASK

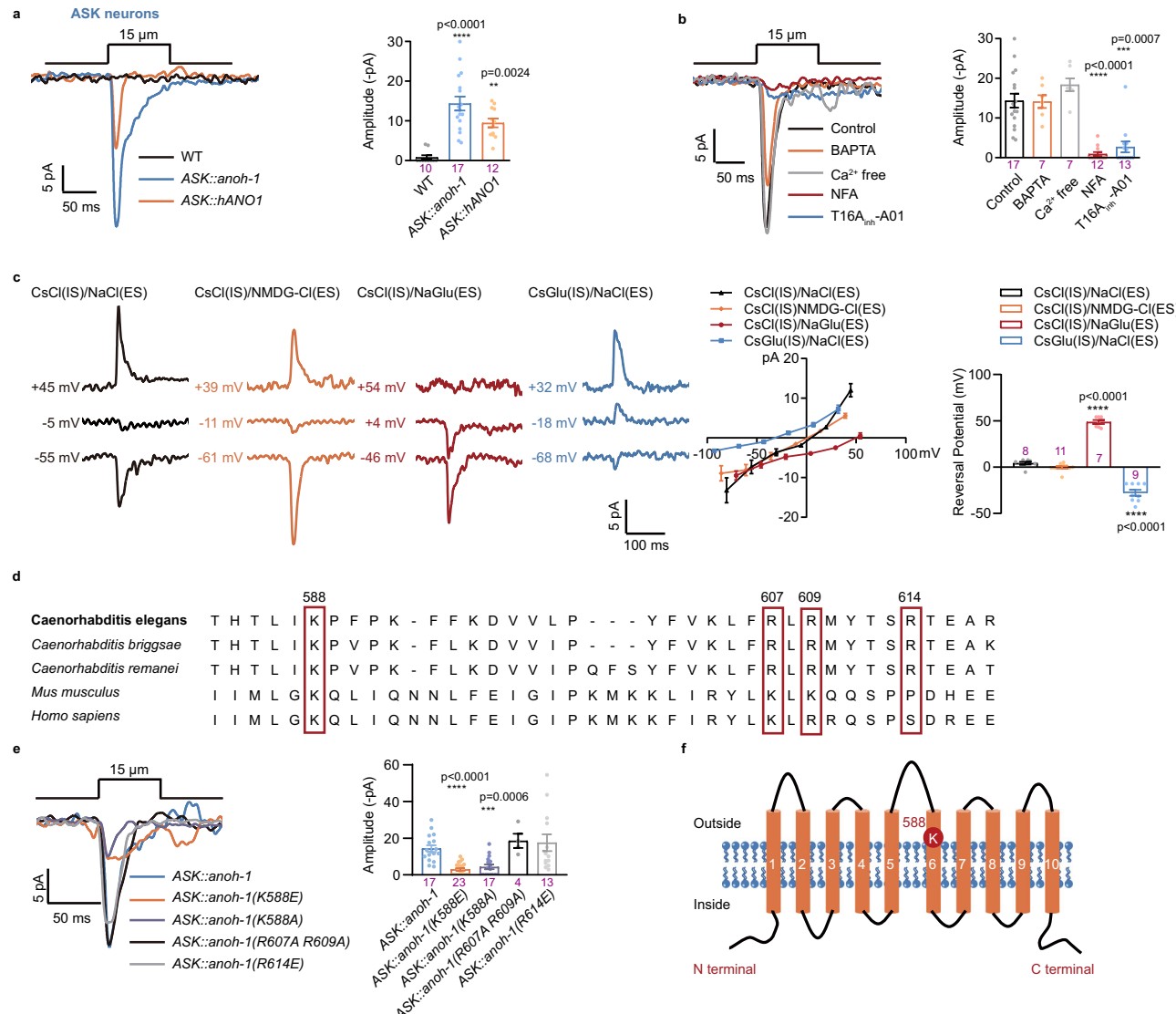

**Fig. 6 | Ectopic expression of nematode ANOH-1 or human ANO1 confers mechanosensitivity to ASK neurons. a** MRCs of wild-type ASK neurons or ASK neurons ectopically expressing nematode ANOH-1 or human ANO1. Left: sample traces; Right: peak MRC amplitudes. Mechanical stimulation: 15 μm displacement. Holding potential: −70 mV. P values were calculated using Kruskal-Wallis test. **b** MRCs in ASK ectopically expressing ANOH-1. Left: sample traces; Right: peak MRC amplitudes. *P* values were calculated using Kruskal-Wallis test. **c** MRCs recording in ASK neurons ectopically expressing ANOH-1. Left: representative traces of MRCs. The cell membrane was initially voltage-clamped at −50 mV, 0 mV, and 50 mV, and the displayed voltages were subsequently corrected for liquid junction potentials (LJPs). Middle: I-V relationship of MRCs. Right: the reversal potential of MRCs. Mechanical stimulation: 15 μm displacement. *P* values were calculated using Brown-Forsythe and Welch ANOVA tests. Mutations in the predicted pore region of ANOH-1 abrogated mechanosensitivity of ASK neurons with ectopic expressing of ANOH-1. **d** Sequence alignment of the putative pore region of ANOH-1 and its homologs. The positively charged residues K and R between TM5-TM6 regions are highly conserved among ANOH-1 and its homologs. Homology alignment was performed using the MEGA11. **e** No MRC was recorded in ASK neurons ectopically expressing ANOH-1 with either K588E or K588A mutations. Left: sample traces. Right: peak MRC amplitudes. Mechanical stimulation: 15 μm displacement. Holding potential: −70 mV. **f** Membrane topology of ANOH-1 and the location of positively charged K588 residue. *P* values were calculated using Kruskal-Wallis test. Day 2 adult hermaphroditic animals were used in these experiments. Each dot represents 1 animal. Data are presented as mean ± SEM. *ns* not significant, **P < 0.01, ***P < 0.001, ****P < 0.0001. Source data are provided as a Source Data file.

neurons, no MRC was recorded. This suggests that the amino acid residue K588 is crucial for the mechanotransduction mediated by ANOH-1 (Fig. 6d–f). Based on sequence alignment, K588 in ANOH-1 corresponds to K645 in human ANO1, which situates near the hydrophobic gate of ANO1 and facilitates an electrostatic basis for chloride ion permeation[60]. Consistently, the expression of wild-type ANOH-1, but not the ANOH-1(K588A) mutation, or wild-type human ANO1, but not the ANO1(K645A) mutation, effectively restored touch-evoked responses in CEM and ASJ neurons and corrected nose touch behavioral defects within the *anoh-1* mutant background (Figs. 1g, 2f, 3b, 4d).

## The mechanotransductive function of ANOH-1/ANO1 requires CIB proteins

Our results demonstrate that the absence of ANOH-1 did not affect the mechanically evoked responses in ASH neurons (Fig. 2b, c), despite the inherent mechanosensitivity of ASH neurons[17,21] and the expression of ANOH-1 in these neurons (Fig. 2a). This suggests that the presence of auxiliary subunit(s) is likely essential for the mechanosensitive function of ANOH-1/ANO1.

Through biased genetic screening, we found that worms carrying a loss-of-function mutation in *calm-1*, a gene encoding a *C. elegans* homolog of calcium- and integrin-binding (CIB) proteins, exhibited

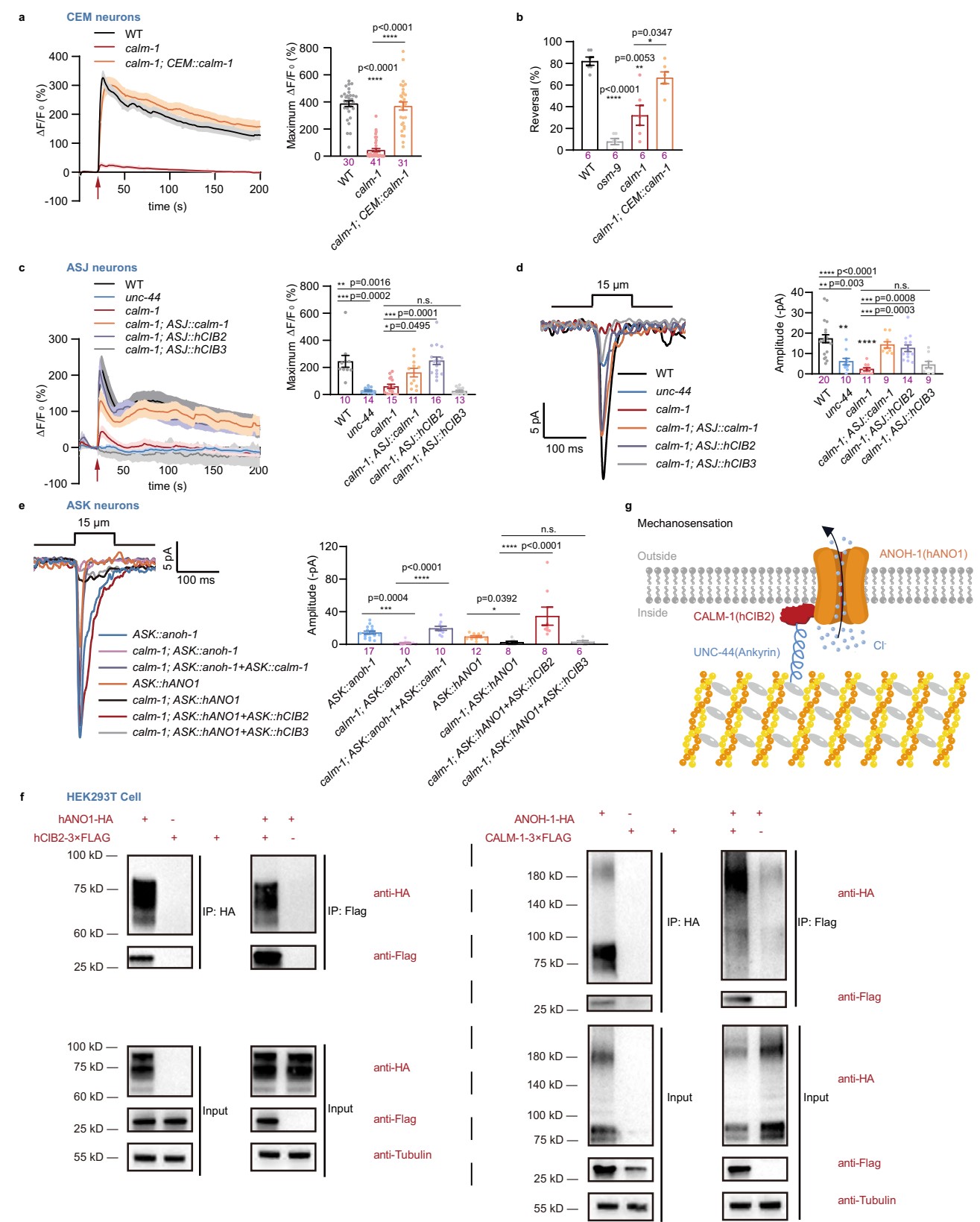

reduced touch-evoked calcium responses in CEM neurons (Fig. 7a). Furthermore, we consistently observed a significant reduction in nose touch responses in *calm-1* mutants (Fig. 7b). Importantly, these defects were effectively rescued by specifically restoring CALM-1 expression in CEM neurons (Fig. 7a, b). Similarly, in the *calm-1* mutant, the touch-evoked calcium increases and MRCs of ASJ neurons were also reduced,

and these were rescued by the expression of either worm CALM-1 or human CIB2, but not human CIB3, in ASJ neurons (Fig. 7c, d). It is worth noting that previous studies have reported CALM-1 expression in certain neurons, such as OLQ and HSN, but not in ASH neurons[17,21], which may explain why we did not observe any impact of ANOH-1 on the mechanosensitivity of ASH. To further verify this, we generated a

**Fig. 7 | CIB proteins are required for the mechanosensory function of ANOH-1/ ANO1. a** The touch-evoked calcium increases in CEM. Each dot represents 1 animal. P values were calculated using Kruskal-Wallis test. **b** Nose touch induced reversals of Day 2 adult male animals. Each dot represents 1 independent assays, and 10 ~ 15 animals were used for each assay. P values were calculated using Brown-Forsythe and Welch ANOVA tests. **c** The touch-evoked calcium increases in ASJ. Expression of either nematode CALM-1 or human CIB2, but not human CIB3, in ASJ neurons was able to rescue the defects observed in the *calm-1* mutants. Left: calcium responses; Right: maximum $\Delta F/F_0$ changes. Mechanical stimulation: 15 μm displacement. Each dot represents 1 animal. P values were calculated using Kruskal-Wallis test. **d** MRCs in ASJ. Left: calcium responses; Right: maximum $\Delta F/F_0$ changes. Each dot represents 1 animal. P values were calculated using Kruskal-Wallis test. **e** MRCs were detected in the ASK neurons of *calm-1* mutant worms that ectopically expressed

nematode ANOH-1 with CALM-1 or human ANO1 with CIB2. Left: sample traces; Right: peak MRC amplitudes. Each dot represents 1 animal. P values were calculated using Kruskal-Wallis test. **f** Co-IP experiments with HEK293T cells showing the direct binding of exogenously expressed nematode ANOH-1 with CALM-1, or human ANO1 with CIB2. Repeated for twice. **g** ANOH-1/ANO1 acts as a core component of a mechanosensory anion channel complex. The CIB and ankyrin proteins are likely auxiliary components of the mechanotransduction channel complex. Ectopic expression of ANOH-1 or human ANO1 channels confers mechanosensitivity to mechanically insensitive neurons through a Calm-1/CIB2-dependent mechanism, suggesting that the role of Anoctamin 1 in mechanotransduction is evolutionarily conserved. Data are presented as mean ± SEM. **ns**: not significant, *$P < 0.05$, **$P < 0.01$, ***$P < 0.001$, ****$P < 0.0001$. Source data are provided as a Source Data file.

transgenic line expressing mCherry driven by a *calm-1* promoter. We observed CALM-1 expression in ASI and ASK neurons. However, no CALM-1 expression was detected in ASH, ASJ, or CEM neurons in this reporter line (Supplementary Fig. 7a-c). Subsequently, we generated a *calm-1::mNeonGreen* knock-in line, and anti-mNeonGreen staining confirmed the distribution of CALM-1::mNeonGreen in ASJ and CEM neurons, but not in ASH neurons (Supplementary Fig. 7d-f).

TMC1, a pore-forming subunit of the mechanotransduction complex responsible for mammalian hearing and *C. elegans* touch sensation, is believed to rely on intracellular auxiliary proteins CIB and ankyrin for mechano-sensing[17,21,27,28,61-63]. Interestingly, the touch-evoked calcium increases and MRCs of ASJ neurons were also absent in *unc-44*, an ankyrin homolog, mutant worms (Fig. 7c, d).

To validate whether CALM-1 is required as an auxiliary subunit for the mechanosensitivity of ANOH-1/ANO1, we expressed ANOH-1/ANO1 in ASK neurons of *calm-1* mutant worms. Compared to the wild type, ASK neurons in *calm-1* mutant worms that ectopically expressed either nematode ANOH-1 or human ANO1 did not exhibit detectable MRCs (Fig. 7e). We therefore co-expressed either CALM-1 along with ANOH-1 or ANO1 with CIB2 and found that ASK neurons responded to nose touch stimulation in the background of *calm-1* mutants (Fig. 7e).

We next confirmed the interaction of ANOH-1 (ANO1) and CALM-1 (CIB2) using a co-immunoprecipitation (Co-IP) assay. The Co-IP experiments with HEK293T cells showed the direct interaction between exogenously expressed nematode ANOH-1 and CALM-1, as well as between human ANO1 and CIB2 (Fig. 7f). The molecular weight of human TMEM16A (695 amino acids) used in this study is about 82 kD with an HA-tag, which is comparable to that of nematode ANOH-1 (840 amino acids) with an HA-tag, which has a molecular weight of 99 kD.

We then transfected nematode ANOH-1 or human ANO1 into HEK-P1KO cells (Piezo1 knockout human embryonic kidney cells)[64]. Consistent with previous studies, outward rectifying calcium-activated currents were recorded in HEK-P1KO cells transfected with human ANO1 (Supplementary Fig. 8a-c). However, no MRC was recorded in these cells. Additionally, neither calcium-activated currents nor MRCs were recorded in HEK-P1KO cells transfected with nematode ANOH-1. We next co-transfected human ANO1 with human CIB2 and UNC-44, or nematode ANOH-1 with nematode CALM-1 and UNC-44, and no MRC was detectable (Supplementary Fig. 8d). We co-transfected either EGFP-hANO1 or EGFP-ANOH-1 with the membrane marker mCherry-Farnesyl-5 into HEK-P1KO cells. EGFP-hANO1 predominantly exhibited cytosolic fluorescence, with only weak membrane localization, which is consistent with the small calcium-activated chloride currents observed in our electrophysiology experiments. In contrast, EGFP-ANOH-1 failed to localize to the membrane (Supplementary Fig. 8e). Additional experiments using C-terminally tagged EGFP constructs for both hANO1 and nematode ANOH-1 produced similar results to those obtained with the N-terminal tags (Supplementary Fig. 8e). In *C. elegans* ASJ neurons, overexpressed ANOH-1 and hANO1 likely co-localized with the membrane marker mCD8 (Supplementary Fig. 8f, g).

## Discussion

ANO1/TMEM16A is known as a calcium-activated chloride channel, which is typically activated by changes in calcium levels or membrane voltage[41,57,58,65]. Here, we demonstrate that ANOH-1, the *C. elegans* homolog of Anoctamin-1, in conjunction with CIB and ankyrin proteins, forms an essential part of a mechanosensory channel complex, and plays a crucial role in mechanosensation.

Nose touch response and mating behavior are two typical behaviors observed in *C. elegans*[12]. Nose touch response is elicited by mechanical forces[66], whereas successful mating in *C. elegans* requires the integration of multiple sensory cues, including both chemical and physical cues[54,67]. Most male-specific sensory neurons involved in mating behaviors are located in the rays of their tail, such as RnA, RnB, and p.s.c. neurons[32,54]. The only exception is four CEM neurons that are localized in the head of male worms (Fig. 1a)[54]. Previous studies have shown that CEM neurons are required for long-range chemo-attraction toward hermaphrodites through pheromone signaling[33,54]. In this study, we found that CEM neurons are also mechanoreceptor cells, thus establishing the male-specific head neuron CEMs as polymodal sensory neurons capable of both chemosensation and mechanosensation. Notably, the mechanosensitivity would also endow CEM neurons with short-range sensory capability. Indeed, disruption of the amphid CEM neurons significantly affected the nose-touch response of male worms (Fig. 3a).

Mechanoreceptor cells use distinct mechanically activated ion channels to transduce mechanical forces into electrical signals[4,68]. Among known mechanotransduction ion channels in metazoans, nearly all of them are cation-selective channels[3-5,68]. As a result, the opening of these mechanically activated channels depolarizes the cells and is considered an excitatory cellular event. Intriguingly, our results demonstrate that the chloride channel ANOH-1 is required for the mechanosensory response in CEM and ASJ sensory neurons, playing a crucial role in the transduction of mechanical stimulation.

The effect of activating a chloride channel on membrane potential is determined by the chloride gradient across the plasma membrane, represented by the equilibrium potential for Cl⁻ ($E_{Cl}$), and the resting membrane potential ($V_{rest}$). If $V_{rest}$ is below $E_{Cl}$, activation of a chloride channel will depolarize the membrane. Conversely, if $V_{rest}$ is above $E_{Cl}$, activation of a chloride channel will hyperpolarize the membrane. In the mature nervous system, neuronal Cl⁻ concentration is typically low, which keeps $E_{Cl}$ hyperpolarized relative to $V_{rest}$, rendering GABAergic and glycinergic synaptic transmission inhibitory. However, in immature neurons, $E_{Cl}$ is more depolarized than $V_{rest}$, thus GABAergic and glycinergic synaptic transmission is excitatory[46]. Additionally, in certain types of neurons, such as vertebrate olfactory receptor neurons, $Ca^{2+}$-gated Cl⁻ channels in the sensory cilia facilitate a depolarizing efflux of Cl⁻ ions due to the elevated concentration of Cl⁻ in the cilia and the low concentration in the mucosa[69,70]. The activation of the ANOH-1 channel in CEM and ASJ neurons leads to an increase in intracellular calcium level presumably through membrane depolarization and subsequent activation of voltage-gated calcium

channels. Our results show that the opening of ectopically expressed HisCl1 channels in CEM and ASJ neurons increases intracellular calcium, suggesting that $V_{rest}$ is likely below $E_{Cl}$ in these cells, making the chloride channel excitatory (Fig. 2g, h). This is consistent with previous findings, as a prior study documented the robust depolarization of mammalian dorsal root ganglion (DRG) neurons in response to heat ramps, which depends on ANO1 channels[71]. Considering the highly diverse composition and functionality of DRG neurons in various mammals, including humans[72], it would be intriguing to investigate whether ANO1 channels contribute to mechanosensory responses in specific DRG neurons subtypes and other mammalian tissues.

Chloride channels are widely regarded as one of the most diverse channel superfamilies in the animal kingdom[7,47]. Electrophysiological studies have revealed an exceptional variety of chloride conductances among distinct organisms, tissues, and cells. Despite the wide range of biophysically identified chloride channels, it seems that none of them has yet met the criteria to be classified as a direct mechanotransduction channel[1,4,11]. FLYCATCHER1 (FLYC1), a newly discovered chloride-permeable mechanosensitive (MS) ion channel, has been found to play a crucial role in Venus flytrap prey recognition[73,74]. It exhibits a strong preference for chloride over sodium, as evidenced by a $P_{Cl}/P_{Na}$ ratio of 9.8[74]. Interestingly, FLYC1 is a plant ortholog of the prokaryotic mechanosensitive (MS) ion channel MscS, which functions as an osmotic release valve[74]. Interestingly, this ortholog of the prokaryotic MS ion channel is absent from the animal kingdom[73,74].

Our study provides a compelling possibility that ANOH-1 functions as a bona fide mechanotransduction channel. This hypothesis is supported by several key findings:: (i) ANOH-1 is essential for mechanosensation in CEM and ASJ neurons during tactile stimulation; (ii) the absence of ANOH-1 leads to defects in nematode avoidance behavior in response to nose touch; (iii) ANOH-1 is responsible for mediating mechanoreceptor currents (MRCs); (iv) The MRCs dependent on ANOH-1 exhibit rapid activation kinetics and very short latency, similar to typical mechanotransduction channels[11,18,19]; (v) mutations in the channel pore of ANOH-1 blocks mechanosensory responses; (vi) the ectopic expression of ANOH-1 induces mechanosensitivity in neurons that are normally insensitive to mechanical stimulation[4,11,19]. These characteristics align with established criteria for identifying channels involved in mechanotransduction. However, further investigation is required to confirm whether ANOH-1 is directly gated by mechanical forces.

Our findings demonstrate that the defects of mechanosensory function in *anoh-1* mutant worms can be fully rescued by transgenically expressing human ANO1, and ectopic expression of human ANO1 confers mechanosensitivity to primary mechanically insensitive neurons, suggesting that the mechanosensitivity of Anoctamin-1 might be evolutionarily conserved in metazoans. Homologous genes for the two major mechanosensitive ion channels in mammals, Piezo and TMC1, have been identified in *C. elegans*. These genes play critical roles in a range of mechanosensitive processes such as touch sensation, mating, and feeding behaviors in worms[16,21,23], highlighting the evolutionary conservation of mechanotransduction mechanisms from *C. elegans* to humans. Given that ANO1 is widely expressed in mammalian tissues and cells[75], it is logical to speculate that ANO1 may also play a crucial role in the mechanotransduction processes of certain tissues and cells in mammals. On the other hand, considering that most of the identified mechano-gated ion channels are primarily excitatory channels[1,3,4], ANO1 may serve as a component of an inhibitory mechanotransduction complex, depending on $E_{Cl}$ and $V_{rest}$ of the cells. In this case, ANO1 may be co-expressed with some excitatory mechanosensitive cation channels, and play a precise modulatory role in balancing the responses to mechanical stimuli of varying magnitudes.

CIB proteins share sequence homology with KChIP proteins, auxiliary subunits of voltage-gated $K_V4$ channels[5,76]. In association with other proteins like ankyrin, CIB proteins have been implicated in

forming an intracellular tether essential for the mechanical gating of the TMC1 channel[5,21,28,29,61,77,78]. This interaction is observed in various organs, including the mouse cochlea and vestibular organs, the zebrafish inner ear and lateral line system, and the *C. elegans* mechanosensitive OLQ neurons[17,21,26–28]. Our results indicate that CIB proteins, possibly in association with ankyrin, are necessary for the mechanosensory function of ANOH-1/ANO1 (Fig. 7). Interestingly, members of the OSCA/TMEM63 family are predicted to exhibit structural similarities with TMEM16 proteins and TMC1[3,62,79]. While our study and others suggest that tether-mediated activation may account for the mechanical gating of TMC1, OSCA proteins are believed to be intrinsic mechanosensitive monomeric channels that can be directly activated by alterations in membrane tension[3,21,80,81]. Further structure-function analysis is invaluable for enhancing our understanding of the gating mechanisms underlying these channels.

In brief, this study reveals that the chloride channel ANOH-1, associated with CIB and ankyrin proteins, plays a crucial role in mechanosensation (Fig. 7g). This discovery offers insights into the intricate mechanisms governing the functionality of anion channels within the structural context of living cells, as well as their contributions to mechanotransduction.

## Methods

### *C. elegans* strains and media

The strains were cultured under standard conditions at a temperature of 20 °C, on nematode growth medium (NGM) plates that had been seeded with the OP50 strain of *E. coli*[30]. *C. elegans* strains used in this study are listed below.

| Strain name | Source | Identifier |
| --- | --- | --- |
| *N2* | CGC | *ST348* |
| *him-5(e1490)V;* | CGC | *ST6137* |
| *anoh-1(tm4762)III;him-5(e1490)V;* | This study | ST6136 |
| *osm-9(ky10)IV;him-5(e1490)V;* | This study | ST6138 |
| *him-5(e1490)V;KanEx45[Ppkd-2::mCherry+Ppkd-2::GCaMP5.0+Podr-1::DsRed]* | This study | ST703 |
| *him-5(e1490)V;KanIs5[Ppkd-2::mCherry+Ppkd-2::GCaMP5.0+Podr-1::DsRed]* | This study | ST1094 |
| *him-5(e1490)V;KanEx519[Ppkd-2::mCherry+Ppkd-2::GCaMP5.0+Ppkd-2::HisCl1+Podr-1::DsRed]* | This study | ST1849 |
| *him-5(e1490)V;KanEx2007[Ppkd-2::GCaMP6s+Ppkd-2::mCherry+Punc-122::mCherry]* | This study | ST6064 |
| *abts-1(ok1566)I;him-5(e1490)V;KanIs5[Ppkd-2::mCherry+Ppkd-2::GCaMP5.0+Podr-1::DsRed]* | This study | ST1797 |
| *abts-4(ok953)X;him-5(e1490)V;KanIs5[Ppkd-2::mCherry+Ppkd-2::GCaMP5.0+Podr-1::DsRed]* | This study | ST1798 |
| *abts-4(ok954)X;him-5(e1490)V;KanIs5[Ppkd-2::mCherry+Ppkd-2::GCaMP5.0+Podr-1::DsRed]* | This study | ST1799 |
| *abts-2(gk951956)X;him-5(e1490)V;KanIs5[Ppkd-2::mCherry+Ppkd-2::GCaMP5.0+Podr-1::DsRed]* | This study | ST1814 |
| *abts-3(ok368)II;him-5(e1490)V;KanIs5[Ppkd-2::mCherry+Ppkd-2::GCaMP5.0+Podr-1::DsRed]* | This study | ST1815 |
| *kcc-2(vs132)IV;him-5(e1490)V;KanIs5[Ppkd-2::mCherry+Ppkd-2::GCaMP5.0+Podr-1::DsRed]* | This study | ST1812 |
| *kcc-3(ok228)II;him-5(e1490)V;KanIs5[Ppkd-2::mCherry+Ppkd-2::GCaMP5.0+Podr-1::DsRed]* | This study | ST1813 |

| | | |
|---|---|---|
| *anoh-1(tm4762)III;him-5(e1490) V;KanEx45[Ppkd-2::mCherry+Ppkd-2::GCaMP5.0+Podr-1::Dsred]* | This study | ST1900 |
| *anoh-2(tm4796)IV;him-5(e1490) V;KanEx45[Ppkd-2::mCherry+Ppkd-2::GCaMP5.0+Podr-1::Dsred]* | This study | ST1895 |
| *clh-1(qa900)II;him-5(e1490)V;KanIs5[Ppkd-2::mCherry+Ppkd-2::GCaMP5.0+Podr-1::DsRed]* | This study | ST1612 |
| *clh-2(ok636)II;him-5(e1490)V;KanIs5[Ppkd-2::mCherry+Ppkd-2::GCaMP5.0+Podr-1::DsRed]* | This study | ST1648 |
| *clh-3(ok763)II;him-5(e1490)V;KanIs5[Ppkd-2::mCherry+Ppkd-2::GCaMP5.0+Podr-1::DsRed]* | This study | ST1649 |
| *clh-4(ok1162)X;him-5(e1490)V;KanIs5[Ppkd-2::mCherry+Ppkd-2::GCaMP5.0+Podr-1::DsRed]* | This study | ST1660 |
| *him-5(e1490)V;KanIs5[Ppkd-2::mCherry+Ppkd-2::GCaMP5.0+Podr-1::DsRed];KaEx577[Ppkd-2::clh-5 RNAi+Punc-122::GFP]* | This study | ST2027 |
| *clh-6(ok791)V;KanIs5[Ppkd-2::mCherry+Ppkd-2::GCaMP5.0+Podr-1::DsRed]* | This study | ST1667 |
| *best-1(tm6816)IV;him-5(e1490)V;KanEx45[Ppkd-2::mCherry+Ppkd-2::GCaMP5.0+Podr-1::DsRed]* | This study | ST1896 |
| *best-2(tm6751)IV;him-5(e1490)V;KanEx45[Ppkd-2::mCherry+Ppkd-2::GCaMP5.0+Podr-1::DsRed]* | This study | ST1898 |
| *best-3(tm6768)II;him-5(e1490)V;KanEx45[Ppkd-2::mCherry+Ppkd-2::GCaMP5.0+Podr-1::DsRed]* | This study | ST1932 |
| *best-4(tm7120)II;him-5(e1490) V;KanEx2007[Ppkd-2::mCherry+Ppkd-2::GCaMP6s+Punc-122::mCherry]* | This study | ST6020 |
| *best-7(gk963665)IV;him-5(e1490) V;KanEx45[Ppkd-2::mCherry+Ppkd-2::GCaMP5.0+Podr-1::DsRed]* | This study | ST1968 |
| *best-8(gk128576)I;him-5(e1490)V;KanEx45[Ppkd-2::mCherry+Ppkd-2::GCaMP5.0+Podr-1::DsRed]* | This study | ST1905 |
| *best-9(tm7876)IV;him-5(e1490)V;KanIs5[Ppkd-2::mCherry+Ppkd-2::GCaMP5.0+Podr-1::DsRed]* | This study | ST2657 |
| *best-10(ok2691)I;him-5(e1490) V;KanEx2007[Ppkd-2::mCherry+Ppkd-2::GCaMP6s+Punc-122::mCherry]* | This study | ST6063 |
| *best-11(tm7837)I;him-5(e1490)V;KanEx45[Ppkd-2::mCherry+Ppkd-2::GCaMP5.0+Podr-1::DsRed]* | This study | ST1899 |
| *best-12(gk257154);KanEx45[Ppkd-2::mCherry + Ppkd-2::GCaMP5.0+Podr-1::Dsred]* | This study | ST1969 |
| *best-13(gk702522);him-5(e1490) V;KanEx45[Ppkd-2::mCherry+Ppkd-2::GCaMP5.0+Podr-1::DsRed]* | This study | ST1903 |
| *best-13(gk452273);him-5(e1490) V;KanEx45[Ppkd-2::mCherry+Ppkd-2::GCaMP5.0+Podr-1::DsRed]* | This study | ST1935 |
| *best-14(gk681146);KanIs5[Ppkd-2::mCherry +Ppkd-2::GCaMP5.0+Podr-1::DsRed]* | This study | ST2664 |
| *best-15(gk443656);him-5(e1490) V;KanEx45[Ppkd-2::mCherry+Ppkd-2::GCaMP5.0+Podr-1::DsRed]* | This study | ST1934 |
| | This study | ST1907 |
| *best-16(gk205018);him-5(e1490) V;KanEx45[Ppkd-2::mCherry+Ppkd-2::GCaMP5.0+Podr-1::DsRed]* | | |
| *best-17(gk960972);him-5(e1490) V;KanEx45[Ppkd-2::mCherry+Ppkd-2::GCaMP5.0+Podr-1::DsRed]* | This study | ST1906 |
| *best-19(gk478492)IV;him-5(e1490) V;KanEx2007[Ppkd-2::mCherry+Ppkd-2::GCaMP6s+Punc-122::mCherry]* | This study | ST6043 |
| *best-20(gk902100);him-5(e1490) V;KanEx45[Ppkd-2::mCherry+Ppkd-2::GCaMP5.0+Podr-1::DsRed]* | This study | ST1897 |
| *best-20(gk705349);him-5(e1490) V;KanEx45[Ppkd-2::mCherry+Ppkd-2::GCaMP5.0+Podr-1::DsRed]* | This study | ST1904 |
| *best-21(gk219765);him-5(e1490) V;KanEx45[Ppkd-2::mCherry+Ppkd-2::GCaMP5.0+Podr-1::DsRed]* | This study | ST1933 |
| *best-24(gk314665);him-5(e1490) V;KanEx45[Ppkd-2::mCherry+Ppkd-2::GCaMP5.0+Podr-1::DsRed]* | This study | ST1901 |
| *acc-1(tm3268)IV;him-5(e1490) V;KanEx2007[Ppkd-2::mCherry+Ppkd-2::GCaMP6s+Punc-122::mCherry]* | This study | ST6062 |
| *acc-2(ok2216)IV;him-5(e1490) V;KanEx2007[Ppkd-2::mCherry+Ppkd-2::GCaMP6s+Punc-122::mCherry]* | This study | ST6031 |
| *acc-3(ok3450)X;him-5(e1490) V;KanEx2007[Ppkd-2::mCherry+Ppkd-2::GCaMP6s+Punc-122::mCherry]* | This study | ST6033 |
| *acc-4(ok2371)III;him-5(e1490) V;KanEx2007[Ppkd-2::mCherry+Ppkd-2::GCaMP6s+Punc-122::mCherry]* | This study | ST6042 |
| *gab-1(tm3577)III;him-5(e1490) V;KanEx2007[Ppkd-2::mCherry+Ppkd-2::GCaMP6s+Punc-122::mCherry]* | This study | ST6048 |
| *lgc-57(tm6214)II;him-5(e1490) V;KanEx2007[Ppkd-2::mCherry+Ppkd-2::GCaMP6s+Punc-122::mCherry]* | This study | ST6021 |
| *lgc-56(tm4317)II;him-5(e1490) V;KanEx2007[Ppkd-2::mCherry+Ppkd-2::GCaMP6s+Punc-122::mCherry]* | This study | ST6022 |
| *glc-2(gk179)I;him-5(e1490)V;KanEx2007[Ppkd-2::mCherry+Ppkd-2::GCaMP6s+Punc-122::mCherry]* | This study | ST6017 |
| *glc-2(ok1047)I;him-5(e1490)V;KanEx2007[Ppkd-2::mCherry+Ppkd-2::GCaMP6s+Punc-122::mCherry]* | This study | ST6018 |
| *glc-3(ok321)V;KanEx2007[Ppkd-2::mCherry +Ppkd-2::GCaMP6s+Punc-122::mCherry]* | This study | ST6019 |
| *glc-4(ok212)II;him-5(e1490)V;KanEx2007[Ppkd-2::mCherry+Ppkd-2::GCaMP6s+Punc-122::mCherry]* | This study | ST6055 |
| *lgc-20(gk353686);him-5(e1490) V;KanEx2007[Ppkd-2::mCherry+Ppkd-2::GCaMP6s+Punc-122::mCherry]* | This study | ST6044 |

| Strain | Source | ID |
|---|---|---|
| lgc-23(tm6258)X;him-5(e1490)V;KanEx2007[Ppkd-2::mCherry+Ppkd-2::GCaMP6s+Punc-122::mCherry] | This study | ST6023 |
| lgc-24(gk295568)X;him-5(e1490)V;KanEx2007[Ppkd-2::mCherry+Ppkd-2::GCaMP6s+Punc-122::mCherry] | This study | ST6049 |
| lgc-34(gk532)II;him-5(e1490)V;KanEx2007[Ppkd-2::mCherry+Ppkd-2::GCaMP6s+Punc-122::mCherry] | This study | ST6047 |
| lgc-36(gk247083)V;KanEx2007[Ppkd-2::mCherry+Ppkd-2::GCaMP6s+Punc-122::mCherry] | This study | ST6045 |
| lgc-36(tm2749)V;KanEx2007[Ppkd-2::mCherry+Ppkd-2::GCaMP6s+Punc-122::mCherry] | This study | ST6061 |
| lgc-37(tm6573)III;him-5(e1490)V;KanEx2007[Ppkd-2::mCherry+Ppkd-2::GCaMP6s+Punc-122::mCherry] | This study | ST6032 |
| lgc-38(kan20)III;him-5(e1490)V;KanEx2007[Ppkd-2::mCherry+Ppkd-2::GCaMP6s+Punc-122::mCherry] | This study | ST6052 |
| lgc-39(gk633958)V;KanEx2007[Ppkd-2::mCherry + Ppkd-2::GCaMP6s+Punc-122::mCherry] | This study | ST6037 |
| lgc-40(tm3377)X;him-5(e1490)V;KanEx2007[Ppkd-2::mCherry+Ppkd-2::GCaMP6s+Punc-122::mCherry] | This study | ST6024 |
| lgc-40(n4545)X;him-5(e1490)V;KanEx2007[Ppkd-2::mCherry+Ppkd-2::GCaMP6s+Punc-122::mCherry] | This study | ST6051 |
| lgc-41(tm8844)X;him-5(e1490)V;KanEx2007[Ppkd-2::mCherry+Ppkd-2::GCaMP6s+Punc-122::mCherry] | This study | ST6025 |
| lgc-42(gk578228)III;him-5(e1490)V;KanEx2007[Ppkd-2::mCherry+Ppkd-2::GCaMP6s+Punc-122::mCherry] | This study | ST6038 |
| lgc-43(tm10110)IV;him-5(e1490)V;KanEx2007[Ppkd-2::mCherry+Ppkd-2::GCaMP6s+Punc-122::mCherry] | This study | ST6057 |
| lgc-44(gk861777)V;KanEx2007[Ppkd-2::mCherry+Ppkd-2::GCaMP6s+Punc-122::mCherry] | This study | ST6039 |
| lgc-46(gk166091)III;him-5(e1490)V;KanEx2007[Ppkd-2::mCherry+Ppkd-2::GCaMP6s+Punc-122::mCherry] | This study | ST6040 |
| lgc-46(ok2949)III;him-5(e1490)V;KanEx2007[Ppkd-2::mCherry+Ppkd-2::GCaMP6s+Punc-122::mCherry] | This study | ST6058 |
| lgc-47(ok2963)X;him-5(e1490)V;KanEx2007[Ppkd-2::mCherry+Ppkd-2::GCaMP6s+Punc-122::mCherry] | This study | ST6056 |
| lgc-49(tm6556)V;KanEx2007[Ppkd-2::mCherry+Ppkd-2::GCaMP6s+Punc-122::mCherry] | This study | ST6026 |
| lgc-49(gk246966)V;KanEx2007[Ppkd-2::mCherry+Ppkd-2::GCaMP6s+Punc-122::mCherry] | This study | ST6050 |
| lgc-50(tm3712)III;him-5(e1490)V;KanEx2007[Ppkd-2::mCherry+Ppkd-2::GCaMP6s+Punc-122::mCherry] | This study | ST6027 |
| lgc-50(gk707666)III;him-5(e1490)V;KanEx2007[Ppkd-2::mCherry+Ppkd-2::GCaMP6s+Punc-122::mCherry] | This study | ST6046 |
| lgc-51(tm4318)I;him-5(e1490)V;KanEx2007[Ppkd-2::mCherry+Ppkd-2::GCaMP6s+Punc-122::mCherry] | This study | ST6030 |
| lgc-52(tm4258)IV;him-5(e1490)V;KanEx2007[Ppkd-2::mCherry+Ppkd-2::GCaMP6s+Punc-122::mCherry] | This study | ST6028 |
| lgc-53(tm2735)X;him-5(e1490)V;KanEx2007[Ppkd-2::mCherry+Ppkd-2::GCaMP6s+Punc-122::mCherry] | This study | ST6029 |
| lgc-53(n4330)X;him-5(e1490)V;KanEx2007[Ppkd-2::mCherry+Ppkd-2::GCaMP6s+Punc-122::mCherry] | This study | ST6059 |
| lgc-54(tm3448)V;KanEx2007[Ppkd-2::mCherry+Ppkd-2::GCaMP6s+Punc-122::mCherry] | This study | ST6054 |
| mca-3(ar492)IV;him-5(e1490)V;KanEx2007[Ppkd-2::mCherry+Ppkd-2::GCaMP6s+Punc-122::mCherry] | This study | ST6035 |
| ttyh-1(tm2415)X;him-5(e1490)V;KanEx2007[Ppkd-2::mCherry+Ppkd-2::GCaMP6s+Punc-122::mCherry] | This study | ST6041 |
| unc-2(ra612)X;him-5(e1490)V;KanEx2007[Ppkd-2::mCherry+Ppkd-2::GCaMP6s+Punc-122::mCherry] | This study | ST6036 |
| unc-49(e382);him-5(e1490)V;KanEx2007[Ppkd-2::mCherry+Ppkd-2::GCaMP6s+Punc-122::mCherry] | This study | ST6034 |
| anoh-1;him-5;KanEx2010[Ppkd-2::anoh-1::sl2::mcherry+Ppkd-2::GCaMP6s+Punc-122::mCherry] | This study | ST6073 |
| anoh-1;him-5;KanEx2014[Ppkd-2::hANO1::sl2::mcherry+Ppkd-2::GCaMP6s+Punc-122::mCherry] | This study | ST6077 |
| anoh-1;him-5;KanEx992[Ppkd-2::anoh-1(K588A)::sl2::mCherry+Pkd-2::GCaMP6s+Punc-122::GFP] | This study | ST3259 |
| him-5(e1490);KanEx520[Ppkd-2::mCherry+Ppkd-2::pH-miniSOG+Podr-1::DsRed] | This study | ST1850 |
| him-5(e1490);KanEx540[Psra-6::pH-miniSOG+Ppkd-2::pH-miniSOG+lin-44::GFP] | This study | ST1966 |
| him-5(e1490);KanEx579[Psra-6::pH-miniSOG+Plin-44::GFP] | This study | ST2029 |
| him-5(e1490)V;KanEx2044[Panoh-1b::GFP+Ppkd-2::mCherry] | This study | ST6115 |
| him-5(e1490)V;KanEx2051[Ptrx-1::mCherry+Ptrx-1::GCaMP6s+Punc-122::GFP] | This study | ST6122 |
| anoh-1(tm4762)III;him-5(e1490)V;KanEx2052[Ptrx-1::mCherry+Ptrx-1::GCaMP6s+Punc-122::GFP] | This study | ST6125 |
| anoh-1(tm4762)III;him-5(e1490)V;KanEx2053[Ptrx-1::anoh-1::sl2::mCherry+Ptrx-1::GCaMP6s+Punc-122::mCherry] | This study | ST6126 |
| anoh-1(tm4762)III;him-5(e1490)V;KanEx2055[Ptrx-1::hANO1::sl2::mCherry+Ptrx-1::GCaMP6s+Punc-122::mCherry] | This study | ST6128 |

| Strain | Source | ID |
|---|---|---|
| anoh-1(tm4762)III;him-5(e1490)V;KanEx2057[Ptrx-1::hANO6::sl2::mCherry+Ptrx-1::GCaMP6s+Punc::122-GFP] | This study | ST6130 |
| N2;KanEx2060[Ptrx-1::HisCl1::sl2::mCherry+Ptrx-1::GCaMP6s+Punc::122-GFP] | This study | ST6133 |
| anoh-1(tm4762)III;him-5(e1490)V;KanEx1010[Ptrx-1::anoh-1(K588A)::sl2::mCherry+Plin44::GFP] | This study | ST3298 |
| N2;KanEx[Psra-9::mCherry+Psra-9::GCaMP6s+Plin-44::GFP] | This study | ST2845 |
| N2;KanEx2028[Psra-9::anoh-1::sl2::mCherry+Psra-9::GCaMP6s+Punc-122::GFP] | This study | ST6099 |
| N2;KanEx2029[Psra-9::hANO1::sl2::mCherry+Psra-9::GCaMP6s+Punc-122::GFP] | This study | ST6100 |
| him-5(e1490)V;KanEx2033[Psra-9::anoh-1(K588A)::sl2::mCherry+Psra-9::GCaMP6s+Punc-122::GFP] | This study | ST6104 |
| N2;KanEx2032[Psra-9::anoh-1(K588E)::sl2::mCherry+Psra-9::GCaMP6s+Punc-122::GFP] | This study | ST6103 |
| N2;KanEx2030[Psra-9::anoh-1(R607A-R607A)::sl2::mCherry+Psra-9::GCaMP6s+Punc-122::GFP] | This study | ST6101 |
| him-5(e1490)V;KanEx2035[Psra-9::anoh-1(R614E)::sl2::mCherry+Psra-9::GCaMP6s+Punc-122::GFP] | This study | ST6106 |
| unc-44(e362)IV;him-5(e1490)V;KanEx2051[Ptrx-1::mCherry+Ptrx-1::GCaMP6s+Punc-122::GFP] | This study | ST3142 |
| calm-1(tm1353)I;him-5(e1490)V;KanEx2051[Ptrx-1::mCherry+Ptrx-1::GCaMP6s+Punc-122::GFP] | This study | ST3124 |
| calm-1(tm1353)I;him-5(e1490)V;KanEx979[Ptrx-1::calm-1::sl2::mCherry+Ptrx-1::GCaMP6s+Punc-122::GFP] | This study | ST3241 |
| calm-1(tm1353)I;him-5(e1490)V;KanEx1007[Ptrx-1::hCIB2::sl2::mCherry+Plin44::GFP]line1 | This study | ST3295 |
| calm-1(tm1353)I;him-5(e1490)V;;KanEx1006[Ptrx-1::hCIB3::sl2::mCherry+Punc-122::GFP] | This study | ST3288 |
| calm-1(tm1353)I;him-5(e1490)V;KanEx1003[Psra-9::anoh-1::sl2::mCherry+Psra-9::GCaMP6s+Punc-122::GFP] | This study | ST3280 |
| calm-1(tm1353)I;him-5(e1490)V;KanEx994[Psra-9::calm-1::sl2::mCherry+Psra-9::anoh-1::sl2::mCherry+Psra-9::GCaMP6s+Punc-122::GFP] | This study | ST3261 |
| calm-1(tm1353)I;him-5(e1490)V;KanEx1009[Psra-9::hANO1::sl2::mCherry+Psra-9::GCaMP6s+Plin44::GFP] | This study | ST3297 |
| calm-1(tm1353)I;him-5(e1490)V;KanEx989[Psra-9::hCIB2::sl2::mCherry+Psra-9::hANO1::sl2::mCherry+Punc-122::GFP] | This study | ST3252 |
| calm-1(tm1353)I;him-5(e1490)V;KanEx980[Psra-9::hCIB3::sl2::mCherry+Psra-9::hANO1::sl2::mCherry+Psra-9::GCaMP6s+Punc-122::GFP] | This study | ST3242 |
| calm-1(tm1353)I;him-5(e1490)V;KanEx991[Ppkd-2::GCaMP6s+Ppkd-2::mCherry+Punc-122::GFP] | This study | ST3292 |
|  |  | ST3290 |
| him-5(e1490)V;KanEx991[Ppkd-2::GCaMP6s+Ppkd-2::mCherry+Punc-122::GFP] | This study |  |
| calm-1(tm1353)I;him-5(e1490)V;KanEx1016[Ppkd-2::calm-1::sl2::mCherry+Ppkd-2::GCaMP6s+Plin-44::GFP] | This study | ST3308 |
| N2;KanIs8[Psra-6::mCherry+Psra-6::GCaMP5.0+Plin-44::GFP] | This study | ST3092 |
| anoh-1(tm4762)III;KanIs8[Psra-6::mCherry+Psra-6::GCaMP5.0+Plin-44::GFP] | This study | ST3090 |
| deg-1(u38)X;anoh-1(tm4762)III;KanIs8[Psra-6::mCherry-+Psra-6::GCaMP5.0+Plin-44::GFP] | This study | ST6092 |
| him-5(e1490)V;KanEx2045[Panoh-1b::mCherry+Ptrx-1::GFP] | This study | ST6116 |
| N2;KanEx2023[Pgpa-4::anoh-1::sl2::mCherry+Pgpa-4::GCaMP5.0+Punc-122::mCherry] | This study | ST6088 |
| N2;KanEx2025[Pgpa-4::hANO1::sl2::mCherry+Pgpa-4::GCaMP5.0+Plin44::GFP] | This study | ST6090 |
| N2;kanEx796[Pgpa-4::GCaMP5.0+Plin-44::GFP] | This study | ST2584 |
| abts-3(ok268)II;him-5(e1490)V;KanEx519[Ppkd-2::HisCl1::sl2::mCherry+Ppkd-2::GCaMP5.0+-Podr-1::DeRed] | This study | ST3580 |
| him-5(e1490)V;KanEx519[Ppkd-2::HisCl1::sl2::mCherry+Ppkd-2::GCaMP5.0+Podr-1::DeRed] | This study | ST3581 |
| anoh-1(tm4762)III;him-5(e1490)V;KanEx1120[Ppkd-2::ANO1(-K645A)::sl2::mCherry+Ppkd-2::GCaMP6s+Punc-122::GFP] | This study | ST3591 |
| anoh-1(tm4762)III;him-5(e1490)V;KanEx1121[Ptrx-1::ANO1(K645A)::sl2::mCherry+Ppkd-2::GCaMP6s+Punc-122::mCherry] | This study | ST3600 |
| him-5(e1490)V;Kan1123[Ppkd-2::mCherry+Punc-122::mCherry] | This study | ST3602 |
| kcc-3(ok228)II;him-5(e1490)V;KanEx519[Ppkd-2::HisCl1::sl2::mCherry+Ppkd-2::GCaMP5.0+-Podr-1::DeRed] | This study | ST3603 |
| KanIs58[calm-1::mNeurogreen];KanEx1124[Ptrx-1::mCherry+Punc-122::mCherry] | This study | ST3604 |
| KanIs58[calm-1::mNeuro-green];KanEx1128[Ppkd-2::mCherry+Punc-122::mCherry] | This study | ST3608 |
| KanIs58[calm-1::mNeurogreen];KanEx1129[Psra-6::mCherry+Punc-122::mCherry] | This study | ST3609 |
| KanIs57[anoh-1::mNeuro-green];KanEx1123[Ppkd-2::mCherry+Punc-122::mCherry] | This study | ST3615 |
| him-5(e1490)V;kanEx796[Ptrx-1::anoh-1::GFP+Ptrx-1::mCD8::mCherry+Punc-122::mCherry] | This study | ST3281 |
| N2;kanEx796[Ptrx-1::ANO1::GFP+Ptrx-1::mCD8::mCherry+Punc-122::mCherry] | This study | ST3692 |

## Electrophysiology

Using a previously described protocol, whole-cell recordings were conducted on an upright Olympus microscope (model BX51WI) equipped with an EPC-10 amplifier and controlled by the Patchmaster software developed by HEKA[19,53]. A glass stimulus probe was used

together with a Piezo actuator (PI) mounted on a micromanipulator and triggered by the EPC-10 amplifier to deliver mechanical stimulation. The worms were placed in a drop of bath solution on a sylgard-coated coverglass and affixed to the coverglass with medical-grade cyanoacrylate-based glue (Gluture Topical Tissue Adhesive, Abbott Laboratories). Subsequently, a small cuticle piece located in the worm's head region was excised and pinned down to the coverglass, thereby enabling the exposure of the cell body of ASJ, ASH, ASK, or ASI neurons. The bath solution contained (in mM): 145 NaCl, 2.5 KCl, 1 $MgCl_2$, 5 $CaCl_2$, 10 HEPES, and 20 glucose (325−335 mOsm, pH adjusted to 7.3 with NaOH). The pipette solution contained (in mM): 145 CsCl, 2.5 KCl, 5 $MgCl_2$, 0.25 $CaCl_2$, 10 HEPES, 10 glucose and 5 mM EGTA (325−335 mOsm, pH adjusted to 7.2 with CsOH).

In the "$Na^+$-free" bath solution, NaCl was substituted with NMDG-Cl. Similarly, in the "$Cl^-$-free" bath solution, NaCl was replaced with Na-gluconate. In the "$Cl^-$-free" pipette solution, Cs-gluconate was used to replace CsCl. In the "$Ca^{2+}$-free" bath solution, $CaCl_2$ was absent. In the "$Ca^{2+}$-free" pipette solution, 1 mM BAPTA was added to chelate $Ca^{2+}$ rapidly. To block $K^+$ currents and minimize noise during whole-cell recordings, CsCl was used to replace KCl in the pipette solution unless otherwise specified.

For patch-clamp recordings of HEK293T cells, the pipette resistance is about 3−5 MΩ, the seal resistance is greater than 1 GΩ, and the series resistance is around 10−15 MΩ.

Given the small size of worm neurons, typically with cell body diameters of 3−5 μm, for recording worm neurons, the recording pipette tips were usually about 1 μm in diameter, and the pipette resistance was approximately 15−20 MΩ. The seal resistance exceeded 1 GΩ, and the series resistance (Rs) was around 50−70 MΩ. Additionally, the pipettes were not fire-polished. Since the amplitude of the MRCs is small, approximately 20 pA when the holding potential is −70 mV, we did not compensated the Rs. The voltage error due to uncompensated series resistance is less than 2 mV, on average, and was not corrected.

Unless stated otherwise, membrane voltages were clamped at −70 mV, and the current acquisition was sampled at 20−40 kHz. Voltage offsets were zeroed using the Patchmaster software prior to obtaining tight seals. Liquid junction potentials (LJPs) were calculated based on the combinations of bath and pipette solutions using an LJP calculator (https://swharden.com/LJPcalc/) and were subsequently applied to posthoc corrections of membrane potential values.

## Calcium imaging
A worm was picked and moved to a droplet of bath solution placed on a coverslip. Subsequently, the worm was secured onto the coverslip using cyanoacrylate-based glue, following this the intracellular calcium transients were detected using GCaMP5.0 or GCaMP6s signal[44,45,53]. To acquire calcium imaging, an Olympus microscope (model BX51WI) equipped with a 60x objective lens was coupled with an Andor DL-604M EMCCD camera. The imaging data were collected using MacroManager software. GCaMP proteins were excited using a Lambda XL light source and the fluorescence signals were obtained at a rate of 1 Hz. For each data point, the average GCaMP signal from the initial ten seconds before stimulation was regarded as $F_0$, and $\Delta F/F_0$ was calculated. We recorded and calculated the calcium response or MRCs from a single CEM neuron in each worm.

## Mechanical stimulation
The touch stimulators were made from borosilicate glass capillaries with an outer diameter of 1.0 mm (WPI, 1B100F-4). These capillaries were heat-pulled using a Sutter electrode puller (P-97). The capillary was driven by a piezoelectric actuator mounted on a micromanipulator manufactured by Sutter. To deliver mechanical stimulation to the nose tip of worms, the tip diameter of the touch stimulator was approximately 10 μm. The capillary was positioned perpendicularly to the nose tip of the worm, and during the "on" phase, it was moved towards the worm to press into the side of the nose and hold for 100 milliseconds. During the "off" phase, the capillary returned to its original position as previously described[19].

The latency time of MRCs was measured between stimulus delivery and opening of the channel (onset of the MRCs). This duration includes the time required to move the probe, the transmission of force from the cuticle to the channels, and the time needed to activate the channels.

For MRCs recording in cultured single ASJ cells, the tips of the capillaries are elongated and have very small openings, around 0.2 μm, which is approximately 1/20 of the diameter of the ASJ cell body. We set the touch pipette to move forward 4 μm to stimulate the cell body; however, the actual displacement might be much shorter due to the bending of the slender touch pipette upon contact with the cell. We carefully controlled the touch probe's heat-pulling procedure and regularly inspected both the tip length and opening diameter, as well as the position of the probe relative to the cells, and replaced the touch pipette for each cell recording to ensure optimal flexibility. A larger tip or reduced flexibility could push the cell out of position, disrupting the seal or even puncturing the cell membrane.

## Embryonic primary cell culture
To prepare the primary cell culture of *C. elegans*, a method previously described was followed[44,82]. Gravid worms were raised on 8× peptone plates with *E. coli* OP50 as their food source, and eggs were isolated and enriched from them. Embryonic cells were dissociated and plated at a low cell density on a cover glass coated with peanut lectin. The L-15 cell culture medium was changed daily, and the isolated mature ASJ cells were identified by their expression of *Ptrx-1::mCherry* after three days of culture.

## Cell culture and transfection
HEK293T cells (ATCC CRL-3216) and HEK-P1KO cells (ATCC CRL-3519) were cultivated in DMEM medium supplemented with 10% fetal bovine serum within a 37 °C incubator containing 5% $CO_2$. Cells were transferred into dishes one day before transfection. *C. elegans* cDNA *anoh-1*, *calm-1*, *unc-44* and human cDNA *ANO1* and *CIB2* were cloned into the mammalian expression vector pEGFP-C3. For Co-IP experiments, PCDNA5 verctor backbone was used. Transfection of the cells was achieved using Lipofectamine 3000 (ThermoFisher) for four hours. Cells were put to use at 24 h post-transfection.

## Co-immunoprecipitation (Co-IP)
Using a protocol previously described[17], transfected HEK293T cells were washed with cold PBS and lysed in TAP buffer (20 mM Tris-HCl, pH 7.5, 150 mM NaCl, 0.5% NP-40, 1 mM NaF, 1 mM $Na_3VO_4$, 1 mM EDTA, 10 nM MG132, Protease cocktail, Phosphatase cocktail) on ice for 30 min. The cell lysates were centrifuged at 14,000 rpm for 10 min. The supernatant was split into two aliquots, one for immunoblot analysis and the other for Co-IP. The supernatant was incubated with anti-HA agarose affinity gel and rotated for 4 h at 4 °C. After incubation, the beads were washed 3 times with TAP buffer. The immuno-precipitated protein complexes were eluted using an SDS loading buffer for 30 min at 37 °C. The samples were separated with SDS-PAGE and transferred to PVDF membranes and probed with the corresponding antibody. For each representative figure that was shown, at least three different experiments were performed. Anti-β-Tubulin (M1305-2; Huaan Hangzhou, 1:5,000), Anti-HA nanobody agarose beads: (KTSM1305, AlpaLifeBio), Anti-Flag beads: (KTSM1308, AlpaLifeBio), Anti-HA-Tag-HRP-direct (M180-7, MBL, 1:5,000), Anti-Flag-Tag-HRP-direct (M185-7, MBL, 1:5,000) were used in this study.

## Confocal microscopy
To prepare an agar-coated glass slide, 2−3% agarose was dropped onto a glass slide, which was then covered with another glass slide. After the

agarose had solidified in a few minutes, the slides were separated and 10 µL of 200 mM sodium azide (NaN₃) was added to the agar patch[83]. A single worm was transferred to the agar, and a coverslip was placed on top. The glass slide was then placed under an inverted confocal laser microscope to capture images of the worms using FV10-ASW Viewer (Olympus). An intensity projection was generated using Image J by selecting a region of interest (ROI) and subtracting the average intensity value of a background area of the same size.

## Behavioral assays

**Nose touch assay.** To prepare the assay plates, a droplet of overnight culture of *E. coli* strain OP50 was spread on NGM plates and used within 4 h. Ten worms per strain were then transferred onto the assay plate. An eyelash was placed on the surface of the plate in front of the moving worms. Whenever the nose tip of a worm touched the eyelash perpendicularly, we recorded whether a reversal response or null response occurred. Each worm was tested three times with at least a one-minute interval between each nose touch stimulation. The nose touch-insensitive mutant, *osm-9(ky10)*, was used as a control[18]. The experiment was repeated for at least five different days.

**Pharyngeal pumping assay.** Day 1 worms were raised on OP50-seeded plates and adapted at room temperature (20 °C) for 10 min. We then recorded pharyngeal pumping for 10 s by using a microscope with a camera and counted the pumping rates.

**Male mating efficiency assay.** Two L4-stage males and two L4-stage N2 hermaphrodites were transferred to a 35-mm NGM plate containing a 5-mm-diameter lawn of OP50 bacteria. After 24 h, the males were removed from the plate, followed by the removal of the hermaphrodites 48 hours later. The number of males and hermaphrodites in the offspring was then counted, and the proportion of males in the offspring was calculated.

**Egg retention assay.** Individual Day 2 worm was dissolved in a droplet of 50% NaClO solution on a plate. We then counted the retained eggs after the worm body split.

**Locomotion assay.** Day 1 worms were transferred to OP50-seeded plates and allowed to adapt for 10 min. Worms' spontaneous movement was tracked for 1 min. Locomotion speed was quantified by analyzing each recording using the Wormlab software.

**Swimming assay.** Day 1 adult worms were transferred to a droplet of M9 buffer. The swimming behavior of the worms was recorded for 1 minute. The swimming frequency was quantified by analyzing each recording using Wormlab software.

## Molecular biology

The promoters, which label specific neurons, were amplified by PCR from N2 genomic DNA and then combined with specific donor vector fragments using the In-Fusion PCR Cloning Kit (TaKaRa, Inc). The following promoters were used for specific neurons: *pkd-2* (2 kb) for CEM neurons, *sra-6* (4 kb) for ASH neurons, *gpa-4* (2.6 kb) for ASI neurons, *trx-1* (1 kb) for ASJ neurons, and *sra-9* (3 kb) for ASK neurons. The promoter located 914 bp upstream of the start codon of *anoh-1* and the promoter located 3 kb upstream of the start codon of *calm-1* were used to determine the expression pattern.

## Immunostaining

Worm staining was performed following the previously described method[53]. In brief, a large quantity of well-fed worms were washed off NGM plates using M9 buffer and then resuspended in distilled water. The worms were frozen on dry ice for at least 20 min, after which they were freeze-cracked using two Super Frost Plus slides. The worms were

then fixed with 4% paraformaldehyde for 15 min at room temperature. Following fixation, the worms were incubated in blocking solution for 30 min before being treated with primary antibodies: anti-mNeuoGreen (A24858, Abclonal, 1:100) and anti-mCherry (E022110-03, EarthOx, 1:500) for overnight incubation at 4 °C. After thorough washing, the samples were incubated with secondary antibodies Abclonal Alexa 488 (AS053, Abclonal, 1:300) and Alexa 594 (A11005, Thermo Fisher Scientific, 1:1000) for 3 h. Finally, the samples were mounted on slides for subsequent imaging experiments.

## Statistics and reproducibility

The Macro-Manager software was used to collect the calcium imaging data, which was then analyzed with Image J. The statistical analysis was performed using GraphPad Prism 8 and the results were presented as mean ± standard error of the mean (S.E.M.). All data in the same graph were analyzed for type of data distribution through the Shapiro-Wilk normality test. For graphs with only two datasets to compare, unpaired two-tailed Student's *t*-test was used for normally distributed data, followed by Welch's correction when groups with unequal variances. The unpaired two-tailed Mann-Whitney U test was used for non-normally distributed data. For three or more test groups, when the data are normally distributed with equal variances, a one-way ANOVA is employed. If the variances are not equal, Bonferroni's multiple comparison correction is applied. The Kruskal-Wallis test was used for non-normally distributed data. P-values were used for the statistical significance, with the following notations: n.s. $P > 0.05$, $*P < 0.05$, $**P < 0.01$, $***P < 0.001$, and $****P < 0.0001$. All behavioral assays were double-blinded. The experiments were randomized, and the sample sizes used were consistent with typical sample sizes in the field and were determined based on experiment replicability.

## Reporting summary

Further information on research design is available in the Nature Portfolio Reporting Summary linked to this article.

## Data availability

Source data for all main figures and Supplementary Figs. are supplied with this paper. Any additional information required to reanalyze the data reported in this paper is available from the lead contact (kanglijun@zju.edu.cn) upon request. Source data are provided with this paper.

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

## Acknowledgements

This work was supported by grants from STI2030 Major Projects (2021ZD0203303 to L.K.), the National Foundation of Natural Science of China (32271017, 31771113 and 31471023 to L.K.; 31800878 to W.J.Z.), Zhejiang Provincial Natural Science Foundation (LZ22C090001 to L.K.; Y24C090011 to W.J.Z.), Research Center of Prevention and Treatment of Senescence Syndrome of School of Medicine Zhejiang University (2022020001), and the Non-profit Central Research Institute Fund of Chinese Academy of Medical Sciences (2023-PT310-01). The funders had no role in study design, data collection and analysis, decision to publish or preparation of the manuscript. We thank Sanhua Fang, Li Liu and Dan Yang from the Core Facilites, Zhejiang University School of Medicine for technical support, and thank the *Caenorhabditis* Genetic Center (CGC), which is supported by the National Institutes of Health—Office of Research Infrastructure Programs (P40 OD010440), for strains.

## Author contributions

W.J.Z. and L.J.K. conceived and designed the experiments, W.J.Z., Y.D.F., J.L., H.K.C., H.T.H., S.T.L., L.H.Z., R.L., and L.Y.H. performed molecular genetics, calcium imaging, electrophysiological and behavioral experiments. W.J.Z., Y.D.F. and L.J.K. analyzed and interpreted the results. Y.Q.T., G.H.Z., Y.M.Z., F.W., R.Y.Z., and X.J.Z. commented on the design and the implementation of the experiments. W.J.Z., Y.D.F., U. A., L.H.Z., and L.J.K. wrote the manuscript. All authors contributed to the final version of the manuscript.

## Competing interests

The authors declare no competing interests.
