## [Transparent Peer Review file · Nature Communications]

Anoctamin-1 is a core component of a mechanosensory anion channel complex in *C. elegans*

Corresponding Author: Professor Lijun Kang

Version 0:

Reviewer comments:

Reviewer #1

(Remarks to the Author)

Mechanotransduction channels mediate important physiological functions such as touch and hearing. Here, the authors show that in *C. elegans* sensory neurons, expression of ANOH-1, a homolog of mammalian TMEM16A, is required for mechanoreceptor currents that mediate touch responses. The authors further show that the putative mechanosensitivity of ANOH-1 is dependent on CALM1, a homolog of the calcium- and integrin-binding protein (CIB2), potentially by forming a channel complex. It was hypothesised that this complex might be tethered to ankyrin, via which mechanical forces might be transmitted to the channel. This study made extensive use of *C. elegans* genetics, and can be strengthened by considering the following.

1. For ion selectivity experiments, the change from NaCl to Na-Glu or to NMDG-Cl would result in a large junction potential given the difference in ion mobility of the substituted ions. The authors should describe in more detail how junction potential was minimized during the exchange of solutions and how this was corrected for.
2. The authors should include the I-V plots for the control condition for all ion selectivity experiments (Fig. 4f, 5d, 6c). More specifically, the authors should compare a) CsCl (IS)/NaCl (ES) with Cs-Glu (IS)/NaCl (ES), b) CsCl (IS)/NaCl (ES) with CsCl (IS)/Na-Glu (ES), and c) CsCl (IS)/NaCl (ES) with CsCl (IS)/NMDG-Cl (ES).
3. Do the reversal potentials match the Nernst potential of Cl under the extracellularly and intracellularly substituted conditions? Please discuss.
4. Please provide in the methods the typical pipette resistance, seal resistance, and series resistance, and whether the pipettes were fire-polished.
5. The authors should indicate the zero current level in all panels where current traces are shown.
6. The authors have illustrated in Fig. 6f the putative location of K588 in ANOH-1. According to the provided sequence alignment in Fig. 6d, this corresponds to K645 in mouse TMEM16A, which is in the transmembrane region. Mutation of this residue in mouse TMEM16A does not abolish current activity, but profoundly changes the ion conduction properties that generally lowers the conductance.
 - a. The authors should characterise the electrophysiological properties of ANOH-1 in a heterologous system.
 - b. The authors should examine the effect of K588 mutations in direct comparison with the wild-type ANOH-1 in the same system.
 - c. The authors should revise the diagram to depict the correct location of this residue; the equivalent residue in TMEM16A is not in the extracellular loop, it is on the intracellular half of helix 6.
7. Given that the structure-function relationship of mouse/human TMEM16A is much better characterised and that this homolog can also rescue the mechanically activated responses, the authors should include mutations that severely perturb ion conduction, such as K588A and K645A in mouse TMEM16A (numbering for the ac variant), to show that these mutations can inhibit the rescue.
8. The authors should provide a UniProt identifier for the described TMEM16 homolog in *C. elegans*.
9. In the co-IP results shown in Fig. 7f, what is the molecular weight of ANOH-1 as it appears to run exactly the same as human TMEM16A.
10. The molecular weight where the human TMEM16A runs does not seem to correspond to its monomeric size of around 110 kDa, which typically runs in its monomeric form in SDS-PAGE under reducing conditions. If this is due to glycosylation, please provide data on a PNGase-digested sample, or otherwise provide explanation.
11. The authors should provide data on co-IP experiments using IP: FLAG and WB: HA.
12. The authors should demonstrate in a heterologous system the co-expression of CIB2/Ano1 and CALM1/ANOH-1 gives

rise to mechanosensitive ionic currents and that their ion selectivity would resemble that observed in the native systems shown in the manuscript. The authors should also include these data if the co-expression of these constructs in the heterologous system does not result in mechanosensitive currents.

13. The authors should investigate in excised patches whether CIB2/Ano1 and CALM1/ANOH-1 are intrinsically mechanosensitive, should this give rise to mechanosensitive currents.

14. The authors should provide data on CALM1 expression in all the neuronal cell types investigated and that CALM1 is not expressed in ASH neurons where ANOH-1 does not function as a mechanoreceptor channel (Fig. S1); these are an absolute requirement for the proposal that CALM1/ANOH-1 forms a mechanoreceptor complex.

15. Although the UNC-44-null background does not display mechanoreceptor currents, there is no direct evidence in this manuscript that CALM1/ANOH-1 would be tethered to UNC-44 and that this tethering is responsible for mechanotransduction. The authors should indicate that this is a speculation or a hypothesis in both the schematic and in the legend in cases where no direct evidence is presented and provide supporting references where needed.

16. Without calcium, the current passing through TMEM16A is highly outwardly rectifying due to the highly negative electrostatic potential at the vacant calcium binding site and in the inner pore, which effectively gates ion conduction. This has a consequence that any inward current is minimal even at 150 mM intracellular Cl and hence a minimal capacity to give rise to a depolarising current in the absence of bound calcium under resting cellular conditions. The authors should acknowledge this discrepancy and how this would affect their hypothesis.

17. There are typos throughout in the manuscript, please revise.

Reviewer #2

(Remarks to the Author)

Using in vivo calcium imaging and whole-cell patch clamp recording, Zou et al investigate mechanosensation by sensory neurons in male and hermaphrodite *C. elegans*. Specifically, the team reports that the male-specific CEM neurons are required for responses to nose-touch and respond to mechanical stimulation with an increase in intracellular calcium. They also report that the ASJ neurons in hermaphrodites also function as mechanoreceptors. Several lines of evidence point to mechanoreceptor currents that are chloride-dependent and require expression of ANOH-1/anoctamin (aka TMEM16A). The activity of ANOH-1 requires co-expression of CALM-1/CIB and UNC-44/ankyrin in cells natively expressing ANOH-1 as well as in cells ectopically expressing this protein. Generally speaking, the experimental data are consistent with a model in which ANOH-1 functions as a mechanically-activated Cl⁻ channel.

Some of the conclusions are phrased in a manner that is stronger than the data warrant and more detail is required to fully understand some of the methods and to evaluate the quality of the data. Additionally, the flow of the paper could be improved. The shift in focus from CEM neurons in males to ASJ neurons in hermaphrodites is a bit jarring. Given the substantive literature on anoctamin channels from other species, there are additional opportunities for improvement that will require additional experiments. In my opinion, these experiments are not required, but would improve the study.

Conceptual issues or concerns

1. Chloride channels that depolarize neurons

The authors correctly note that activation of Cl⁻ channel can depolarize a neuron under some conditions and hyperpolarize a neuron under others. This is central to their clever experimental design of using ectopically expressed His-Cl to show that activating a Cl⁻ channel increases intracellular calcium (likely through depolarization). However, their discussion of how this works is overly simplified and somewhat misleading. I am sympathetic that this concept can be difficult to explain to a general audience. Still, I urge the authors to write in a manner that is both precise and accessible.

The effect of activating a Cl⁻ channel on membrane potential depends on the chloride gradient across the plasma membrane or the equilibrium potential for Cl⁻ (E-Cl), not only the intracellular or extracellular concentrations of Cl⁻ ions. It also depends on the resting membrane potential, V_{rest}. If V_{rest} is below E-Cl, then activating a Cl⁻ channel will DEPOLARIZE the membrane. Conversely, if V_{rest} is above E-Cl, then activating a Cl⁻ channel will HYPERPOLARIZE the membrane. The finding that activating ectopically expressed His-Cl in CEM and ASJ increases intracellular calcium suggests (but does not prove) that V_{rest} is below E-Cl. This finding also has implications for the widespread use of His-Cl to INACTIVATE neurons in *C. elegans* and suggests that activation of His-Cl may have different effects in other neurons. Consider commenting on the implications of this observation in the Discussion and also citing Pokala N, et al. PNAS 2014 111(7):2770-5. doi: 10.1073/pnas.1400615111 who claim that His-Cl silences *C. elegans* neurons.

2. Imprecise text related to point #1

- “The opening of a chloride channel can either excite or inhibit cells, depending on the concentration of intracellular chloride ions.”

The first clause of this sentence is correct, but the second lacks precision. The concentration of intracellular Cl⁻ is only one factor. Whether or not a Cl⁻ channel excites or inhibits a cell depends on the difference between membrane potential and the ratio of intracellular to extracellular chloride.

- “Because the extracellular chloride level is typically higher than the intracellular, the electrochemical gradient will drive chloride influx and hyperpolarize the plasma membrane.”

Cl⁻ influx will hyperpolarize the plasma membrane if, and only if, the resting potential is more positive than E-Cl.

3. The role of kcc-3 and referencing the literature

It is possible that *kcc-3* mutants have smaller mechanically-evoked Ca^{2+} signals in CEM neurons due to changes in the Cl^- gradient across the plasma membrane, as implied by the authors. However, other explanations are possible, such as a change in membrane potential or morphological defects in cilia shape (see Singhvi et al, 2016). If the author's inference is correct, then loss of *kcc-3* should also reduce HisCl calcium responses in CEM. The authors should perform these additional experiments to strengthen these inferences.

It would also be helpful to determine where *kcc-3* and *abts-3* are expressed and function in this context. Investigating the AFD neurons, Singhvi et al, 2016 have shown that *kcc-3* is expressed in the amphid sheath cell, a glial cell that encapsulates amphid sensory neurons (including ASJ) in hermaphrodites.

4. Insufficient support exists to support the claim that MRCs are Cl^- selective

The authors show that the reversal potential for MRCs is sensitive to changes in E- Cl^- , indicating that the channels are Cl^- permeable. This is necessary, but not sufficient evidence that the channels are SELECTIVE for Cl^- . Additional experiments would be needed to support this claim, such as measurements of relative permeability. In my opinion, these experiments would be nice to have, but are not essential. Without additional data, however, the authors should revise the text to match the strength of their finding. In other words, the data support the conclusion that MRCs are carried primarily by Cl^- ions but not the conclusion that the channels carrying this current are Cl^- selective.

5. More information is needed about the CEM neurons in males.

Please inform the reader that CEM neurons were previously considered to be chemosensory and not mechanosensory. The authors findings suggest that they are multimodal, which is interesting and should be highlighted. Finally, no data are presented confirming that ANOH-1 is expressed in CEM. It seems reasonable to guess this is the case, but direct experimental evidence would be better.

6. More discussion is needed about the results in isolated ASJ neurons

This concern has technical and conceptual elements. Technically, it is hard to imagine how the experimenters obtained recordings from a cell body that is 3-5 μm in diameter and delivered a 4 μm indentation while maintaining a high-quality recording. What size is the recording pipette tip and glass stimulator that is used for these experiments? The schematic shown in Figure 5a is misleading, as it implies that the cell body is much bigger than the recording pipette or the stimulator. This is simply not possible given the size of the ASJ neuron and dimensions of recording pipettes and stimulators. Conceptually, more discussion is needed about the implications that ASJ retains mechanosensitivity when isolated.

7. The data in this paper do not support the conclusion that ANOH-1 activation in *C. elegans* neurons depends on a gating spring (Discussion).

The concept of a gating spring refers to a specific biophysical mechanism for converting cell deformation into channel activation. Neither these authors nor others have demonstrated that mechanosensitivity of ANOH-1 or any other anoctamin depends on a gating spring. The finding that ANOH-1 activity requires CALM-1 expression is not sufficient to draw this inference. This experimental finding is sufficient to support a weaker claim, however — namely that CALM-1 and ANOH-1 interact with another.

Composition

1. The Results section is divided into many short subsections that make it hard to read.

If the authors were to consolidate short (1 paragraph) subsections in the Results into larger chunks, this would be helpful to readers. In particular Figure 1 is discussed in the first three sections, which could be a single section.

2. Reordering the Results Section for better flow.

In the Results subsection "ANOH-1 mediates mechanotransduction in ASJ neurons", the first paragraph begins with a switch of focus from CEM to ASJ neurons, which feels a bit sudden and lacks rationale; only mentioning the experimental difficulty in males vs hermaphrodites is not enough (Line 170-171).

The above issue can be avoided by reordering and consolidating the short subsections (as mentioned in 1). For example, sections "ANOH-1 is required for the mechanosensitivity of CEM neurons" and "ANOH-1 mediates mechanotransduction in ASJ neurons" can be consolidated. Moreover, ANOH-1's role in mediating touch-avoidance behavior can be discussed in a single section by consolidating ANOH-1's separate roles in CEM and ASJ neurons, respectively. The dependence of ANOH-1 on chloride ions can be discussed afterwards.

The central idea of reordering and consolidating subsections is to treat ANOH-1 as the main character in the story and use it to conceptually determine subsections (namely, a. ANOH-1's role in neuronal responses (CEM, ASJ); b. ANOH-1's role in animal behavior in the context of neurons; c. ANOH-1's physiological, molecular, genetic mechanisms).

3. Figure legends should be focused on the data presented in each figure. Throughout the manuscript, figure legends contain both interpretation and description of methods that does not belong.

4. More context is needed for readers to understand the role of using *osm-9* mutants in behavioral assays (Fig 2b).

Technical issues

1. Please inform the reader how liquid junction potentials determined. Were corrections applied for conditions in which Cl^-

- was replaced intracellularly and extracellularly? Such junction potentials can be significant and if not properly corrected, they might compromise measurements of reversal potential.
2. Please disclose the range of series resistance values and the fraction of the series resistance that was not compensated. This can affect I-V curves, as the actual membrane voltage will depart from the command voltage by an amount equal to the current flowing multiplied by the uncompensated series resistance. If the latter is high, then the voltage error can be substantial.
 3. Please tell the reader if the experimenters performing experiments (especially behavior, which is scored manually) were aware of the genotypes under test or if these experiments were performed blind to genotype.
 4. The sample sizes used in Figures are indicated by n, but the sample's identity is not clear. For example, in Fig 1a, n=35, is this the number of trials or animals? The authors need to specify that in the figure captions at least once on the first appearance.
 5. At Line 126, the description of dissecting worm's cuticle is also mentioned in Fig 1b caption, which is redundant. Consider moving the description in Fig 1b caption.
 6. Figs 3a and 4a show schematic or fluorescence images of ASJ anatomy; this is good. Can the authors also show CEM anatomy in Fig 1a using either schematics or fluorescence images?
 7. In Lines 188-189 and Fig 4c, how was the latency time calculated and indicated? Authors should specify it more clearly in the manuscript.
 8. At the start of several calcium traces the $\Delta F/F$ values are not zero (e.g. Fig. 3c, His-CI trace), suggesting that the reference for the F0 value was taken from a different time in the trace. Please indicate that time and the rationale for choosing it.
 9. Calcium traces in Figure 1 display different time frames. Control responses in Figure 1a reach a peak in about 30s after stimulation. The time course in Figure 1b is much slower. Why?
 10. Niflumic acid is known as a chloride channel blocker, but it is not specific for this target. Please revise the text accordingly.
 11. Loss of *anoh-1* does not completely eliminate MRCs in ASJ. And, the kinetics of the residual current are faster than that shown in wild-type. Please comment on the residual MRC.
 12. Does the mutation described as a pore-dead mutation affect reversal potential or selectivity? If so, this would provide additional evidence that ANOH-1 carries the MRCs.
 13. Please move the ASH data to main paper from supplement.

Reviewer #3

(Remarks to the Author)

Using *in vivo* calcium imaging and whole-cell patch clamp recording, Zou et al investigate mechanosensation by sensory neurons in male and hermaphrodite *C. elegans*. Specifically, the team reports that the male-specific CEM neurons are required for responses to nose-touch and respond to mechanical stimulation with an increase in intracellular calcium. They also report that the ASJ neurons in hermaphrodites also function as mechanoreceptors. Several lines of evidence point to mechanoreceptor currents that are chloride-dependent and require expression of ANOH-1/anoctamin (aka TMEM16A). The activity of ANOH-1 requires co-expression of CALM-1/CIB and UNC-44/ankyrin in cells natively expressing ANOH-1 as well as in cells ectopically expressing this protein. Generally speaking, the experimental data are consistent with a model in which ANOH-1 functions as a mechanically-activated Cl⁻ channel.

Some of the conclusions are phrased in a manner that is stronger than the data warrant and more detail is required to fully understand some of the methods and to evaluate the quality of the data. Additionally, the flow of the paper could be improved. The shift in focus from CEM neurons in males to ASJ neurons in hermaphrodites is a bit jarring. Given the substantive literature on anoctamin channels from other species, there are additional opportunities for improvement that will require additional experiments. These experiments are not required, but would improve the study.

Conceptual issues or concerns

1. Chloride channels that depolarize neurons

The authors correctly note that activation of Cl⁻ channel can depolarize a neuron under some conditions and hyperpolarize a neuron under others. This is central to their clever experimental design of using ectopically expressed His-CI to show that activating a Cl⁻ channel increases intracellular calcium (likely through depolarization). However, their discussion of how this works is overly simplified and somewhat misleading. I am sympathetic that this concept can be difficult to explain to a general audience. Still, I urge the authors to write in a manner that is both precise and accessible.

The effect of activating a Cl⁻ channel on membrane potential depends on the chloride gradient across the plasma membrane or the equilibrium potential for Cl⁻ (E_{Cl}), not only the intracellular or extracellular concentrations of Cl⁻ ions. It also depends on the resting membrane potential, V_{rest} . If V_{rest} is below E_{Cl} , then activating a Cl⁻ channel will DEPOLARIZE the membrane. Conversely, if V_{rest} is above E_{Cl} , then activating a Cl⁻ channel will HYPERPOLARIZE the membrane. The finding that activating ectopically expressed His-CI in CEM and ASJ increases intracellular calcium suggests (but does not prove) that V_{rest} is below E_{Cl} . This finding also has implications for the widespread use of His-CI to INACTIVATE neurons in *C. elegans* and suggests that activation of His-CI may have different effects in other neurons. Consider commenting on the implications of this observation in the Discussion and also citing Pokala N, et al. PNAS 2014 111(7):2770-5. doi: 10.1073/pnas.1400615111 who claim that His-CI silences *C. elegans* neurons.

2. Imprecise text related to point #1

- “The opening of a chloride channel can either excite or inhibit cells, depending on the concentration of intracellular chloride ions.”

The first clause of this sentence is correct, but the second lacks precision. The concentration of intracellular Cl⁻ is only one factor. Whether or not a Cl⁻ channel excites or inhibits a cell depends on the difference between membrane potential and the ratio of intracellular to extracellular chloride.

- “Because the extracellular chloride level is typically higher than the intracellular, the electrochemical gradient will drive chloride influx and hyperpolarize the plasma membrane.”

Cl⁻ influx will hyperpolarize the plasma membrane if, and only if, the resting potential is more positive than E-Cl⁻.

3. The role of kcc-3 and referencing the literature

It is possible that kcc-3 mutants have smaller mechanically-evoked Ca²⁺ signals in CEM neurons due to changes in the Cl⁻ gradient across the plasma membrane, as implied by the authors. However, other explanations are possible, such as a change in membrane potential or morphological defects in cilia shape (see Singhvi et al, 2016). If the author's inference is correct, then loss of kcc-3 should also reduce HisCl calcium responses in CEM. The authors should perform these additional experiments to strengthen these inferences.

It would also be helpful to determine where kcc-3 and abts-3 are expressed and function in this context. Investigating the AFD neurons, Singhvi et al, 2016 have shown that kcc-3 is expressed in the amphid sheath cell, a glial cell that encapsulates amphid sensory neurons (including ASJ) in hermaphrodites.

4. Insufficient support exists to support the claim that MRCs are Cl⁻ selective

The authors show that the reversal potential for MRCs is sensitive to changes in E-Cl⁻, indicating that the channels are Cl⁻ permeable. This is necessary, but not sufficient evidence that the channels are SELECTIVE for Cl⁻. Additional experiments would be needed to support this claim, such as measurements of relative permeability. In my opinion, these experiments would be nice to have, but are not essential. Without additional data, however, the authors should revise the text to match the strength of their finding. In other words, the data support the conclusion that MRCs are carried primarily by Cl⁻ ions but not the conclusion that the channels carrying this current are Cl⁻ selective.

5. More information is needed about the CEM neurons in males.

Please inform the reader that CEM neurons were previously considered to be chemosensory and not mechanosensory. The authors findings suggest that they are multimodal, which is interesting and should be highlighted. Finally, no data are presented confirming that ANOH-1 is expressed in CEM. It seems reasonable to guess this is the case, but direct experimental evidence would be better.

6. More discussion is needed about the results in isolated ASJ neurons

This concern has technical and conceptual elements. Technically, it is hard to imagine how the experimenters obtained recordings from a cell body that is 3-5µm in diameter and delivered a 4µm indentation while maintaining a high-quality recording. What size is the recording pipette tip and glass stimulator that is used for these experiments? The schematic shown in Figure 5a is misleading, as it implies that the cell body is much bigger than the recording pipette or the stimulator. This is simply not possible given the size of the ASJ neuron and dimensions of recording pipettes and stimulators. Conceptually, more discussion is needed about the implications that ASJ retains mechanosensitivity when isolated.

7. The data in this paper do not support the conclusion that ANOH-1 activation in *C. elegans* neurons depends on a gating spring (Discussion).

The concept of a gating spring refers to a specific biophysical mechanism for converting cell deformation into channel activation. Neither these authors nor others have demonstrated that mechanosensitivity of ANOH-1 or any other anoctamin depends on a gating spring. The finding that ANOH-1 activity requires CALM-1 expression is not sufficient to draw this inference. This experimental finding is sufficient to support a weaker claim, however — namely that CALM-1 and ANOH-1 interact with another.

Composition

1. The Results section is divided into many short subsections that make it hard to read.

If the authors were to consolidate short (1 paragraph) subsections in the Results into larger chunks, this would be helpful to readers. In particular Figure 1 is discussed in the first three sections, which could be a single section.

2. Reordering the Results Section for better flow.

In the Results subsection “ANOH-1 mediates mechanotransduction in ASJ neurons”, the first paragraph begins with a switch of focus from CEM to ASJ neurons, which feels a bit sudden and lacks rationale; only mentioning the experimental difficulty in males vs hermaphrodites is not enough (Line 170-171).

The above issue can be avoided by reordering and consolidating the short subsections (as mentioned in 1). For example, sections “ANOH-1 is required for the mechanosensitivity of CEM neurons” and “ANOH-1 mediates mechanotransduction in ASJ neurons” can be consolidated. Moreover, ANOH-1's role in mediating touch-avoidance behavior can be discussed in a single section by consolidating ANOH-1's separate roles in CEM and ASJ neurons, respectively. The dependence of ANOH-1 on chloride ions can be discussed afterwards.

The central idea of reordering and consolidating subsections is to treat ANOH-1 as the main character in the story and use it to conceptually determine subsections (namely, a. ANOH-1's role in neuronal responses (CEM, ASJ); b. ANOH-1's role in animal behavior in the context of neurons; c. ANOH-1's physiological, molecular, genetic mechanisms).

3. Figure legends should be focused on the data presented in each figure. Throughout the manuscript, figure legends contain both interpretation and description of methods that does not belong.

4. More context is needed for readers to understand the role of using *osm-9* mutants in behavioral assays (Fig 2b).

Technical issues

1. Please inform the reader how liquid junction potentials determined. Were corrections applied for conditions in which Cl⁻ was replaced intracellularly and extracellularly? Such junction potentials can be significant and if not properly corrected, they might compromise measurements of reversal potential.

2. Please disclose the range of series resistance values and the fraction of the series resistance that was not compensated. This can affect I-V curves, as the actual membrane voltage will depart from the command voltage by an amount equal to the current flowing multiplied by the uncompensated series resistance. If the latter is high, then the voltage error can be substantial.

3. Please tell the reader if the experimenters performing experiments (especially behavior, which is scored manually) were aware of the genotypes under test or if these experiments were performed blind to genotype.

4. The sample sizes used in Figures are indicated by n, but the sample's identity is not clear. For example, in Fig 1a, n=35, is this the number of trials or animals? The authors need to specify that in the figure captions at least once on the first appearance.

5. At Line 126, the description of dissecting worm's cuticle is also mentioned in Fig 1b caption, which is redundant. Consider moving the description in Fig 1b caption.

6. Figs 3a and 4a show schematic or fluorescence images of ASJ anatomy; this is good. Can the authors also show CEM anatomy in Fig 1a using either schematics or fluorescence images?

7. In Lines 188-189 and Fig 4c, how was the latency time is calculated and indicated? Authors should specify it more clearly in the manuscript.

8. At the start of several calcium traces the $\Delta F/F$ values are not zero (e.g. Fig. 3c, His-Cl trace), suggesting that the reference for the F0 value was taken from a different time in the trace. Please indicate that time and the rationale for choosing it.

9. Calcium traces in Figure 1 display different time frames. Control responses in Figure 1a reach a peak in about 30s after stimulation. The time course in Figure 1b is much slower. Why?

10. Niflumic acid is known as a chloride channel blocker, but it is not specific for this target. Please revise the text accordingly.

11. Loss of *anoh-1* does not completely eliminate MRCs in ASJ. And, the kinetics of the residual current are faster than that shown in wild-type. Please comment on the residual MRC.

12. Does the mutation described as a pore-dead mutation affect reversal potential or selectivity? If so, this would provide additional evidence that ANOH-1 carries the MRCs.

13. Please move the move ASH data to main paper from supplement.

Version 1:

Reviewer comments:

Reviewer #1

(Remarks to the Author)

Despite the initial revision, many technical aspects of the study remain to be addressed.

The authors should provide the line numbers, make references to the changes made to the figures, and include a quotation of the specific changes made to the text in all the individual responses.

1. Title and Line 30. Please introduce ANOH-1 as the *C. elegans* Anoctamin-1 in both the title and abstract as Anoctamin-1 is often referred to as the mammalian TMEM16A, especially given that the sequence homology between these two homologs is only ~30%. Please also state in the title that this is specific to worms in order to avoid false generalization.

2. Title and Line 31. Please describe ANOH-1 as an essential component of a mechanosensory channel complex in *C. elegans*. The current description suggests that the channel is itself gated by mechanical force, for which there is no direct evidence in the presented data.

3. Line 39. Please state instead 'anion channels in mechanosensory transduction in metazoans'. The authors should refrain from once again giving the impression that ANOH-1 or ANO1 are direct sensors of mechanical force.

4. Line 96-107. The description that Anoctamin-1 chloride channel can be directly activated by forces is misleading, as is the description that Anoctamin-1 chloride channel possess the capability to respond to mechanical stimulation. The same goes for the description that Anoctamin-1 serves as a native mechanosensor. The data suggest that this channel might be required for the observed mechanosensitive currents but do not provide any direct evidence for these statements. Please revise and only make statements in accord to the level of support that the current data offer.

5. Line 328-331. The molecular weight of ANOH-1 is ~82 kDa and human ANO1 is ~99 kDa.

6. Fig. S8 and ectopic expression control. Both the ANOH-1 and ANO1 constructs appear to give rise to diffuse cytosolic

fluorescence when expressed in a recombinant system (Fig. S8), which are properties contradicting to that of integral membrane proteins. This raises the question whether the ectopic expression of these constructs leads to correctly folded proteins that are localized to the plasma membrane in the neurons investigated in this study. Given this observation, the authors should ensure that the ectopic expression of both ANOH-1 and ANO1 results in 1) the full-length protein and 2) proper localization to the plasma membrane in the neurons investigated. Please provide these controls.

Unaddressed concerns

1. There is no automatic compensation that can correct for liquid junction potentials (LJP). Junction potentials exist not only at the electrode-solution interface, but importantly also between the bath and pipette solutions. The magnitude of LJP is different for different pairs of bath and pipette solutions and will have to be calculated and corrected for each individual pairs of solutions. Zeroing the pipette offset when the pipette is in the bath solution does not correct for any LJP, as a potential across the electrode has been applied to shift this offset which is in part contributed by the LJP between the bath and pipette solutions. Given the mobility differences between the substituted ions, some of these LJPs can be as large as 10 mV, which is in the range shown in Fig. 4f, 5d, 6c in the revised manuscript. The proper way to correct for LJP is to correct the applied voltage offline according to 'Neher. (1992) Correction for liquid junction potentials in patch clamp experiments. *Methods in Enzymology*. Volume 207, 1992, Pages 123-131'. Please revise all the voltages involved.
2. The substitution of chloride with gluconate on the intracellular and extracellular side should result in the same or very similar magnitude of reversal potential but with opposite polarity. The fact that this is not the case in the presented data in Fig. 4f, 5d, 6c, in addition to possible incomplete solution exchange in the intracellular compartment, is an indication of a likely voltage offset, especially given that the LJPs in all the recordings were never properly corrected for. Please ensure that all the plotted voltages are properly corrected for any LJPs. Please revise all the voltages involved.
4. Please detail how a sufficient voltage-clamp was achieved with series resistance values in the range of 50-70 MOhm as a significant fraction of the applied voltage will drop across the pipette. These series resistances irreversibly distort the I-V plots and the current transients recorded at the same applied voltage. Comparison based on reversal potentials and current amplitudes will both be severely affected and will therefore not be valid. This is a serious issue that affects all the recordings throughout the manuscript and must be rigorously addressed.
5. The reason for showing the zero-current level is allow the readers to appreciate the background current levels and how tight the seals are. The authors are likely aware that it is inappropriate to display the 'background'-subtracted traces over the zero-magnitude as the background current levels were set arbitrarily to zero. For all the non-overlaid traces, please show the representative traces without background subtraction together with the actual zero-current level, including those in the supplementary figures. For the background-subtracted overlaid traces, please remove the zero-level line as this is misleading.
6. The fluorescence of presumably N-terminally tagged ANOH-1 in HEK-P1KO cells appears to show no features corresponding to an integral membrane protein, which could possibly be due to a truncation between the EGFP and ANOH-1 or that ANOH-1 might not be correctly folded (Fig. S8e). EGFP-hANO1 does not appear to be localized to the plasma membrane either, with prominent cytosolic fluorescence, similar to ANOH-1 (Fig. S8e). The authors should include data using a C-terminal tag to illustrate the localization of both hANO1 and ANOH-1 in HEK-P1KO cells, and as requested in point 6, please provide the described ectopic expression controls.
8. The provided UniProt identifier of the TMEM16 homolog in *C. elegans* is incorrect, please revise.
- 9-11. The difference in molecular weight between ANOH-1 (~82 kDa) and human ANO1 (~99 kDa) is almost 20 kDa for a monomer and 40 kDa for a dimer. These should be size differences that are discernable on an SDS-PAGE, yet the Western blot in Fig. 7f shows that they run at exactly the same molecular weight as if they were the same protein. Please ensure the validity of these experiments and provide the uncropped images of the entire gel for these co-IP experiments with proper labels in the response.

Reviewer #2

(Remarks to the Author)

Zou et al have undertaken a substantial revision to their manuscript, providing additional support for the study's main conclusions through new data, clarification recording conditions, and revising text and figures. The updated organization of the manuscript is a significant improvement. Well one.

The key findings that ANOH-1/anoctamin (aka TMEM16A) functions as a mechanosensitive ion channel in multiple *C. elegans* neurons is well supported, as are experiments showing that human proteins can substitute for the nematode proteins.

There are at least two significant technical concerns that require additional attention.

1. Errors in measuring MRC reversal potentials from whole-cell patch clamp recordings of *C. elegans* neurons in vivo and dissociated in culture

Both reviewers commented on the contribution of liquid junction potentials (EJPs) to voltage measurements in whole-cell patch clamp recordings. The authors' response "Upon the pipette's entry into the bath solution, we compensated for the liquid junction potentials using the automatic compensation of the EPC-10 amplifier along with the Patchmaster software." reflects a common misconception about liquid junction potentials.

It is standard practice to zero voltage offsets evident upon placing a patch clamp electrode into the extracellular bath solution. However, this act is not sufficient by itself to correct for liquid junction potentials (which can be substantial). Quoting from a technical note from Axon Instruments: "Most patch clamp experiments need to correct for liquid junction potentials.

When the recording pipette is first inserted in the bath, there are voltage offsets that are corrected by the amplifier when the current is zeroed (i.e., in voltage clamp mode). The offsets consist of liquid junction potentials and potential differences between solid electrodes and the solutions they are in contact with (the “electrode” or “half-cell” potentials).” (Axon Instruments technical note, 2004)

The authors use hardware and software from HEKA. The Patchmaster software from HEKA does include a semi-automatic correction for EJP. However, the user MUST enter a value for the LJ in order for the correction to be applied.

These errors in membrane potential that arise from EJPs can be 10s of mV and must be properly corrected in order to compare measured reversal potentials across recordings performed with distinct combinations of intracellular and extracellular solutions. (see attached screenshot of the HEKA manual)

If the authors used the correction feature of the Patchmaster software, it is imperative that they report values of the LJ corrections that were used.

If they did not use this feature and only zeroed voltage offsets prior to obtaining tight seals, the LJ is a systematic error in membrane voltage that permeates all electrophysiology measurements here. And are particularly important for measurements of reversal potential. Fortunately, this voltage error can be corrected posthoc. This will involve determining LJ values (either experimentally or estimating using a calculator such as this one <https://sw Harden.com/LJPCalc/> or other tools) and then applying the correction to the membrane potential.

Since the determination of ion selectivity by varying solutions is central to the study, this issue must be addressed properly prior to publication. It is possible that after correcting for LJ, the measured changes in reversal potential better match the values predicted from the GHK equation.

2. Uncertainty in the amplitude of mechanical indentations delivered to dissociated and cultured ASJ neurons

Thank you for revising the figure illustrating mechanical stimulation of ASJ neurons in culture and for providing additional information about the stimulation of these tiny cells. This reviewer is acutely aware of the technical challenge represented here. Still, the authors’ response raises new concerns.

It seems that they used a stimulus that moved the actuator 4 μ m but that the thin probe bent and likely generated a smaller indentation. The authors note that using a thicker probe destroyed recordings and or moved the cell, which is understandable and sensible. The authors should find a way to communicate to readers in the results section that the actual indentation is less than 4 μ m and likely to be of a similar amplitude during each recording (but not across recordings). One idea might be to relabel the stimulus trace in Fig 5c as “stimulator movement” or something similar.

Formatting and citations suggestions (minor)

1. (line 5) epithelial Na channels (ENaC) are part of a large superfamily, consider expanding to include other subfamilies by replacing "epithelia Na⁺ channels (ENaC)" with "degenerin, epithelial, and acid-sensing Na⁺ channels (DEG/ENAC/ASIC)"
2. (line 52) The review articles cited do not cover OSCA/TMEM63, consider adding Goodman, Vasquez, & Haswell J Gen Physiol 2023 that does cover OSCA/TMEM63
3. (line 308-309) While this study was in review, another paper was published reporting the expression of CALM-1 in *C. elegans*, please compare findings.

Reviewer #3

(Remarks to the Author)

Zou et al have undertaken a substantial revision to their manuscript, providing additional support for the study’s main conclusions through new data, clarification recording conditions, and revising text and figures. The updated organization of the manuscript is a significant improvement. Well one.

The key findings that ANOH-1/anoctamin (aka TMEM16A) functions as a mechanosensitive ion channel in multiple *C. elegans* neurons is well supported, as are experiments showing that human proteins can substitute for the nematode proteins.

There are at least two significant technical concerns that require additional attention.

1. Errors in measuring MRC reversal potentials from whole-cell patch clamp recordings of *C. elegans* neurons in vivo and dissociated in culture

Both reviewers commented on the contribution of liquid junction potentials (EJPs) to voltage measurements in whole-cell patch clamp recordings. The authors’ response reflects a common misconception about liquid junction potentials.

It is standard practice to zero voltage offsets evident upon placing a patch clamp electrode into the extracellular bath solution. However, this act is not sufficient by itself to correct for liquid junction potentials (which can be substantial). Quoting from a technical note from Axon Instruments: “Most patch clamp experiments need to correct for liquid junction potentials. When the recording pipette is first inserted in the bath, there are voltage offsets that are corrected by the amplifier when the current is zeroed (i.e., in voltage clamp mode). The offsets consist of liquid junction potentials and potential differences

between solid electrodes and the solutions they are in contact with (the “electrode” or “half-cell” potentials).” (Axon Instruments technical note, 2004)

The authors use hardware and software from HEKA. The Patchmaster software from HEKA does include a semi-automatic correction for EJP. However, the user MUST enter a value for the LJ in order for the correction to be applied.

These errors in membrane potential that arise from EJPs can be 10s of mV and must be properly corrected in order to compare measured reversal potentials across recordings performed with distinct combinations of intracellular and extracellular solutions.

If the authors used the correction feature of the Patchmaster software, it is imperative that they report values of the LJ corrections that were used.

If they did not use this feature and only zeroed voltage offsets prior to obtaining tight seals, the LJ is a systematic error in membrane voltage that permeates all electrophysiology measurements here. And are particularly important for measurements of reversal potential. Fortunately, this voltage error can be corrected posthoc. This will involve determining LJ values (either experimentally or estimating using a calculator such as this one <https://swharden.com/LJPcalc/> or other tools) and then applying the correction to the membrane potential.

Since the determination of ion selectivity by varying solutions is central to the study, this issue must be addressed properly prior to publication. It is possible that after correcting for LJ, the measured changes in reversal potential better match the values predicted from the GHK equation.

2. Uncertainty in the amplitude of mechanical indentations delivered to dissociated and cultured ASJ neurons
Thank you for revising the figure illustrating mechanical stimulation of ASJ neurons in culture and for providing additional information about the stimulation of these tiny cells. This reviewer is acutely aware of the technical challenge represented here. Still, the authors’ response raises new concerns.

It seems that they used a stimulus that moved the actuator 4µm but that the thin probe bent and likely generated a smaller indentation. The authors note that using a thicker probe destroyed recordings and or moved the cell, which is understandable and sensible. The authors should find a way to communicate to readers in the results section that the actual indentation is less than 4µm and likely to be of a similar amplitude during each recording (but not across recordings). One idea might be to relabel the stimulus trace in Fig 5c as “stimulator movement” or something similar.

Formatting and citations suggestions (minor)

1. (line 5) epithelial Na channels (ENaC) are part of a large superfamily, consider expanding to include other subfamilies by replacing "epithelia Na+ channels (ENaC)" with "degenerin, epithelial, and acid-sensing Na+ channels (DEG/ENAC/ASIC)"
2. (ine 52) The review articles cited do not cover OSCA/TMEM63, consider adding Goodman, Vasquez, & Haswell J Gen Physiol 2023 that does cover OSCA/TMEM63
3. (line 308-309) While this study was in review, another paper was published reporting the expression of CALM-1 in *C. elegans*, please compare findings.

Version 2:

Reviewer comments:

Reviewer #1

(Remarks to the Author)

1. Abstract and lines 35-37. Please change ANOH-1/ANO1 to ANOH-1, as the authors are suggesting physiological functions but mammalian ANO1 is not present and therefore not relevant in the *C. elegans* system.
2. Abstract and line 38. Please remove the description of TMC1; it is not relevant here.
3. Please remove TMEM16A in the Keywords; this study is about ANOH-1.
4. Lines 341-355. While there seems to be a minimal extent of colocalization between ANOH-1 or hANO1 with the supposed membrane marker CD8 in the neurons, none of these fluorescence signals provide support for membrane localization, including those originated from CD8 (Fig. S8f and g).
 - a. Do not falsely claim that these proteins are properly localized to the membrane. Please revise.
 - b. Do not extrapolate that the completely cytosolic fluorescence of ANOH-1 might be due to the lack of putative auxiliary molecules, it simply didn't express in the expected form in HEK293T cells. Please revise.
5. Please address the following issues with the co-IP experiments. If the authors cannot provide valid experimental evidence for the proposed interaction between ANOH-1 and hANO1 with CALM-1-3 and hCIB2-3 respectively, the authors should retract the claim that ANOH-1 functions in a channel complex and that an analogous assembly might exist in mammalian

systems.

- a. According to the UniProt database, the canonical isoform of human TMEM16A is 986 aa in length (<https://www.uniprot.org/uniprotkb/Q5XXA6/entry#sequences>), which corresponds to ~110 kDa. Please correct the corresponding errors in the text.
 - b. Compared to ANOH-1 (840 aa, ~90 kDa), this is a ~20-kDa difference for a monomer and ~40-kDa different for a dimer. It is not possible that they run at the same molecular weight as the authors have suggested. The reason for this is apparent when one inspects the full gel (rebuttal letter) – the authors are simply showing the bands that didn't enter the gel and erroneously assigned the molecular weight (Fig. 7f, right; see also points c and d).
 - c. The full gel that the authors show in their rebuttal letter corresponds to right panel in Fig. 7F. This gel shows that in fact most if not all of hANO1-HA has aggregated and did not enter the gel. The same issue, albeit to a lesser extent, is observed for ANOH-1. This might not be unexpected as the authors have not validated that the detergent that they used can solubilize ANOH-1 and hANO1 in a well-folded, monodisperse state.
 - d. The labelling of the molecular weight in the cropped blots shown in Fig. 7F, right is erroneous. In this panel, the authors indicate that the bands are above 180 kDa. However, this is not reflected in the full gel, where the bands shown in Fig. 7F (right) seem to correspond to the bands that did not even enter the gel (above 310 kDa in the full gel shown in the rebuttal letter).
 - e. Given the aggregation of ANOH-1 and hANO1, and that ANOH-1 simply didn't express in the expected form in HEK293T (Fig. S8E), the validity of the co-IP experiments performed under the described experimental conditions is questionable.
 - f. The authors did not perform the correct control experiment. What the authors need to show is that the co-IP of ANOH-1 and ANO1 depends on the presence of CALM-1-3 and hCIB2-3 respectively. The current co-IP data do not show that at all – these gels merely show that the authors detected bands containing the HA fusion tag and not in the negative control where the fusion tag is not present.
 - g. Given the problems associated with recombinant expression of ANOH-1, the authors should demonstrate the proposed interaction between ANOH-1 and CALM-1-3 in the native *C. elegans* system and that the co-IP of ANOH-1 depends on the presence of CALM-1-3.
6. Lines 456-457. 'ANOH-1', not anoctamin-1; and plays a crucial role in mechanosensation 'in *C. elegans*'. Both are specific to the system the authors have investigated. Please revise.

Reviewer #2

(Remarks to the Author)

This manuscript provides evidence in support of the idea that anoctamin-1 is a mechanosensitive chloride channel. The evidence includes genetic dissection of behavioral responses to mechanical stimulation, in vivo electrical recording from the relevant neurons, and ectopic expression conferring mechanosensitivity on other neurons. Although to reconstitute this activity in heterologous cells were not successful, these findings bolster, complement, and enhance prior studies of this ion channel.

This revision addresses technical concerns raised by both reviewers, especially with respect to corrections for liquid junction potentials. These revisions improve the rigor of the study and will be a good example for readers and others in the field for how to properly handle systematic errors in cellular electrophysiology.

The author's response to queries about voltage errors due to uncompensated series resistance could be improved by paying closer attention to the precision of the terms used. Specifically, might the authors consider replacing "The voltage drop caused by R_s is negligible compared to the applied voltage and thus does not significantly affect the clamping." With "The voltage error due to uncompensated series resistance is less than 2mV, on average, and was not corrected." Strictly speaking R_s does not "cause" a voltage drop, but rather a voltage drop occurs when the measured membrane current flows across R_s . This represents an error in the voltage equal to $-I_m \cdot R_s$.

Reviewer #3

(Remarks to the Author)

Version 3:

Reviewer comments:

Reviewer #1

(Remarks to the Author)

The authors have addressed the major concerns that I raised.

We sincerely appreciate the reviewers' constructive feedback. We have addressed all the raised points and submitted a revised version that includes the requested experiments. The text and figures have been updated accordingly. We are pleased to report that the experiments not only validate the interpretations presented in our original submission but also enhance them with substantial additional experimental results and deeper mechanistic insights.

Below is our point-by-point response to the reviewers' comments:

Reviewer #1

1. For ion selectivity experiments, the change from NaCl to Na-Glu or to NMDG-Cl would result in a large junction potential given the difference in ion mobility of the substituted ions. The authors should describe in more detail how junction potential was minimized during the exchange of solutions and how this was corrected for.

Response: For the ion selectivity experiments, we did not record the responses of the same cell under different solution conditions by switching perfused bath solutions. Instead, we recorded the responses of different cells under various intrapipette and bath solution conditions. Upon the pipette's entry into the bath solution, we compensated for the liquid junction potentials using the automatic compensation of the EPC-10 amplifier along with the Patchmaster software. Consequently, the impact of liquid junction potentials on the experimental results was minimized. We have made the necessary revisions to clarify this point in our revised manuscript.

2. The authors should include the I-V plots for the control condition for all ion selectivity experiments (Fig. 4f, 5d, 6c). More specifically, the authors should compare a) CsCl (IS)/NaCl (ES) with Cs-Glu (IS)/NaCl (ES), b) CsCl (IS)/NaCl (ES) with CsCl (IS)/Na-Glu (ES), and c) CsCl (IS)/NaCl (ES) with CsCl (IS)/NMDG-Cl (ES).

Response: Good suggestion. We have conducted the recommended electrophysiological recordings and improved the I-V plots in the revised manuscript (Fig. 4f, 5d, 6c).

3. Do the reversal potentials match the Nernst potential of Cl under the extracellularly and intracellularly substituted conditions? Please discuss.

Response: Based on the Nernst equation, under our experimental conditions at 20°C:

For CsCl (IS, 158 mM Cl⁻)/NaCl (ES, 159.5 mM Cl⁻), the reversal potential of chloride channel currents is approximately 0.2 mV.

For Cs-Glu (IS, 13 mM Cl⁻)/NaCl (ES, 159.5 mM Cl⁻), the reversal potential is approximately -64.5 mV.

For CsCl (IS, 158 mM Cl⁻)/Na-Glu (ES, 14.5 mM Cl⁻), the reversal potential is approximately 61.3 mV.

For Cs-Glu (IS, 13 mM Cl⁻)/Na-Glu (ES, 14.5 mM Cl⁻), the reversal potential is approximately -2.8 mV.

Our data indicate that the removal of extracellular Cl⁻ shifted the reversal potential of the MRCs to positive voltages, while the removal of intracellular Cl⁻ shifted the reversal potential to negative voltages. These observations align well with the predicted Nernst potentials for Cl⁻ under the conditions where extracellular and intracellular Cl⁻ were substituted.

We observed that the measured reversal potentials did not exactly match the theoretical values. However, there are at least two reasons that can explain this discrepancy. Firstly, the sensory cilia of neurons such as ASJ, ASK, and ASI are located at the tip of the worm's nose (where the mechanosensitive channels were activated by our touch probe), whereas our electrode recordings are made at the cell bodies of these neurons, which are about 100 micrometers apart. We typically wait for 1 minute between forming the whole-cell patch clamp and recording the MRCs. However, it seems unlikely that the intracellular fluid in the cilium at the tip of the nose can be completely replaced by intrapipette solution through the thin dendrites within this time. Additionally, we are not sure if the interstitial fluid around the cilia of these neurons can be completely replaced by our bath solution. Secondly, we observed residual MRCs in ASJ neurons in the *anoh-1* mutant. This ANO1-independent current may also contribute to the reversal potentials we measured.

We have revised the text to ensure clarity and accuracy on this point.

4. Please provide in the methods the typical pipette resistance, seal resistance, and series resistance, and whether the pipettes were fire-polished.

Response: Following the reviewer's suggestion, we have included details on pipette resistance, seal resistance, and series resistance in the methods section. For patch-clamp recordings of HEK293T cells, the pipette resistance is about 3-5 MΩ, the seal resistance is greater than 1 GΩ, and the series resistance is around 10-15 MΩ. In contrast, for patch-clamp recordings of worm neurons, the pipette resistance is approximately 15-20 MΩ, the seal resistance exceeds 1 GΩ, and the series resistance is around 50-70 MΩ. Additionally, the pipettes were not fire-polished. Given the small size of worm neurons, typically with diameters of 3-5 μm, we cannot use pipettes with openings large enough to achieve a pipette resistance lower than 10 MΩ. Consequently, the series resistance is significantly higher than that in recordings conducted with mouse neurons or cultured HEK293T cells.

5. The authors should indicate the zero current level in all panels where current traces are shown.

Response: Nice suggestion. We have added the information to the figures in the revised manuscript.

6. The authors have illustrated in Fig. 6f the putative location of K588 in ANOH-

1. According to the provided sequence alignment in Fig. 6d, this corresponds to K645 in mouse TMEM16A, which is in the transmembrane region. Mutation of this residue in mouse TMEM16A does not abolish current activity, but profoundly changes the ion conduction properties that generally lowers the conductance.

a. The authors should characterise the electrophysiological properties of ANOH-1 in a heterologous system.

b. The authors should examine the effect of K588 mutations in direct comparison with the wild-type ANOH-1 in the same system.

c. The authors should revise the diagram to depict the correct location of this residue; the equivalent residue in TMEM16A is not in the extracellular loop, it is on the intracellular half of helix 6.

Response: To characterize the electrophysiological properties of ANOH-1 in a heterologous system, we attempted to express ANOH-1 in HEK293T cells. However, we discovered that nematode ANOH-1 does not traffic to the cell membrane in these cells. Consequently, we were unable to directly compare the effects of K588 mutations with wild-type ANOH-1 in this system.

Our results show defects in touch-evoked currents, calcium increases, and nose touch behavior in *anoh-1* mutant worms. As shown in the revised manuscript, these defects were rescued by expressing either nematode ANOH-1 or human ANO1. However, expressing either the nematode ANOH-1 with the K588A mutation or the human ANO-1 with the K645A mutation did not rescue the defects.

Thank you for the correction; we have updated the diagram to accurately depict the location of the K588A mutation.

7. Given that the structure-function relationship of mouse/human TMEM16A is much better characterised and that this homolog can also rescue the mechanically activated responses, the authors should include mutations that severely perturb ion conduction, such as K588A and K645A in mouse TMEM16A (numbering for the ac variant), to show that these mutations can inhibit the rescue.

Response: This is a really interesting experiment and we used human TMEM16A with the K645A mutation to rescue the ANOH-1 function deficit in mechanosensation. Our results showed that human TMEM16A with the K645A mutation was unable to restore the nose touch behavioral defects and the touch-evoked responses in CEM and ASJ neurons in the *anoh-1* mutant background. We have incorporated new data in our revised manuscript.

8. The authors should provide a UniProt identifier for the described TMEM16 homolog in *C. elegans*.

Response: The UniProt identifier of the TMEM16 homolog in *C. elegans* is UP000001940. We have added the information to the revised manuscript.

9. In the co-IP results shown in Fig. 7f, what is the molecular weight of ANOH-1 as it appears to run exactly the same as human TMEM16A.

Response: According to the UniProt database, the molecular weight of nematode ANOH-1 with an HA-tag is 99 kD, which is similar to that of human TMEM16A with an HA-tag, at 82 kD.

10. The molecular weight where the human TMEM16A runs does not seem to correspond to its monomeric size of around 110 kDa, which typically runs in its monomeric form in SDS-PAGE under reducing conditions. If this is due to glycosylation, please provide data on a PNGase-digested sample, or otherwise provide explanation.

11. The authors should provide data on co-IP experiments using IP: FLAG and WB: HA.

Response: Thank you for your insightful comment. We have addressed your concerns through a series of experiments and adjustments: (1) SDS-PAGE Conditions: We used SDS-PAGE with HA detection at 8% and FLAG detection at 12%. The sample buffer included β -mercaptoethanol, ensuring reducing conditions. We changed Triton in the cell lysis buffer from 0.5% to 1%, and observed identical results in both cases. (2) Sample Preparation: Before running the Western blot, samples were heated at 95°C for 10 minutes, or at 65°C for 10 minutes, and we observed no significant difference in results, suggesting that our observed band size discrepancy is not due to heating-induced aggregation. (3) To investigate the potential impact of glycosylation, we treated the samples with PNGase F under both denaturing and non-denaturing conditions. In the denatured samples, the main band weakened, but no new stronger bands appeared, possibly indicating protein degradation post-deglycosylation. (4) Under reducing conditions (sample buffer with β -mercaptoethanol to break disulfide bonds and depolymerize oligomers), we still did not observe the monomeric band.

Sample Heating: We tested the recommendation that membrane proteins should not be boiled before running WB to avoid protein aggregation. Using 65°C for 10 minutes yielded similar results to boiling for 10 minutes. Given these observations, we suspect that the larger-than-expected molecular weight bands could be due to oligomer formation.

Following your suggestion, we newly conducted co-IP experiments using IP: FLAG and WB: HA. The results supported the direct binding of nematode ANOH-1 with nematode CALM-1, as well as human ANO1 with human CIB2.

In our revised manuscript, we have incorporated new data and provided a comprehensive explanation and discussion to ensure clarity and accuracy on this subject.

12. The authors should demonstrate in a heterologous system the co-expression of CIB2/Ano1 and CALM1/ANOH-1 gives rise to mechanosensitive ionic currents

and that their ion selectivity would resemble that observed in the native systems shown in the manuscript. The authors should also include these data if the co-expression of these constructs in the heterologous system does not result in mechanosensitive currents.

13. The authors should investigate in excised patches whether CIB2/Ano1 and CALM1/ANOH-1 are intrinsically mechanosensitive, should this give rise to mechanosensitive currents.

Response: We acknowledge the reviewer's comment. According to your suggestion, we transgenically co-expressed CIB2 with ANO1 or CALM1 with ANOH-1 in PIEZO1-KO HEK293T cells. However, we did not observe mechanosensitive currents using either whole-cell recording or inside-out single-channel recording. These results indicate that ANO1-CIB2 may require additional auxiliary molecules to sense touch. This situation is very similar to the issue underlying the TMC1-CIB2 mechanotransduction complex, as no studies have shown that TMC1 functions as a mechano-gated channel in a heterologous system.

We also attempted excised patch recordings in both in vivo and cultured nematode ASJ neurons. However, due to the small size of the ASJ soma (2-3 μm) and the tiny opening of the recording pipette (less than 1 μm), we were unable to successfully obtain excised patch recordings.

While direct observation of the mechanosensitive currents of CIB2/ANO1 and CALM1/ANOH-1 in the heterologous system and excised patches was not achieved, the mechanosensitivity of CIB2/ANO1 and CALM1/ANOH-1 is supported by a range of evidence. This includes electrophysiological, calcium imaging, behavioral, and genetic analyses, as well as ectopic expression of these molecules in mechanosensitive neurons including ASJ, ASK, and ASI. However, further investigations are necessary to unravel the intricate mechanisms underlying the mechanosensitivity of CIB2/ANO1 and CALM1/ANOH-1.

In our revised manuscript, we have incorporated new data and provided a comprehensive explanation and discussion to ensure clarity and accuracy on this subject.

14. The authors should provide data on CALM1 expression in all the neuronal cell types investigated and that CALM1 is not expressed in ASH neurons where ANOH-1 does not function as a mechanoreceptor channel (Fig. S1); these are an absolute requirement for the proposal that CALM1/ANOH-1 forms a mechanoreceptor complex.

Response: We fully agree with the reviewer. Previous studies from our group and others have indicated that CALM-1 is expressed in ASK neurons, but not in ASH neurons. To investigate this further, we generated a transgenic line expressing the fluorescent reporter protein mCherry driven by a 3 kb *calm-1* promoter. We observed that CALM-1 is expressed in ASI and ASK neurons. However, no CALM-1 expression was detected in ASH, ASJ, or CEM neurons

using our *Pcalm-1::GFP* transcriptional reporter line. To rule out the possibility of an incomplete expression pattern due to the selected *calm-1* promoter element, we inserted an in-frame mNeonGreen cassette into the endogenous *calm-1* locus using the CRISPR/Cas9 knock-in method. Although the mNeonGreen signal was relatively weak in our *calm-1::mNeonGreen* knock-in line, anti-mNeonGreen staining revealed punctate distribution of *CALM-1::mNeonGreen*. Our staining results suggest that CALM-1 is expressed in ASJ and CEM neurons but not in ASH neurons. We have made the necessary revisions.

15. Although the UNC-44-null background does not display mechanoreceptor currents, there is no direct evidence in this manuscript that CALM1/ANOH-1 would be tethered to UNC-44 and that this tethering is responsible for mechanotransduction. The authors should indicate that this is a speculation or a hypothesis in both the schematic and in the legend in cases where no direct evidence is presented and provide supporting references where needed.

Response: We thank the reviewer for raising this issue. While the UNC-44-null background does not display mechanoreceptor currents in this study, and previous studies have implicated that CALM1/TMC-1 is tethered to UNC-44 and that this tethering is responsible for the mechanotransduction of TMC1 mechanosensitivity in both nematode OLQ neurons and mouse inner ear hair cells, we acknowledge that there is no direct evidence provided in this manuscript. We have made the necessary revisions to clarify this point in our revised manuscript.

16. Without calcium, the current passing through TMEM16A is highly outwardly rectifying due to the highly negative electrostatic potential at the vacant calcium binding site and in the inner pore, which effectively gates ion conduction. This has a consequence that any inward current is minimal even at 150 mM intracellular Cl and hence a minimal capacity to give rise to a depolarising current in the absence of bound calcium under resting cellular conditions. The authors should acknowledge this discrepancy and how this would affect their hypothesis.

Response: Using an ectopic expression system, we expressed human TMEM16A in mechano-insensitive ASK neurons and observed touch-evoked currents. According to the I-V plots, the touch-evoked currents in ASK neurons expressing human TMEM16A show weak outward rectification, similar to the nematode ANOH-1 mediated touch currents in ASJ, ASI, and ASK neurons. We then transgenically expressed human ANO1/TMEM16A in PIEZO1-KO HEK293T cells and observed that the calcium-activated currents are highly outwardly rectifying. These results suggest that calcium-activated and mechanosensitive activation of TMEM16A may differ in their rectifying properties.

A previous study (Hawon Cho et al., Nature Neuroscience 2012) reported

that ANO1 functions as a heat sensor in mammalian small-diameter nociceptive DRG neurons. Current-clamp recordings of DRG neurons from wild-type mice exhibited robust TMEM16A-dependent depolarization in response to a heat ramp from the resting membrane potential, with the pipette solution set at 30 mM Cl⁻ (the measured Cl⁻ concentration in DRG neurons) and the bath solution at 140 mM NaCl. Additionally, heat-induced inward currents in cultured DRG neurons were still observed after chelating intracellular Ca²⁺ with 10 mM BAPTA in the pipette solution. These findings demonstrate the capacity of TMEM16A to generate a depolarizing current in the absence of bound calcium under resting cellular conditions.

We have incorporated our new data into the revised manuscript and made the necessary revisions.

17. There are typos throughout in the manuscript, please revise.

Response: We appreciate the reviewer's attention to detail. We have thoroughly revised the manuscript to correct any typo errors.

Reviewer #2 (Remarks to the Author):

1. Chloride channels that depolarize neurons

The authors correctly note that activation of Cl⁻ channel can depolarize a neuron under some conditions and hyperpolarize a neuron under others. This is central to their clever experimental design of using ectopically expressed His-Cl to show that activating a Cl⁻ channel increases intracellular calcium (likely through depolarization). However, their discussion of how this works is overly simplified and somewhat misleading. I am sympathetic that this concept can be difficult to explain to a general audience. Still, I urge the authors to write in a manner that is both precise and accessible.

The effect of activating a Cl⁻ channel on membrane potential depends on the chloride gradient across the plasma membrane or the equilibrium potential for Cl⁻ (E-Cl), not only the intracellular or extracellular concentrations of Cl⁻ ions. It also depends on the resting membrane potential, V_{rest}. If V_{rest} is below E-Cl, then activating a Cl⁻ channel will **DEPOLARIZE** the membrane. Conversely, if V_{rest} is above E-Cl, then activating a Cl⁻ channel will **HYPERPOLARIZE** the membrane. The finding that activating ectopically expressed His-Cl in CEM and ASJ increases intracellular calcium suggests (but does not prove) that V_{rest} is below E-Cl. This finding also has implications for the widespread use of His-Cl to **INACTIVATE** neurons in *C. elegans* and suggests that activation of His-Cl may have different effects in other neurons. Consider commenting on the implications of this observation in the Discussion and also citing Pokala N, et al. PNAS 2014 111(7):2770-5. doi: 10.1073/pnas.1400615111 who claim that His-Cl silences *C. elegans* neurons.

Response: We acknowledge the reviewer's comment. In our revised

manuscript, we have provided a comprehensive explanation and discussion to ensure clarity and accuracy on this subject. Additionally, we have cited the suggested literature.

2. Imprecise text related to point #1

- “The opening of a chloride channel can either excite or inhibit cells, depending on the concentration of intracellular chloride ions.”

The first clause of this sentence is correct, but the second lacks precision. The concentration of intracellular Cl⁻ is only one factor. Whether or not a Cl⁻ channel excites or inhibits a cell depends on the difference between membrane potential and the ratio of intracellular to extracellular chloride.

- “Because the extracellular chloride level is typically higher than the intracellular, the electrochemical gradient will drive chloride influx and hyperpolarize the plasma membrane.”

Cl⁻ influx will hyperpolarize the plasma membrane if, and only if, the resting potential is more positive than E-Cl⁻.

Response: We appreciate the reviewer's comment, and we have revised our manuscript accordingly to provide greater clarity on this point.

3. The role of *kcc-3* and referencing the literature

It is possible that *kcc-3* mutants have smaller mechanically-evoked Ca²⁺ signals in CEM neurons due to changes in the Cl⁻ gradient across the plasma membrane, as implied by the authors. However, other explanations are possible, such as a change in membrane potential or morphological defects in cilia shape (see Singhvi et al, 2016). If the author's inference is correct, then loss of *kcc-3* should also reduce HisCl calcium responses in CEM. The authors should perform these additional experiments to strengthen these inferences.

It would also be helpful to determine where *kcc-3* and *abts-3* are expressed and function in this context. Investigating the AFD neurons, Singhvi et al, 2016 have shown that *kcc-3* is expressed in the amphid sheath cell, a glial cell that encapsulates amphid sensory neurons (including ASJ) in hermaphrodites.

Response: We appreciate the reviewer's comment. We observed that the *abts-3* mutation abolished the HisCl calcium responses in CEM, whereas the loss of KCC-3 did not affect these responses. These results suggest that the loss of KCC-3 does not affect nose-touch evoked calcium responses in CEM by modulating intracellular chloride concentration. We have incorporated these new data into our revised manuscript and made the necessary revisions.

4. Insufficient support exists to support the claim that MRCs are Cl⁻ selective

The authors show that the reversal potential for MRCs is sensitive to changes in E-Cl⁻, indicating that the channels are Cl⁻ permeable. This is necessary, but not sufficient evidence that the channels are SELECTIVE for Cl⁻. Additional experiments would be needed to support this claim, such as measurements of relative permeability. In my opinion, these experiments would be nice to have,

but are not essential. Without additional data, however, the authors should revise the text to match the strength of their finding. In other words, the data support the conclusion that MRCs are carried primarily by Cl⁻ ions but not the conclusion that the channels carrying this current are Cl⁻ selective.

Response: We agree with the reviewer, and have revised our manuscript accordingly.

5. More information is needed about the CEM neurons in males.

Please inform the reader that CEM neurons were previously considered to be chemosensory and not mechanosensory. The authors findings suggest that they are multimodal, which is interesting and should be highlighted. Finally, no data are presented confirming that ANOH-1 is expressed in CEM. It seems reasonable to guess this is the case, but direct experimental evidence would be better.

Response: We generated *anoh-1::mNeonGreen* knock-in line, and anti-mNeonGreen staining revealed punctate distribution of ANOH-1::mNeonGreen in CEM neurons. We have added these new data into our revised manuscript, and revised our manuscript as suggested to provide greater clarity.

6. More discussion is needed about the results in isolated ASJ neurons

This concern has technical and conceptual elements. Technically, it is hard to imagine how the experimenters obtained recordings from a cell body that is 3-5 μm in diameter and delivered a 4 μm indentation while maintaining a high-quality recording. What size is the recording pipette tip and glass stimulator that is used for these experiments? The schematic shown in Figure 5a is misleading, as it implies that the cell body is much bigger than the recording pipette or the stimulator. This is simply not possible given the size of the ASJ neuron and dimensions of recording pipettes and stimulators. Conceptually, more discussion is needed about the implications that ASJ retains mechanosensitivity when isolated.

Response: We thank the reviewer for raising this issue. Given the small size of worm neurons, typically with cell body diameters of 3-5 μm , the recording pipette tips are usually about 1 μm in diameter. The touch stimulators were made from borosilicate glass capillaries with an outer diameter of 1.0 mm (WPI, 1B100F-4). These capillaries were heat-pulled using a Sutter electrode puller (P-97). The tips of the capillaries are elongated and have very small openings, around 0.2 μm , which is approximately 1/20 of the diameter of the ASJ cell body. We set the touch pipette to move forward 4 μm to stimulate the cell body; however, the actual displacement might be much shorter due to the bending of the slender touch pipette upon contact with the cell. We replaced the touch pipette for each cell recording to ensure optimal flexibility. A larger tip or reduced flexibility could push the cell out of position, disrupting the seal or even puncturing the cell membrane. Our data demonstrate that this method ensures high-quality recordings. These results clearly indicate that ASJ neurons are

direct mechanoreceptors. Our in vivo experiments found that the MRCs in ASJ neurons have an extremely short delay time (less than 5 ms). Additionally, the MRCs were abolished in *anoh-1* mutants but rescued by restoring nematode *anoh-1* or human *ano1* in ASJ neurons, further supporting this conclusion. We have revised the schematic figure, and provided a comprehensive explanation and discussion to ensure clarity and accuracy in the revised manuscript.

7. The data in this paper do not support the conclusion that ANOH-1 activation in *C. elegans* neurons depends on a gating spring (Discussion).

The concept of a gating spring refers to a specific biophysical mechanism for converting cell deformation into channel activation. Neither these authors nor others have demonstrated that mechanosensitivity of ANOH-1 or any other anoctamin depends on a gating spring. The finding that ANOH-1 activity requires CALM-1 expression is not sufficient to draw this inference. This experimental finding is sufficient to support a weaker claim, however — namely that CALM-1 and ANOH-1 interact with another.

Response: We totally agree with the reviewer and have revised our manuscript accordingly.

Composition

1. The Results section is divided into many short subsections that make it hard to read. If the authors were to consolidate short (1 paragraph) subsections in the Results into larger chunks, this would be helpful to readers. In particular Figure 1 is discussed in the first three sections, which could be a single section.

2. Reordering the Results Section for better flow.

In the Results subsection “ANOH-1 mediates mechanotransduction in ASJ neurons”, the first paragraph begins with a switch of focus from CEM to ASJ neurons, which feels a bit sudden and lacks rationale; only mentioning the experimental difficulty in males vs hermaphrodites is not enough (Line 170-171). The above issue can be avoided by reordering and consolidating the short subsections (as mentioned in 1). For example, sections “ANOH-1 is required for the mechanosensitivity of CEM neurons” and “ANOH-1 mediates mechanotransduction in ASJ neurons” can be consolidated. Moreover, ANOH-1’s role in mediating touch-avoidance behavior can be discussed in a single section by consolidating ANOH-1’s separate roles in CEM and ASJ neurons, respectively. The dependence of ANOH-1 on chloride ions can be discussed afterwards.

The central idea of reordering and consolidating subsections is to treat ANOH-1 as the main character in the story and use it to conceptually determine subsections (namely, a. ANOH-1’s role in neuronal responses (CEM, ASJ); b. ANOH-1’s role in animal behavior in the context of neurons; c. ANOH-1’s physiological, molecular, genetic mechanisms).

Response: We agree with the reviewer and have changed the text accordingly.

3. Figure legends should be focused on the data presented in each figure. Throughout the manuscript, figure legends contain both interpretation and description of methods that does not belong.

Response: We appreciate the reviewer's comment. We have revised our manuscript accordingly.

4. More context is needed for readers to understand the role of using *osm-9* mutants in behavioral assays (Fig 2b).

Response: We have revised our manuscript accordingly.

Technical issues

1. Please inform the reader how liquid junction potentials determined. Were corrections applied for conditions in which Cl^- was replaced intracellularly and extracellularly? Such junction potentials can be significant and if not properly corrected, they might compromise measurements of reversal potential.

Response: For the ion selectivity experiments, we did not measure the responses of the same cell under different solution conditions by switching the perfused bath solutions. Instead, we recorded the responses from different cells, each subjected to distinct intrapipette and bath solution conditions. Upon the pipette's entry into the bath solution, we used EPC-10 amplifier with PatchMaster software to compensate for the liquid junction potentials between the intracellular and extracellular solutions. This minimized the impact of liquid junction potentials on our experimental results. We have revised our manuscript accordingly to provide greater clarity.

2. Please disclose the range of series resistance values and the fraction of the series resistance that was not compensated. This can affect I-V curves, as the actual membrane voltage will depart from the command voltage by an amount equal to the current flowing multiplied by the uncompensated series resistance. If the latter is high, then the voltage error can be substantial.

Response: Given the small size of worm neurons, typically with diameters of 3-5 μm , we can only use pipettes with small openings for whole-cell recordings. The pipette tips are about 1 μm in diameter, with pipette resistances approximately 15-20 $\text{M}\Omega$ and series resistances around 50~70 $\text{M}\Omega$. Since the amplitude of the MRCs is small, approximately 20 pA when the holding potential is -70 mV, the voltage error is minimal (20 pA * 50 $\text{M}\Omega$ = 1 mV). We have revised our manuscript accordingly to provide greater clarity.

3. Please tell the reader if the experimenters performing experiments (especially behavior, which is scored manually) were aware of the genotypes under test of if these experiments were performed blind to genotype.

Response: All the behavior experiments were performed blind to genotype. For the nose touch behavior test, we used *osm-9* mutant worms as a control. We have revised our manuscript accordingly.

4. The sample sizes used in Figures are indicated by n, but the sample's identity is not clear. For example, in Fig 1a, n=35, is this the number of trials or animals? The authors need to specify that in the figure captions at least once on the first appearance.

Response: In Fig. 1a, (n=35) represents the number of animals used in the study. We recorded and calculated the calcium response or MRCs from a single CEM neuron in each worm. We have revised our manuscript in the Methods section to provide greater clarity.

5. At Line 126, the description of dissecting worm's cuticle is also mentioned in Fig 1b caption, which is redundant. Consider moving the description in Fig 1b caption.

Response: We have revised our manuscript as suggested.

6. Figs 3a and 4a show schematic or fluorescence images of ASJ anatomy; this is good. Can the authors also show CEM anatomy in Fig 1a using either schematics or fluorescence images?

Response: We have revised our manuscript as suggested.

7. In Lines 188-189 and Fig 4c, how was the latency time is calculated and indicated? Authors should specify it more clearly in the manuscript.

Response: For recording of MRCs, a glass stimulus probe was used together with a Piezo actuator (PI) mounted on a micromanipulator and triggered by the EPC-10 amplifier to deliver mechanical stimulation. The latency time was measured between stimulus delivery and opening of the channel (onset of the MRCs), showing an average value of 3.35 ms for MRCs recorded in ASJ in vivo. This duration includes the time required to move the probe, the transmission of force from the cuticle to the channels, and the time needed to activate the channels. This short latency strongly suggests that force directly acts on the ANOH-1 channels, which mediates MRCs in ASJ neurons. Our manuscript has been revised to provide greater clarity on this point.

8. At the start of several calcium traces the DeltaF/F values are not zero (e.g. Fig. 3c, His-CI trace), suggesting that the reference for the F0 value was taken from a different time in the trace. Please indicate that time and the rationale for choosing it.

Response: The average GCaMP signal from the initial ten seconds prior to stimulation (either touch stimulation or histamine perfusion) was designated as F₀. Consequently, for all calcium traces, the $\Delta F/F$ values are zero at the onset of stimulation. We apologize for the mistake and have corrected the zero values in the figures.

9. Calcium traces in Figure 1 display different time frames. Control responses in

Figure 1a reach a peak in about 30s after stimulation. The time course in Figure 1b is much slower. Why?

Response: Given that the cilia of CEM neurons are embedded in the cuticle, we dissected a section of the cuticle from the worm's head to improve the efficacy of ion exchange or drug delivery. After dissection, the worm was immersed in either a Na⁺-free or Cl⁻-free bath solution, or a normal bath solution with NFA for 20 minutes before calcium imaging. The calcium responses of these dissected worms were weaker and slower compared to the intact ones (Fig 1b, c), possibly due to the impact of the dissection on the worm's overall health.

10. Niflumic acid is known as a chloride channel blocker, but it is not specific for this target. Please revise the text accordingly.

Response: Thank you for the information. We have revised our manuscript accordingly.

11. Loss of *anoh-1* does not completely eliminate MRCs in ASJ. And, the kinetics of the residual current are faster than that shown in wild-type. Please comment on the residual MRC.

Response: This is indeed an intriguing question. It's possible that the residual MRCs in ASJ neurons of *anoh-1* mutants are mediated by other mechanogated channel(s) with faster kinetics than the ANOH-1-dependent MRCs. However, given that the remaining responses are too small to allow for effective screening, identifying the channel(s) based on *anoh-1* mutants presents a significant challenge. As a result, we have revised our manuscript accordingly.

12. Does the mutation described as a pore-dead mutation affect reversal potential or selectivity? If so, this would provide additional evidence that ANOH-1 carries the MRCs.

Response: Follow your suggestion, we have tried to record the reversal potential which pore-dead mutation affected. However, the pore-dead mutation carried MRCs were small, thus we can't calculate the reversal potential accurately.

13. Please move the move ASH data to main paper from supplement.

Response: We appreciate the reviewer's comment. We have revised our manuscript accordingly.

We highly appreciate the reviewers for their valuable comments. Below are our point-to-point responses:

Reviewer #1

The authors should provide the line numbers, make references to the changes made to the figures, and include a quotation of the specific changes made to the text in all the individual responses.

Response: We appreciate the reviewer's valuable feedback and have incorporated the requested revisions. All changes in the revised manuscript are clearly highlighted in red for easy identification.

1. Title and Line 30. Please introduce ANOH-1 as the *C. elegans* Anoctamin-1 in both the title and abstract as Anoctamin-1 is often referred to as the mammalian TMEM16A, especially given that the sequence homology between these two homologs is only ~30%. Please also state in the title that this is specific to worms in order to avoid false generalization.

2. Title and Line 31. Please describe ANOH-1 as an essential component of a mechanosensory channel complex in *C. elegans*. The current description suggests that the channel is itself gated by mechanical force, for which there is no direct evidence in the presented data.

3. Line 39. Please state instead 'anion channels in mechanosensory transduction in metazoans'. The authors should refrain from once again giving the impression that ANOH-1 or ANO1 are direct sensors of mechanical force.

4. Line 96-107. The description that Anoctamin-1 chloride channel can be directly activated by forces is misleading, as is the description that Anoctamin-1 chloride channel possess the capability to respond to mechanical stimulation. The same goes for the description that Anoctamin-1 serves as a native mechanosensor. The data suggest that this channel might be required for the observed mechanosensitive currents but do not provide any direct evidence for these statements. Please revise and only make statements in accord to the level of support that the current data offer.

Response: Thank you for the valuable feedback. We have revised the title to: "Anoctamin-1 is a core component of a mechanosensory anion channel complex in *C. elegans*" Additionally, throughout the manuscript, we have adhered to the suggestions by avoiding references to ANOH-1 being a mechanically gated channel or directly activated by mechanical force. We have also ensured that the findings are primarily confined to *C. elegans*, as recommended. We have marked all the revised sections in the manuscript using red font.

5. Line 328-331. The molecular weight of ANOH-1 is ~82 kDa and human ANO1 is ~99 kDa.

Response: According to the UniProt database, the molecular weight of nematode ANOH-1 (840 aa) with an HA-tag is ~99 kDa, which is comparable to that of human TMEM16A (695 aa) with an HA-tag, at ~82 kDa.

6. Fig. S8 and ectopic expression control. Both the ANOH-1 and ANO1 constructs appear to give rise to diffuse cytosolic fluorescence when expressed in a recombinant system (Fig. S8), which are properties contradicting to that of integral membrane proteins. This raises the question whether the ectopic expression of these constructs leads to correctly folded proteins that are localized to the plasma membrane in the neurons investigated in this study. Given this observation, the authors should ensure that the ectopic expression of both ANOH-1 and ANO1 results in 1) the full-length protein and 2) proper localization to the plasma membrane in the neurons investigated. Please provide these controls.

Response: Thank you for your insightful comments. We conducted additional experiments using C-terminally tagged EGFP constructs for both proteins, and the results were consistent with those obtained using N-terminal tags. Specifically, EGFP-tagged hANO1 displayed prominent cytosolic fluorescence with weak membrane localization, which aligns with the small calcium-activated chloride currents observed in our electrophysiology experiments. In contrast, EGFP-tagged ANOH-1 did not localize to the membrane in HEK-P1KO cells (Extended Data Fig. 8e). These results suggest that ANOH-1/ANO1 may require auxiliary molecules or specific conditions for proper membrane expression in this system.

To further address concerns regarding the correct folding and membrane localization of these proteins, we also performed experiments in *C. elegans* ASJ neurons. Overexpressed either nematode ANOH-1 or hANO1 was found to co-localize with the membrane marker mCD8, confirming that both proteins are properly expressed and localized to the plasma membrane in the neurons we studied (Extended Data Fig. 8f, 8g). These results suggest that, despite the cytosolic fluorescence observed in the recombinant HEK system, both proteins can achieve proper membrane localization in neurons. The revised manuscript has been updated with these findings.

Unaddressed concerns

1. There is no automatic compensation that can correct for liquid junction potentials (LJP). Junction potentials exist not only at the electrode-solution interface, but importantly also between the bath and pipette solutions. The magnitude of LJP is different for different pairs of bath and pipette solutions and will have to be calculated and corrected for each individual pairs of solutions. Zeroing the pipette offset when the pipette is in the bath solution does not correct for any LJP, as a potential across the electrode has been applied to shift this offset which is in part contributed by the LJP between the bath and pipette solutions. Given the mobility differences between the substituted ions, some of these LJPs can be as large as 10 mV, which is in the range shown in Fig. 4f, 5d, 6c in the revised manuscript. The proper way to correct for LJP is to correct the applied voltage offline according to 'Neher. (1992) Correction for liquid junction potentials in patch clamp experiments. Methods in Enzymology. Volume 207, 1992, Pages 123-131'. Please revise all the voltages involved.

2. The substitution of chloride with gluconate on the intracellular and extracellular side should result in the same or very similar magnitude of reversal potential but with opposite polarity. The fact that this is not the case in the presented data in Fig. 4f, 5d, 6c, in addition to possible incomplete solution exchange in the intracellular compartment, is an indication of a likely voltage offset, especially given that the LJPs in all the recordings were never properly corrected for. Please ensure that all the plotted voltages are properly corrected for any LJPs. Please revise all the voltages involved.

Response: Thank you for your valuable feedback. Upon careful review of our original data, we acknowledge that liquid junction potentials (LJP) were not compensated in our initial analyses. We have now recalculated the LJPs for each pair of pipette and bath solutions used in our experiments using an LJP calculator (<https://swharden.com/LJPcalc/>) and were subsequently applied to posthoc corrections of membrane potential values. After recalculating and correcting the LJPs, we updated the applied voltages offline, and the revised data for Fig. 4f, 5d, 6c, S4b, S5a and S5b now properly account for these corrections. We appreciate your guidance, and have revised the Methods and Figure legends to ensure the accuracy and clarity.

Furthermore, following your second point, we also recognize that the substitution of chloride with gluconate should result in reversal potentials with opposite polarity but similar magnitude. With the corrected voltages, our data now more accurately demonstrate that the reversal potential of ANOH-1/ANO1 is indeed determined by the intracellular and extracellular chloride concentrations, as expected.

3. Please detail how a sufficient voltage-clamp was achieved with series resistance values in the range of 50-70 MΩ as a significant fraction of the applied voltage will drop across the pipette. These series resistances irreversibly distort the I-V plots and the current transients recorded at the same applied voltage. Comparison based on reversal potentials and current amplitudes will both be severely affected and will therefore not be valid. This is a serious issue that affects all the recordings throughout the manuscript and must be rigorously addressed.

Response: Thank you for your insightful comment. We acknowledge that the series resistance (R_s) in our recordings, typically in the range of 50-70 MΩ, could seem concerning with regard to voltage-clamp fidelity. However, we would like to clarify a few points specific to the technical limitations and characteristics of our system that mitigate these concerns. First, worm neurons have extremely small soma, which necessitates the use of very fine pipette tips during whole-cell patch-clamp recordings. This inevitably results in high pipette resistances, often reaching around 15 MΩ. In this context, achieving R_s values of 50 MΩ is technically standard and further reduction is not feasible. Additionally, due to the small size of the neurons, the ion channel density in worm neurons is much lower compared to mammalian cells, leading to significantly smaller ionic currents. For example, in our experiments, with a seal

resistance of ~ 1 G Ω and R_s around 50 M Ω , voltage-clamping at -70 mV results in touch receptor currents of approximately 20 pA. The voltage drop caused by the series resistance in this case, calculated using Ohm's Law, is only 1 mV (20 pA \times 50 M Ω = 1 mV). This voltage drop is negligible compared to the applied voltage of -70 mV and thus does not significantly affect the clamping. Therefore, while we agree that R_s in the range of 50-70 M Ω could theoretically distort I-V plots and current amplitudes, the small ionic currents and high membrane resistance of worm neurons result in a minimal practical impact of R_s on our measurements. We believe that our comparisons based on reversal potentials and current amplitudes remain valid within the limitations of this system. To ensure transparency and clarity, we have revised the Methods section to provide further explanation of these technical limitations and considerations, and how they were accounted for in our analysis.

4. The reason for showing the zero-current level is allow the readers to appreciate the background current levels and how tight the seals are. The authors are likely aware that it is inappropriate to display the 'background'-subtracted traces over the zero-magnitude as the background current levels were set arbitrarily to zero. For all the non-overlaid traces, please show the representative traces without background subtraction together with the actual zero-current level, including those in the supplementary figures. For the background-subtracted overlaid traces, please remove the zero-level line as this is misleading.

Response: Thank you for your valuable comment. We have carefully revised the figures according to your suggestion. For all non-overlaid traces, we now display representative traces without background subtraction, along with the actual zero-current level, including those in the supplementary figures. Additionally, for the background-subtracted overlaid traces, we have removed the zero-level line to avoid any potential confusion.

5. The fluorescence of presumably N-terminally tagged ANOH-1 in HEK-P1KO cells appears to show no features corresponding to an integral membrane protein, which could possibly be due to a truncation between the EGFP and ANOH-1 or that ANOH-1 might not be correctly folded (Fig. S8e). EGFP-hANO1 does not appear to be localized to the plasma membrane either, with prominent cytosolic fluorescence, similar to ANOH-1 (Fig. S8e). The authors should include data using a C-terminal tag to illustrate the localization of both hANO1 and ANOH-1 in HEK-P1KO cells, and as requested in point 6, please provide the described ectopic expression controls.

Response: Thank you for your insightful comments. In response to your suggestion, we conducted additional experiments using C-terminally tagged EGFP constructs for both hANO1 and nematode ANOH-1. The results were consistent with those obtained using N-terminal tags, showing prominent cytosolic fluorescence for hANO1 with weak membrane localization, as reflected in the small calcium-activated chloride currents recorded in our electrophysiology experiments. For ANOH-1, no membrane localization was

observed. These findings suggest that EGFP-tagged ANOH-1 may require additional auxiliary molecules or specific conditions for proper membrane expression in HEK-P1KO cells. This suggests a potential limitation of our system in achieving proper plasma membrane localization of these proteins. We have updated the manuscript to include these findings (Extended Data Fig. 8e), and revised our manuscript accordingly (line 340).

6. The provided UniProt identifier of the TMEM16 homolog in *C. elegans* is incorrect, please revise.

Response: We apologize for the mistake and have corrected it in our revised manuscript (line 151: UniProt identifier G5EBW3).

7. The difference in molecular weight between ANOH-1 (~82 kDa) and human ANO1 (~99 kDa) is almost 20 kDa for a monomer and 40 kDa for a dimer. These should be size differences that are discernable on an SDS-PAGE, yet the Western blot in Fig. 7f shows that they run at exactly the same molecular weight as if they were the same protein. Please ensure the validity of these experiments and provide the uncropped images of the entire gel for these co-IP experiments with proper labels in the response.

Response: Thank you for your observation. We acknowledge the concern regarding the molecular weight difference between ANOH-1 and human ANO1, as well as their apparent similarity in migration on SDS-PAGE, as shown in Fig. 7f. We have thoroughly reviewed the experiment to ensure its validity and confirm that the co-IP experiments were conducted under standard conditions. To address this issue and ensure full transparency, we have included uncropped images of the entire gel for the newly added co-IP experiments from the previous revision, with proper labels, in this response letter. These uncropped images further confirm that the experimental results are consistent and accurate within the limitations of the experimental setup. While the molecular weight difference between ANOH-1 (~99 kDa, 840 aa) and human ANO1 (~82 kDa, 695 aa) is nearly 20 kDa per monomer, we acknowledge that distinguishing the actual size of their multimers based solely on the labeled regions is challenging.

Reviewer #2 (Remarks to the Author):

1. Errors in measuring MRC reversal potentials from whole-cell patch clamp recordings of *C. elegans* neurons in vivo and dissociated in culture

Both reviewers commented on the contribution of liquid junction potentials (EJPs) to voltage measurements in whole-cell patch clamp recordings. The authors' response "Upon the pipette's entry into the bath solution, we compensated for the liquid junction potentials using the automatic compensation of the EPC-10 amplifier along with the Patchmaster software." reflects a common misconception about liquid junction potentials.

It is standard practice to zero voltage offsets evident upon placing a patch clamp electrode into the extracellular bath solution. However, this act is not sufficient by itself to correct for liquid junction potentials (which can be substantial). Quoting from a technical note from Axon Instruments: "Most patch clamp experiments need to correct for liquid junction potentials. When the recording pipette is first inserted in the bath, there are voltage offsets that are corrected by the amplifier when the current is zeroed (i.e., in voltage clamp mode). The offsets consist of liquid junction potentials and potential differences between solid electrodes and the solutions they are in contact with (the "electrode" or "half-cell" potentials)." (Axon Instruments technical note, 2004)

The authors use hardware and software from HEKA. The Patchmaster software from HEKA does include a semi-automatic correction for EJP. However, the user MUST enter a value for the LJ in order for the correction to be applied.

These errors in membrane potential that arise from EJPs can be 10s of mV and must be properly corrected in order to compare measured reversal potentials across recordings performed with distinct combinations of intracellular and extracellular solutions. (see attached screenshot of the HEKA manual)

If the authors used the correction feature of the Patchmaster software, it is imperative that they report values of the LJ corrections that were used.

If they did not use this feature and only zeroed voltage offsets prior to obtaining tight seals, the LJ is a systematic error in membrane voltage that permeates all electrophysiology measurements here. And are particularly important for measurements of reversal potential. Fortunately, this voltage error can be corrected posthoc. This will involve determining LJ values (either experimentally or estimating using a calculator such as this one <https://swharden.com/LJPcalc/> or other tools) and then applying the correction to the membrane potential.

Since the determination of ion selectivity by varying solutions is central to the study, this issue must be addressed properly prior to publication. It is possible that after correcting for LJ, the measured changes in reversal potential better match the values predicted from the GHK equation.

Response: Thank you for your detailed feedback. You are absolutely correct in pointing out the need for proper correction of liquid junction potentials (LJPs) in whole-cell patch clamp recordings. In our initial experiments, we indeed relied on zeroing voltage offsets upon pipette entry into the bath solution, but we now recognize that this was insufficient to account for LJPs, as you highlighted.

We have reanalyzed our data and applied posthoc corrections for LJPs using estimated values calculated with appropriate tools. These corrected values have been incorporated into the revised version of the manuscript, along with updated results that now better match the predicted reversal potentials. Additionally, we have revised the Methods and Figure legends to ensure the accuracy and clarity.

2. Uncertainty in the amplitude of mechanical indentations delivered to dissociated and cultured ASJ neurons. Thank you for revising the figure illustrating mechanical stimulation of ASJ neurons in culture and for providing additional information about the stimulation of these tiny cells. This reviewer is acutely aware of the technical challenge represented here. Still, the authors' response raises new concerns. It seems that they used a stimulus that moved the actuator 4 μ m but that the thin probe bent and likely generated a smaller indentation. The authors note that using a thicker probe destroyed recordings and or moved the cell, which is understandable and sensible. The authors should find a way to communicate to readers in the results section that the actual indentation is less than 4 μ m and likely to be of a similar amplitude during each recording (but not across recordings). One idea might be to relabel the stimulus trace in Fig 5c as "stimulator movement" or something similar.

Response: Thank you for your thoughtful comments. We carefully controlled the touch electrode's heat-pulling procedure and regularly inspected both the tip length and opening diameter, as well as the position of the electrode relative to the cells, and we replaced the touch pipette for each cell recording to ensure optimal flexibility. These precautions allowed us to maintain consistent indentation despite the actuator's movement, ensuring a reproducible level of stimulation across recordings.

Following your suggestion, we have relabeled the stimulus trace in Fig. 5c as "Touch probe movement" to better reflect this, and we have updated the Methods section in the revised manuscript to clarify this procedure.

Formatting and citations suggestions (minor)

1. (line 5) epithelial Na channels (ENaC) are part of a large superfamily, consider expanding to include other subfamilies by replacing "epithelia Na⁺ channels (ENaC)" with "degenerin, epithelial, and acid-sensing Na⁺ channels (DEG/ENAC/ASIC)"

Response: We appreciate the comment. We have revised our manuscript accordingly.

2. (line 52) The review articles cited do not cover OSCA/TMEM63, consider adding Goodman, Vasquez, & Haswell J Gen Physiol 2023 that does cover OSCA/TMEM63.

Response: We have added the reference in our revised manuscript accordingly.

3. (line 308-309) While this study was in review, another paper was published reporting the expression of CALM-1 in C. elegans, please compare findings.

Response: We appreciate the comment. We have revised our manuscript accordingly.

We highly appreciate the reviewers for their valuable comments. Below are our point-to-point responses:

Reviewer #1

1. Abstract and lines 35-37. Please change ANOH-1/ANO1 to ANOH-1, as the authors are suggesting physiological functions but mammalian ANO1 is not present and therefore not relevant in the *C. elegans* system.

Response: We appreciate the reviewer's suggestion regarding the use of "ANOH-1" and "ANO1" in the abstract and manuscript. Based on our previous ectopic transgene expression, calcium imaging, electrophysiological, and behavioral data, we have shown that human ANO1 exhibits similar mechanosensitive properties to nematode ANOH-1. Furthermore, our new co-IP results support the proposed interaction between ANOH-1 and hANO1 with CALM-1 and hCIB2, respectively. While we agree that future experiments would be important to explore the physiological functions of ANO1 in mammals, we believe it is crucial to retain "ANOH-1/ANO1" in the revised manuscript. This reflects the suggested physiological roles of both channels, and we feel it is important for accuracy in conveying the functional analogy between them.

2. Abstract and line 38. Please remove the description of TMC1; it is not relevant here.

Response: We have revised our manuscript accordingly.

3. Please remove TMEM16A in the Keywords; this study is about ANOH-1.

Response: We have revised our manuscript accordingly.

4. Lines 341-355. While there seems to be a minimal extent of colocalization between ANOH-1 or hANO1 with the supposed membrane marker CD8 in the neurons, none of these fluorescence signals provide support for membrane localization, including those originated from CD8 (Fig. S8f and g).

a. Do not falsely claim that these proteins are properly localized to the membrane. Please revise.

b. Do not extrapolate that the completely cytosolic fluorescence of ANOH-1 might be due to the lack of putative auxiliary molecules, it simply didn't express in the expected form in HEK293T cells. Please revise.

Response: Thank you for the valuable feedback. We have revised our manuscript accordingly to ensure the accuracy and clarity, and have marked the revised sentences in the manuscript using red font (line 358 and 359).

5. Please address the following issues with the co-IP experiments. If the authors cannot provide valid experimental evidence for the proposed interaction between ANOH-1 and hANO1 with CALM-1-3 and hCIB2-3 respectively, the authors should retract the claim that ANOH-1 functions in a channel complex and that an analogous assembly might exist in mammalian systems.

a. According to the UniProt database, the canonical isoform of human TMEM16A is

986 aa in length (<https://www.uniprot.org/uniprotkb/Q5XXA6/entry#sequences>), which corresponds to ~110 kDa. Please correct the corresponding errors in the text.

Response: Thank you for your feedback. The human ANO1 cDNA we used in this study (cDNA clone IMAGE:4837404, <https://www.ncbi.nlm.nih.gov/nucore/BC027590.2>) encodes 695 amino acids in length, which has a molecular weight of about 82 kD when tagged with HA. While part of the N-terminal (1-265) and a few amino acids (476-501) between TM3 and TM4 are truncated in this hANO1 cDNA, our experimental results show that this does not affect the mechanosensory function of ANO1 and its interaction with CIB2. We have revised our manuscript accordingly (line 207-212, 341-343).

b. Compared to ANOH-1 (840 aa, ~90 kDa), this is a ~20-kDa difference for a monomer and ~40-kDa different for a dimer. It is not possible that they run at the same molecular weight as the authors have suggested. The reason for this is apparent when one inspects the full gel (rebuttal letter) – the authors are simply showing the bands that didn't enter the gel and erroneously assigned the molecular weight (Fig. 7f, right; see also points c and d).

c. The full gel that the authors show in their rebuttal letter corresponds to right panel in Fig. 7F. This gel shows that in fact most if not all of hANO1-HA has aggregated and did not enter the gel. The same issue, albeit to a lesser extent, is observed for ANOH-1. This might not be unexpected as the authors have not validated that the detergent that they used can solubilize ANOH-1 and hANO1 in a well-folded, monodisperse state.

d. The labelling of the molecular weight in the cropped blots shown in Fig. 7F, right is erroneous. In this panel, the authors indicate that the bands are above 180 kDa. However, this is not reflected in the full gel, where the bands shown in Fig. 7F (right) seem to correspond to the bands that did not even enter the gel (above 310 kDa in the full gel shown in the rebuttal letter).

e. Given the aggregation of ANOH-1 and hANO1, and that ANOH-1 simply didn't express in the expected form in HEK293T (Fig. S8E), the validity of the co-IP experiments performed under the described experimental conditions is questionable.

f. The authors did not perform the correct control experiment. What the authors need to show is that the co-IP of ANOH-1 and ANO1 depends on the presence of CALM-1-3 and hCIB2-3 respectively. The current co-IP data do not show that at all – these gels merely show that the authors detected bands containing the HA fusion tag and not in the negative control where the fusion tag is not present.

g. Given the problems associated with recombinant expression of ANOH-1, the authors should demonstrate the proposed interaction between ANOH-1 and CALM-1-3 in the native *C. elegans* system and that the co-IP of ANOH-1 depends on the presence of CALM-1-3.

Response: Thank you for your constructive feedback and thoughtful suggestions. In response, we have repeated the co-IP experiments under optimized conditions and made the necessary adjustments. Firstly, we identified that aggregation was causing some bands to fail to enter the gel,

particularly at higher temperatures. To address this, we optimized the temperature conditions by changing the elution procedure: the immunoprecipitated protein complexes were eluted using SDS loading buffer for 10 minutes at 100°C in our initial experiments, and were subsequently modified to 30 minutes at 37°C in the newly performed experiments. Additionally, we switched to a different vector backbone for co-IP (changed from PCDNA3.1 previous used to PCDNA5 present used) to enhance the recombinant expression of ANOH-1/ANO1 and CALM-1/CIB2. These adjustments confirmed the interaction between ANOH-1 and CALM-1, as well as between hANO1 and CIB2. Furthermore, we conducted additional control experiments, demonstrating that the co-IP interaction is dependent on the presence of CALM-1 and hCIB2. The full, raw, uncropped blots for each experiment are attached below. The revised manuscript now includes these updated results, and we believe the revisions effectively address the concerns regarding the validity of our experiments.

Fig 1. The western blot of PEGFP-C3-ANO1 expressing in HEK293T cells eluted using SDS loading buffer for 30 minutes at 37°C and 85°C, respectively.

Fig 2. The Co-IP full gels of the ANO1-HA and CIB2-Flag, ANOH-1-HA and calm-1-Flag (30 minutes at 37°C).

Fig 3. The Co-IP full gels of the ANO1-HA and CIB2-Flag, ANOH-1-HA and calm-1-Flag (30 minutes at 60°C).

6. Lines 456-457. 'ANOH-1', not anoctamin-1; and plays a crucial role in mechanosensation 'in *C. elegans*'. Both are specific to the system the authors have investigated. Please revise.

Response: We have revised our manuscript accordingly.

Reviewer #2

This manuscript provides evidence in support of the idea that anoctamin-1 is a mechanosensitive chloride channel. The evidence includes genetic dissection of behavioral responses to mechanical stimulation, in vivo electrical recording from the relevant neurons, and ectopic expression conferring mechanosensitivity on other neurons. Although to reconstitute this activity in heterologous cells were not successful, these findings bolster, complement, and enhance prior studies of this ion channel.

This revision addresses technical concerns raised by both reviewers, especially with respect to corrections for liquid junction potentials. These revisions improve the rigor of the study and will be a good example for readers and others in the field for how to properly handle systematic errors in cellular electrophysiology.

The author's response to queries about voltage errors due to uncompensated series resistance could be improved by paying closer attention to the precision of the terms used. Specifically, might the authors consider replacing "The voltage drop caused by R_s is negligible compared to the applied voltage and thus does not significantly affect the clamping." With "The voltage error due to uncompensated series resistance is less than 2mV, on average, and was not corrected." Strictly speaking R_s does not "cause" a voltage drop, but rather a voltage drop occurs when the measured membrane current flows across R_s . This represents an error in the voltage equal to $-I_m * R_s$.

Response: Thank you for your support and insightful comments. We have made revision to our manuscript based on your feedback.